# MINIBATCH VS LOCAL SGD WITH SHUFFLING: TIGHT CONVERGENCE BOUNDS AND BEYOND

**Chulhee Yun**
KAIST AI
chulhee.yun@kaist.ac.kr

**Shashank Rajput**
Univ. of Wisconsin-Madison CS
rajput3@wisc.edu

**Suvrit Sra**
MIT EECS
suvrit@mit.edu

## ABSTRACT

In distributed learning, **local SGD** (also known as federated averaging) and its simple baseline **minibatch SGD** are widely studied optimization methods. Most existing analyses of these methods assume independent and unbiased gradient estimates obtained via *with-replacement* sampling. In contrast, we study *shuffling-based* variants: **minibatch** and **local Random Reshuffling**, which draw stochastic gradients without replacement and are thus closer to practice. For smooth functions satisfying the Polyak-Łojasiewicz condition, we obtain convergence bounds (in the large epoch regime) which show that these shuffling-based variants converge *faster* than their with-replacement counterparts. Moreover, we prove matching lower bounds showing that our convergence analysis is *tight*. Finally, we propose an algorithmic modification called **synchronized shuffling** that leads to convergence rates *faster* than our lower bounds in near-homogeneous settings.

## 1 INTRODUCTION

Distributed learning within the framework of federated learning (Konečný et al., 2016; McMahan et al., 2017) has witnessed increasing interest recently. A key property of this framework is that models are trained locally using only private data on devices/machines distributed across a network, while parameter updates are aggregated and synchronized at a server.[1] Communication is often the key bottleneck for federated learning, which drives the search for algorithms that can train fast while requiring less communication—see Li et al. (2020a); Kairouz et al. (2021) for recent surveys.

A basic algorithm for federated learning is local stochastic gradient descent (SGD), also known as federated averaging. The goal is to minimize the global objective that is an average of the local objectives. In **local SGD**, we have $M$ machines and a server. After each round of communication, each of the $M$ machines locally runs $B$ steps of SGD on its local objective. Every $B$ iterations, the server aggregates the updated local iterates from the machines, averages them, and then synchronizes the machines with the average. Convergence analysis of local SGD and its variants has drawn great interest recently (Dieuleveut & Patel, 2019; Haddadpour et al., 2019; Haddadpour & Mahdavi, 2019; Stich, 2019; Yu et al., 2019; Li et al., 2020b;c; Koloskova et al., 2020; Khaled et al., 2020; Spiridonoff et al., 2020; Karimireddy et al., 2020; Stich & Karimireddy, 2020; Qu et al., 2020).

Of the many, the biggest motivation for our paper comes from the line of work by Woodworth et al. (2020a;b; 2021). In (Woodworth et al., 2020a;b), **minibatch SGD** is studied as a simple yet powerful baseline for this intermittent communication setting. Instead of locally updating the iterates $B$ times, minibatch SGD aggregates $B$ gradients (evaluated at the last synced iterate) from each of the $M$ machines, forms a minibatch of size $MB$, and then updates the shared iterate. Given the same $M$ and $B$, local SGD and minibatch SGD have the same number of gradient computations per round of communication, so it is worthwhile to understand which converges faster. Woodworth et al. (2020a;b) point out that many existing analyses on local SGD show inferior convergence rate compared to minibatch SGD. Through their new upper and lower bounds, they identify regimes where local SGD can be faster than minibatch SGD.

While the theory of local and minibatch SGD has seen recent progress, there is still a gap between what is analyzed versus what is actually used. Most theoretical results assume independent and

---

[1]A distinctive feature of federated learning is that not all devices necessarily participate in the updates; however, we focus on the full participation setting in this paper.

unbiased gradient estimates obtained via *with-replacement* sampling of stochastic gradients (i.e., choosing training data indices uniformly at random). In contrast, most practitioners use *without-replacement* sampling, where they shuffle indices randomly and access them sequentially.

Convergence analysis of without-replacement methods is challenging because gradients sampled within an epoch lack independence. As a result, the standard theory based on independent gradient estimates does not apply to shuffling-based methods. While shuffling-based methods are believed to be faster in practice (Bottou, 2009), broad theoretical understanding of such methods remains elusive, except for noteworthy recent progress mainly focusing on the analysis of SGD (Gürbüzbalaban et al., 2019; Haochen & Sra, 2019; Nagaraj et al., 2019; Nguyen et al., 2020; Safran & Shamir, 2020; 2021; Rajput et al., 2020; 2021; Ahn et al., 2020; Mishchenko et al., 2020; 2021; Tran et al., 2021). These results indicate that in the *large-epoch regime* (where the number of epochs is greater than some threshold), without-replacement SGD converges faster than with-replacement SGD.

## 1.1 OUR CONTRIBUTIONS

We analyze convergence rates of without-replacement versions of local and minibatch SGD, where local component functions are reshuffled at every epoch. We call the respective algorithms **local RR** (Algorithm 1) and **minibatch RR** (Algorithm 2), and their with-replacement counterparts **local SGD** and **minibatch SGD**. Our key contributions are as follows:

- In Section 3, we present convergence bounds on minibatch and local RR for $L$-smooth functions satisfying the $\mu$-Polyak-Łojasiewicz condition (Theorems 1 & 2). Our theorems give *high-probability* bounds, a departure from the common in-expectation bounds in the literature. We show that minibatch and local RR converge *faster* than minibatch and local SGD when the number of epochs is sufficiently large. We also identify a regime where local RR converges as fast as minibatch RR: when synchronization happens frequently enough and local objectives are not too heterogeneous. See also Appendix A for a detailed comparison with existing upper bounds.

- In Section 4, we prove that the upper bounds obtained in Section 3 are *tight*, in all factors except $L$ and $\mu$. We present Theorems 3 & 4 and Proposition 5 which show lower bounds that match the upper bound up to a factor of $L^2/\mu^2$. Our lower bound on local RR indicates that if the synchronization interval $B$ is too large, then local RR has *no gain* from parallel computation.

- In Section 5, we propose a simple modification called synchronized shuffling that allows us to *bypass* the lower bounds in Section 4, at the cost of a slight increase in communication. By having the server broadcast random permutations to local machines, we show that in near-homogeneous settings, the modified algorithms converge faster than the lower bounds (Theorems 6 & 7).

- In Appendix C, we present numerical experiments that corroborate our theoretical findings.

## 2 PROBLEM SETUP

**Notation.** For a natural number $a \in \mathbb{N}$, let $[a] := \{1, 2, \ldots, a\}$. Let $\mathcal{S}_a$ be the set of all permutations of $[a]$. Since our indices start from 1, we redefine the modulo operation between $a \in \mathbb{Z}$ and $b \in \mathbb{N}$ as $a \bmod b := a - \lfloor \frac{a-1}{b} \rfloor b$, to make $a \bmod b \in [b]$.

**Optimization task.** Consider $M$ machines, each with its objective $F^m(\boldsymbol{x}) := \frac{1}{N} \sum_{i=1}^N f_i^m(\boldsymbol{x})$, for $m \in [M]$. The $m$-th machine has access only to the gradients of its own $N$ local components $f_1^m(\boldsymbol{x}), \ldots, f_N^m(\boldsymbol{x})$. In this setting, we wish to minimize the global objective function which is an average of the local objectives: $F(\boldsymbol{x}) := \frac{1}{M} \sum_{m=1}^M F^m(\boldsymbol{x}) = \frac{1}{MN} \sum_{m=1}^M \sum_{i=1}^N f_i^m(\boldsymbol{x})$.

Further, we assume that each individual component function $f_i^m$ is $L$-smooth, so that

$$f_i^m(\boldsymbol{y}) \le f_i^m(\boldsymbol{x}) + \langle \nabla f_i^m(\boldsymbol{x}), \boldsymbol{y} - \boldsymbol{x} \rangle + \frac{L}{2} \|\boldsymbol{y} - \boldsymbol{x}\|^2, \ \text{ for all } \boldsymbol{x}, \boldsymbol{y} \in \mathbb{R}^d, \tag{1}$$

and that the global objective $F$ satisfies the $\mu$-Polyak-Łojasiewicz (PŁ) condition.[2]

$$\tfrac{1}{2} \|\nabla F(\boldsymbol{x})\|^2 \ge \mu(F(\boldsymbol{x}) - F^*) \text{ for all } \boldsymbol{x} \in \mathbb{R}^d, \quad \text{where } \mu > 0. \tag{2}$$

**Algorithms.** Under the above setting, we analyze local RR (Algorithm 1) and minibatch RR (Algorithm 2) and characterize their worst-case convergence rates.[3] The algorithms are run over $K$ epochs,

---

[2]PŁ functions can be thought as a nonconvex generalization of strongly convex functions.

[3]In Algorithms 1 and 2, consider SYNCSHUF as FALSE for now. We will discuss SYNCSHUF in Section 5.

---

**Algorithm 1** Local RR (with and without SYNCSHUF)

---

Input: Initialization $\boldsymbol{y}_0$, step-size $\eta$, # machines $M$, # components $N$, # epochs $K$, sync interval $B$.
1: Initialize $\boldsymbol{x}_{1,0}^m := \boldsymbol{y}_0$ for all $m \in [M]$.
2: **for** $k \in [K]$ **do**
3:      **if** SYNCSHUF = TRUE **then**                            ▷ Local RR with SYNCSHUF
4:          Sample $\sigma \sim \mathrm{Unif}(\mathcal{S}_N), \pi \sim \mathrm{Unif}(\mathcal{S}_M)$.
5:          Set $\sigma_k^m(i) := \sigma((i + \frac{N}{M}\pi(m)) \bmod N)$ for all $m \in [M], i \in [N]$.
6:      **else**                                                  ▷ Local RR
7:          Sample $\sigma_k^m \sim \mathrm{Unif}(\mathcal{S}_N)$ independently and locally, for all $m \in [M]$.
8:      **end if**
9:      **for** $i \in [N]$ **do**
10:          **for** $m \in [M]$ **do locally**
11:              Update $\boldsymbol{x}_{k,i}^m := \boldsymbol{x}_{k,i-1}^m - \eta\nabla f_{\sigma_k^m(i)}^m(\boldsymbol{x}_{k,i-1}^m)$.
12:          **end for**
13:          **if** $B$ divides $i$ **then**
14:              Aggregate and average $\boldsymbol{y}_{k,\frac{i}{B}} := \frac{1}{M}\sum_{m=1}^M \boldsymbol{x}_{k,i}^m$.
15:              Synchronize $\boldsymbol{x}_{k,i}^m := \boldsymbol{y}_{k,\frac{i}{B}}$, for all $m \in [M]$.
16:          **end if**
17:      **end for**
18:      $\boldsymbol{x}_{k+1,0}^m := \boldsymbol{y}_{k,\frac{N}{B}}$, for all $m \in [M]$.
19: **end for**
20: **return** the last iterate $\boldsymbol{y}_{K,\frac{N}{B}}$.

---

**Algorithm 2** Minibatch RR (with and without SYNCSHUF)

---

Input: Initialization $\boldsymbol{x}_0$, step-size $\eta$, # machines $M$, # components $N$, # epochs $K$, sync interval $B$.
1: Initialize $\boldsymbol{x}_{1,0} := \boldsymbol{x}_0$.
2: **for** $k \in [K]$ **do**
3:      **if** SYNCSHUF = TRUE **then**                      ▷ Minibatch RR with SYNCSHUF
4:          Sample $\sigma \sim \mathrm{Unif}(\mathcal{S}_N), \pi \sim \mathrm{Unif}(\mathcal{S}_M)$.
5:          Set $\sigma_k^m(i) := \sigma((i + \frac{N}{M}\pi(m)) \bmod N)$ for all $m \in [M], i \in [N]$.
6:      **else**                                            ▷ Minibatch RR
7:          Sample $\sigma_k^m \sim \mathrm{Unif}(\mathcal{S}_N)$ independently and locally, for all $m \in [M]$.
8:      **end if**
9:      **for** $i \in [\frac{N}{B}]$ **do**
10:          Update $\boldsymbol{x}_{k,i} := \boldsymbol{x}_{k,i-1} - \frac{\eta}{M}\sum_{m=1}^M \underbrace{\frac{1}{B}\sum_{j=(i-1)B+1}^{iB} \nabla f_{\sigma_k^m(j)}^m(\boldsymbol{x}_{k,i-1})}_{\text{averaging done \textbf{locally}}}$.
11:      **end for**
12:      $\boldsymbol{x}_{k+1,0} := \boldsymbol{x}_{k,\frac{N}{B}}$.
13: **end for**
14: **return** the last iterate $\boldsymbol{x}_{K,\frac{N}{B}}$.

---

i.e., $K$ passes over the entire component functions. At the beginning of epoch $k$, each machine $m$ shuffles its local component functions $\{f_i^m\}_{i=1}^N$ using a random permutation $\sigma_k^m \sim \mathrm{Unif}(\mathcal{S}_N)$. In local RR, each machine makes $B$ local RR updates to its iterate by sequentially accessing its shuffled component functions, before the server aggregates iterates from all the machines and then synchronizes the machines with the average iterate. In minibatch RR, instead of making $B$ local updates, each machine collects $B$ gradients evaluated at the last iterate, and the server aggregates them to make an update using these $MB$ gradients. Since these two algorithms use the same amount of communication and local gradients, minibatch RR is a simple yet powerful baseline for local RR.

Below, we collect our assumptions on the algorithm parameters used throughout the paper.

**Assumption 1** (Algorithm parameters). *We assume $M \geq 1$, $N \geq 2$, and $K \geq 1$. Also, assume that $B$ divides $N$. We restrict $1 \leq B \leq \frac{N}{2}$ for minibatch RR because $B = N$ makes the algorithm equal to GD. We also assume $2 \leq B \leq N$ for local RR because $B = 1$ makes the two algorithms the same. We choose a constant step-size scheme, i.e., $\eta > 0$ is kept constant over all updates.*

We next state assumptions on intra- and inter-machine deviations used in this paper.[4]

---

[4]Assumptions 2, 3 & 4 require that they hold for the whole $\mathbb{R}^d$. We discuss ways to avoid it in Appendix D.7.

**Assumption 2** (Intra-machine deviation). *There exists $\nu \geq 0$ such that for all $m \in [M]$ and $i \in [N]$,*

$$\|\nabla f_i^m(\boldsymbol{x}) - \nabla F^m(\boldsymbol{x})\| \leq \nu, \ \text{for all } \boldsymbol{x} \in \mathbb{R}^d.$$

Assumption 2 requires that the difference between the gradient of each local component function $f_i^m(\boldsymbol{x})$ and its corresponding local objective function $F^m(\boldsymbol{x})$ is uniformly bounded. It models the variance of local components $f_i^m$ *within* each machine. While the uniform boundedness requirement may look strong, we use this assumption to prove high-probability upper bounds, which are stronger than the common in-expectation bounds. See Appendix A for comparisons with other assumptions, and also Appendix D.7 for ways to avoid uniform boundedness over the *entire* $\mathbb{R}^d$.

The next two assumptions capture the deviation *across* different machines, i.e., the degree of heterogeneity, in two different levels of granularity: *objective-wise* and *component-wise*.

**Assumption 3** (Objective-wise inter-machine deviation). *There exist $\tau \geq 0$ and $\rho \geq 1$ such that*

$$\frac{1}{M} \sum_{m=1}^{M} \|\nabla F^m(\boldsymbol{x})\| \leq \tau + \rho \|\nabla F(\boldsymbol{x})\|, \ \text{for all } \boldsymbol{x} \in \mathbb{R}^d.$$

Assumption 3 models the heterogeneity by bounding the mean of $\|\nabla F^m\|$ by a constant plus a multiplicative factor times $\|\nabla F\|$. The assumption includes the homogeneous case (i.e., $F^1 = \cdots = F^M = F$) by $\tau = 0$ and $\rho = 1$. Assumption 3 is weaker than many other heterogeneity assumptions in the literature (e.g., Karimireddy et al. (2020)); see Appendix A for detailed comparisons.

Assumption 3 measures heterogeneity by only considering the local objectives $F^m$, not the local components $f_i^m$. We consider a more fine-grained notion of heterogeneity in Assumption 4:

**Assumption 4** (Component-wise inter-machine deviation). *For all $i \in [N]$, let $\bar{f}_i := \frac{1}{M} \sum_{m=1}^{M} f_i^m$. There exist $\lambda \geq 0$ such that for all $m \in [M]$ and $i \in [N]$,*

$$\left\|\nabla f_i^m(\boldsymbol{x}) - \nabla \bar{f}_i(\boldsymbol{x})\right\| \leq \lambda, \ \text{for all } \boldsymbol{x} \in \mathbb{R}^d.$$

Assumption 4 states that the gradients of the $i$-th components of local machines are "close" to each other. The assumption subsumes the component-wise homogeneous setting, i.e., $f_i^1 = f_i^2 = \cdots = f_i^M$, by $\lambda = 0$. In distributed learning, this choice corresponds to the setting where each machine has the same training dataset. Assumption 4 with $\lambda > 0$ is also relevant to the case where each device has a slightly perturbed (e.g., by data augmentation techniques) version of a certain dataset. It is straightforward to check that Assumption 4 implies Assumption 3 with $\tau = \lambda$ and $\rho = 1$.

We conclude this section by defining the function classes we study in this paper.

**Definition 1** (Function classes). *We consider two classes of global objective functions $F$, also taking into account their local objectives $F^m$ and local components $f_i^m$. We assume throughout that $f_i^m$ are differentiable and $F$ is bounded from below.*

$$\mathcal{F}_{\text{obj}}(L, \mu, \nu, \tau, \rho) := \big\{ F \mid F \text{ is } \mu\text{-P\L}; \ f_i^m \text{ are } L\text{-smooth}; \ F, F^m, f_i^m \text{ satisfy Assumptions 2 \& 3} \big\},$$

$$\mathcal{F}_{\text{cmp}}(L, \mu, \nu, \lambda) := \big\{ F \mid F \text{ is } \mu\text{-P\L}; \ f_i^m \text{ are } L\text{-smooth}; \ F, F^m, f_i^m \text{ satisfy Assumptions 2 \& 4} \big\}.$$

Notice that $\mathcal{F}_{\text{obj}}(L, \mu, \nu, \tau, \rho) \supset \mathcal{F}_{\text{cmp}}(L, \mu, \nu, \tau)$ for any $\rho \geq 1$. We only make the P\L assumption on the *global objective $F$*, not on the *local objectives $F^m$* nor on the *local components $f_i^m$*. Using $L$ and $\mu$, we define the *condition number* $\kappa := {}^L/_\mu \geq 1$.

## 3 CONVERGENCE ANALYSIS OF MINIBATCH AND LOCAL RR

### 3.1 UPPER BOUND FOR MINIBATCH RR

We first begin with the convergence result for minibatch RR on $\mathcal{F}_{\text{obj}}(L, \mu, \nu, \tau, \rho)$, which exhibits a faster large-epoch rate compared to the single-machine setting. For upper bounds, we use $\tilde{\mathcal{O}}(\cdot)$ to hide universal constants and logarithmic factors of $\frac{1}{\delta}$, $M$, $N$, $K$, and $B$.

**Theorem 1** (Upper bound for minibatch RR). *Suppose that minibatch RR has parameters satisfying Assumption 1. For any $F \in \mathcal{F}_{\text{obj}}(L, \mu, \nu, \tau, \rho)$, consider running the algorithm using step-size $\eta = \frac{B \log(MNK^2)}{\mu NK}$ for epochs $K \geq 6\kappa \log(MNK^2)$. Then, with probability at least $1 - \delta$,*

$$F(\boldsymbol{x}_{K, \frac{N}{B}}) - F^* \leq \frac{F(\boldsymbol{x}_0) - F^*}{MNK^2} + \tilde{\mathcal{O}}\left(\frac{L^2}{\mu^3} \frac{\nu^2}{MNK^2}\right). \tag{3}$$

*Proof.* The proof is in Appendix D.2. The key challenge in the convergence analysis of our shuffling-based method stems from the indices sampled within an epoch being dependent on each other. For example, if $f_1^m$ is accessed already, then the index $i = 1$ will not be used in later iterations of the epoch; this dependence significantly complicates the analysis. Our approach starts with realizing that for any permutation $\sigma$, $\sum_{i=1}^{N} f_{\sigma(i)}^m = NF^m$. We decompose gradients $\nabla f_{\sigma_k^m(j)}^m(\boldsymbol{x}_{k,i-1})$ (see Line 10 of Algorithm 2) into $\nabla f_{\sigma_k^m(j)}^m(\boldsymbol{x}_{k,0})$ plus noise, then aggregate all updates over an epoch to get "one big step of GD plus noise": $\boldsymbol{x}_{k+1,0} = \boldsymbol{x}_{k,0} - \eta N \nabla F(\boldsymbol{x}_{k,0}) + \eta^2 \boldsymbol{r}_k$. We bound the noise $\boldsymbol{r}_k$ using Lemma 8 (Appendix D.6), which is our extension of the Hoeffding-Serfling inequality to the mean of $M$ independent without-replacement sums of vectors; the lemma might be of independent interest too. Lemma 8 shows that averaging accumulated gradients over $M$ machines reduces variance by $M$, which leads to the reduction by a factor of $M$ in the bound (3). □

Theorem 1 shows that for large enough epochs $K \gtrsim \kappa$, minibatch RR converges at a rate of $\tilde{\mathcal{O}}(\frac{L^2\nu^2}{\mu^3 MNK^2})$, with high probability. Compared to the large-epoch rate $\tilde{\mathcal{O}}(\frac{L^2\nu^2}{\mu^3 NK^2})$ of single-machine RR (e.g., Ahn et al. (2020)), we see an additional factor $M$ in the denominator, which highlights the advantage of multiple machines. If we compare against the with-replacement counterpart, it is known that for strongly convex and smooth $F$, the optimal convergence rate of minibatch SGD is $\Theta(\frac{\nu^2}{\mu MNK})$,[5] which is worse than our bound (3) if $K \gtrsim \kappa^2$. Also notable is that the convergence rate does *not* depend on the heterogeneity constants (i.e., $\tau$ and $\rho$ from Assumption 3) of the local objective functions. This observation that minibatch RR is "immune" to heterogeneity is consistent with minibatch SGD in the with-replacement setting (Woodworth et al., 2020b).

**Epoch vs communication complexity.** One might wonder why (3) does not have the batch size $B$. In (3), we wrote convergence rates in terms of epochs $K$, which captures the gradient computation complexity because the same number of gradients are evaluated in a single epoch regardless of $B$. If we are interested in communication complexity instead, we can write (3) in terms of the number of communication rounds $R := \frac{NK}{B}$ and get a rate of $\tilde{\mathcal{O}}(\frac{L^2\nu^2 N}{\mu^3 MB^2 R^2})$. From these, we can also discuss the *overall cost* of the algorithm. If the cost of a communication round is $c_c$, and the cost of local gradient computations over an epoch is $c_e$, then the total cost to obtain an $\epsilon$-accurate solution is

$$C_{\text{minibatch}}(\epsilon) = \tilde{\mathcal{O}}\left(\frac{c_c \nu \sqrt{N}}{B\sqrt{M\epsilon}} + \frac{c_e \nu}{\sqrt{MN\epsilon}}\right), \tag{4}$$

omitting $L$ and $\mu$ for simplicity. The total cost shows that there is essentially no harm increasing the batch size $B$ in minibatch RR, as we can get more accurate estimates of true gradients as $B$ becomes larger. In the next subsection, we will see that this is not the case in local RR.

**What about $K \lesssim \kappa$?** We remark that all upper bounds in this paper hold only for the "large-epoch" regime, where $K \gtrsim \kappa$. Such requirements are common in the literature of without-replacement SGD (Haochen & Sra, 2019; Nagaraj et al., 2019; Rajput et al., 2020; Ahn et al., 2020), and there is a recent result (Safran & Shamir, 2021) suggesting that faster convergence of without-replacement SGD may *not* be possible in the $K \lesssim \kappa$ regime. We defer a more detailed discussion on this regime to Section 4, after Theorem 3.

## 3.2 UPPER BOUND FOR LOCAL RR

Next, we are interested in how fast local RR can converge, what is the optimal batch size $B$, and whether local RR can be as fast as minibatch RR.

**Theorem 2** (Upper bound for local RR). *Suppose that local RR has parameters satisfying Assumption 1. For any $F \in \mathcal{F}_{\text{obj}}(L, \mu, \nu, \tau, \rho)$, consider running the algorithm using step-size $\eta = \frac{\log(MNK^2)}{\mu NK}$ for epochs $K \geq 7\rho\kappa \log(MNK^2)$. Then, with probability at least $1 - \delta$,*

$$F(\boldsymbol{y}_{K,\frac{N}{B}}) - F^* \leq \frac{F(\boldsymbol{y}_0) - F^*}{MNK^2} + \tilde{\mathcal{O}}\left(\frac{L^2}{\mu^3}\left(\frac{\nu^2}{MNK^2} + \frac{\nu^2 B}{N^2 K^2} + \frac{\tau^2 B^2}{N^2 K^2}\right)\right). \tag{5}$$

---

[5]The optimal rate for (with-replacement) SGD after $R$ iterations is $\Theta(\frac{\nu^2}{\mu R})$ (see e.g., Rakhlin et al. (2012)). With-replacement minibatching reduces the variance $\nu^2$ to $\frac{\nu^2}{MB}$, and $R = \frac{NK}{B}$. However, achieving the optimal rate for *last iterates* typically requires carefully designed step-size schemes (Jain et al., 2019).

*Proof.* The proof is in Appendix D.3. We take the same "big GD step plus noise" approach as in Theorem 1; however, due to local updates, bounding the noise is much more involved. In the proof, we obtain the epoch update $\boldsymbol{y}_{k+1,0} = \boldsymbol{y}_{k,0} - \eta N \nabla F(\boldsymbol{x}_{k,0}) + \eta^2 \boldsymbol{r}_{k,1} + \eta^2 \boldsymbol{r}_{k,2} - \eta^3 \boldsymbol{r}_{k,3}$, where $\boldsymbol{r}_{k,1}$ and $\boldsymbol{r}_{k,3}$ contain errors introduced by local updates. Noise from local updates accumulates over $B$ iterations, which cannot be remedied by averaging over $M$ machines. They result in two additional terms in the rate (5), one from intra-machine variance and the other from heterogeneity. □

### 3.2.1 DISCUSSION OF THEOREM 2

Let us compare our high-probability bound (5) with existing in-expectation bounds. For strongly convex $F$, the corresponding *last-iterate* bound of local SGD is $\tilde{\mathcal{O}}(\frac{L\nu^2}{\mu^2 MNK} + \frac{L^2\nu^2 B}{\mu^3 N^2 K^2} + \frac{L^2\tau^2 B^2}{\mu^3 N^2 K^2})^6$ (Khaled et al., 2020; Spiridonoff et al., 2020; Qu et al., 2020). Notice that (5) is better than this with-replacement bound when $K \gtrsim \kappa$. For *average iterates*, there are known bounds $\tilde{\mathcal{O}}(\frac{\nu^2}{\mu MNK} + \frac{L\nu^2 B}{\mu^2 N^2 K^2} + \frac{L\tau^2 B^2}{\mu^2 N^2 K^2})^6$ (Koloskova et al., 2020; Woodworth et al., 2020b) which are smaller than the last-iterate bound by a factor of $\kappa$. It is unclear if averaging iterates could improve our rate, because most such analyses exploit Jensen's inequality, which we cannot use for nonconvex $F$.

**Dependence on $\tau$ and $\rho$.** Out of the two heterogeneity constants $\tau$ and $\rho$ (Assumption 3), $\rho$ does not appear in (5), and it only affects the epoch requirement $K \gtrsim \rho\kappa$. Consider the case $\tau = 0$ and $\rho > 1$, which is heterogeneous but in the "interpolation regime," because $\nabla F^m(\boldsymbol{x}) = \boldsymbol{0}$ whenever $\nabla F(\boldsymbol{x}) = \boldsymbol{0}$. In such a case, the rate (5) is equal to the homogeneous case.

**Using $B = \Theta(N)$ is no better than single-machine.** A close look at Theorem 2 reveals a rather surprising fact. Even in the homogeneous case ($\tau = 0$), if we choose $B = \Theta(N)$, then local RR converges at the rate of $\tilde{\mathcal{O}}(\frac{1}{NK^2})$: the same rate as the single-machine RR! In Section 4, we show that this observation is not due to a suboptimal analysis; the rate $\tilde{\mathcal{O}}(\frac{1}{NK^2})$ is tight for $B = \Theta(N)$.

**Trade-off in the choice of $B$.** As done for Theorem 1, we can compute from (5) that the total cost of local RR for $\epsilon$-accuracy is (omitting $L$ and $\mu$ for simplicity)

$$C_{\text{local}}(\epsilon) = \tilde{\mathcal{O}}\left( c_c \left( \frac{\nu\sqrt{N}}{B\sqrt{M}\epsilon} + \frac{\nu}{\sqrt{B}\epsilon} + \frac{\tau}{\sqrt{\epsilon}} \right) + c_e \left( \frac{\nu}{\sqrt{MN}\epsilon} + \frac{\nu\sqrt{B}}{N\sqrt{\epsilon}} + \frac{\tau B}{N\sqrt{\epsilon}} \right) \right). \quad (6)$$

Note that for local RR, there exists a trade-off between communication and epoch complexity in the choice of $B$. If $B$ is too small, this reduces the number of epochs required but increases communication costs. On the other hand, if $B$ is too large, this reduces communication rounds but errors that accumulate in local updates get severer, resulting in the need for more epochs. Hence, the optimal choice of $B$ must balance the two complexity measures. The existence of this trade-off is indeed different from minibatch RR where larger $B$ always reduces the total cost $C_{\text{minibatch}}(\epsilon)$.

**When can local RR match minibatch RR?** Comparing the convergence rates (3) and (5), we can identify some regimes in which local RR converges as fast as minibatch RR. In a nutshell, if machines are not too heterogeneous and communication happens frequently, then local RR can have the same upper bound as minibatch RR. For example, if $B$ is chosen to be a constant, $M \lesssim N$, and $\tau \lesssim \nu\sqrt{N/M}$, then the $\tilde{\mathcal{O}}(\frac{L^2\nu^2}{\mu^3 MNK^2})$ term in (5) becomes the dominating factor and hence matches (3). Another example of such a regime is when $B \lesssim \frac{N}{M}$ and $\tau \lesssim \nu\sqrt{M/N}$. Note that this comparison assumes that the same values of $B$ are chosen for both algorithms. Also, such "frequent communication" regimes are favorable if the communication cost $c_c$ is small.

**Can local RR ever beat minibatch RR?** The upper bounds (3) and (5) indicate that local RR is always no better than minibatch RR, at least for the function class $\mathcal{F}_{\text{obj}}(L, \mu, \nu, \tau, \rho)$. This is in fact consistent with Woodworth et al. (2020a;b), because the authors identify a regime where local SGD performs better than minibatch SGD for convex objective functions, but fail to do so for strongly convex functions. However, as was also pointed out in Woodworth et al. (2020a), there is a simple extreme scenario in which local RR can be faster: when $\nu \approx \tau \approx 0$ and $\rho \approx 1$. In this case, we have $f_i^m \approx F$ for all $m$ and $i$, so local RR corresponds to $NK$ steps of GD, whereas minibatch RR corresponds to $\frac{NK}{B}$ steps of GD. Clearly, local RR will converge faster, exploiting the advantage of more updates. Finding out other such regimes is an important future direction.

---

[6] Due to differences in assumptions, many existing rates cannot be compared directly. These rates are the ones we consider "comparable" to our bound. See Appendix A for more detailed comparisons.

## 4 MATCHING LOWER BOUNDS

In Section 3, we presented large-epoch upper bounds (i.e., for $K \gtrsim \kappa$) for constant step-size mini-batch and local RR. In this section, we prove matching lower bounds to show that the upper bounds are tight, in all factors except $L$ and $\mu$. We use $\Omega(\cdot)$ to hide universal constants in lower bounds.

### 4.1 LOWER BOUND FOR MINIBATCH RR

**Theorem 3** (Lower bound for minibatch RR). *Suppose that minibatch* RR *has parameters satisfying Assumption 1. Additionally, assume that $N$ is a multiple of* 2. *Then, there exist large enough constants $c_1, c_2 > 0$ such that the following holds: For $L$ and $\mu$ satisfying $\kappa = \frac{L}{\mu} \geq c_1$, there exists a function $F \in \mathcal{F}_{\mathrm{cmp}}(L, \mu, \nu, 0)$ such that for any constant step-size $\eta$,*

$$\mathbb{E}\left[ F(\boldsymbol{x}_{K, \frac{N}{K}}) - F^* \right] = \begin{cases} \Omega\left( \frac{\nu^2}{\mu M N K} \right) & \text{if } K < c_2 \kappa, \\ \Omega\left( \frac{\nu^2}{\mu M N K^2} \right) & \text{if } K \geq c_2 \kappa. \end{cases} \tag{7}$$

*Proof.* We prove Theorem 3 in Appendix E. The proof is an extension of Rajput et al. (2020); Safran & Shamir (2020; 2021) to minibatch RR. We will sketch some key intuitions after Theorem 4. $\quad\square$

First notice that the function $F$ is from $\mathcal{F}_{\mathrm{cmp}}(L, \mu, \nu, 0)$, where all the machines are component-wise homogeneous. As seen in Definition 1, $\mathcal{F}_{\mathrm{cmp}}(L, \mu, \nu, 0) \subset \mathcal{F}_{\mathrm{obj}}(L, \mu, \nu, \tau, \rho)$ for any $\tau \geq 0$ and $\rho \geq 1$, so Theorem 3 provides a lower bound for $\mathcal{F}_{\mathrm{cmp}}(\cdot)$ and $\mathcal{F}_{\mathrm{obj}}(\cdot)$, with arbitrary heterogeneity constants. We assume that $N$ is even because we construct functions $g_1$ and $g_2$ such that $f_i^m := g_1$ if $i \leq \frac{N}{2}$, and $f_i^m := g_2$ if $i > \frac{N}{2}$. One can remove this assumption by using a zero function when $N$ is odd (see e.g., Safran & Shamir (2020)). It is rather unsatisfactory that our theorem requires large enough constants $c_1$ and $c_2$; we believe a tighter analysis can relax this restriction.

Theorem 3 proves lower bounds for two different regimes: $K \gtrsim \kappa$ and $K \lesssim \kappa$. In the large-epoch regime ($K \gtrsim \kappa$), we can observe that the lower bound $\Omega(\frac{\nu^2}{\mu M N K^2})$ matches the upper bound (3) in Theorem 1, modulo a factor of $\kappa^2$. Tightening the $\kappa^2$ gap between upper and lower bounds is left for future work. In the small-epoch regime ($K \lesssim \kappa$), we observe that the lower bound $\Omega(\frac{\nu^2}{\mu M N K})$ exactly matches the convergence rate of (with-replacement) minibatch SGD; hence, the lower bound implies that minibatch RR has no hope for faster convergence than minibatch SGD, at least in the constant step-size and small-epoch regime. This observation is in line with Safran & Shamir (2021).

**Upper bounds for $K \lesssim \kappa$?** Even for single-machine RR ($M = 1$), proving an upper bound that matches the small-epoch lower bound $\Omega(\frac{\nu^2}{\mu N K})$ still remains a challenge. Nagaraj et al. (2019, Theorem 2) prove an upper bound for non-quadratic strongly convex functions that matches $\Omega(\frac{\nu^2}{\mu N K})$ if $N K \gtrsim \kappa^2$; however, they use suffix averaging, so it is not directly comparable to Theorem 3 which considers last iterates. Safran & Shamir (2021) prove upper bounds for quadratic strongly convex functions, but assume that their Hessian matrices commute. For noncommutative cases, proving a small-epoch upper bound seems to require some form of matrix AM-GM inequalities, whose availability is an open problem (Recht & Ré, 2012; Lai & Lim, 2020; De Sa, 2020; Yun et al., 2021).

**Remark 1** (Strong convexity in construction). We note that all lower bounds in this paper are constructed with strongly convex functions, a stronger assumption than PŁ functions (2). Thus, our lower bounds are also applicable to strong convexity counterparts of $\mathcal{F}_{\mathrm{obj}}(\cdot)$ and $\mathcal{F}_{\mathrm{cmp}}(\cdot)$.

### 4.2 LOWER BOUNDS FOR LOCAL RR

In this subsection, we present lower bounds for local RR. We prove two bounds that correspond to homogeneous and heterogeneous cases. By combining the two bounds, we get a lower bound that matches our upper bound (5) in Theorem 2 up to a factor of $\kappa^2$.

**Theorem 4** (Lower bound for local RR: homogeneous case). *Suppose that local* RR *has parameters satisfying Assumption 1. Additionally, assume that $B$ is a multiple of* 4. *Then, there exist large enough constants $c_3, c_4 > 0$ such that the following holds: For $L$ and $\mu$ satisfying $\kappa = \frac{L}{\mu} \geq c_3$, there exists a function $F \in \mathcal{F}_{\mathrm{cmp}}(L, \mu, \nu, 0)$ such that for any constant step-size $\eta$,*

$$\mathbb{E}\left[ F(\boldsymbol{y}_{K, \frac{N}{K}}) - F^* \right] = \begin{cases} \Omega\left( \frac{\nu^2}{\mu M N K} \right) & \text{if } K < \max\left\{ c_4 \kappa, \frac{MB}{N} \right\}, \\ \Omega\left( \frac{\nu^2}{\mu M N K^2} + \frac{\nu^2 B}{\mu N^2 K^2} \right) & \text{if } K \geq \max\left\{ c_4 \kappa, \frac{MB}{N} \right\}. \end{cases} \tag{8}$$

*Proof.* The proof is in Appendix G. For the large-epoch lower bounds in Theorems 3 and 4, we use "skewed" quadratics $f_i^m(x) = (L1_{x \leq 0} + \mu 1_{x>0})\frac{x^2}{2} + z_i \nu x$, where $z_i = +1$ if $i \leq \frac{N}{2}$ and $z_i = -1$ otherwise. For $x \approx 0$, the imbalance results in a "drift" towards positive $x$, whose strength is approximately proportional to the absolute value of partial sums of random permutations over $\frac{N}{2}$ $+1$'s and $\frac{N}{2}$ $-1$'s. By averaging the sums over $M$ machines (minibatch RR), their absolute values shrink by $\frac{1}{\sqrt{M}}$; in contrast, if each machine makes local updates (local RR), the magnitude of the drift cannot be reduced with $M$, because we average after local iterates already have taken $B$ "big" steps. The proof uses techniques from Rajput et al. (2020). □

**Proposition 5** (Lower bound for local RR: heterogeneous case). *Suppose that local RR has parameters satisfying Assumption 1. Additionally, assume that $B$ is a multiple of $2$ and $\kappa = \frac{L}{\mu} \geq 2$. Then, there exists a function $F \in \mathcal{F}_{\mathrm{obj}}(L, \mu, 0, \tau, 1)$ such that for any constant step-size $\eta$,*

$$\mathbb{E}\left[F(\boldsymbol{y}_{K, \frac{N}{K}}) - F^*\right] = \Omega\left(\frac{\tau^2 B^2}{\mu N^2 K^2}\right). \tag{9}$$

*Proof.* We note that Proposition 5 is almost identical to Theorem II of Karimireddy et al. (2020); however, we provide a proof specific to our algorithm in Appendix I. □

Theorem 4 constructs a component-wise homogeneous function from $\mathcal{F}_{\mathrm{cmp}}(L, \mu, \nu, 0)$ and Proposition 5 constructs a heterogeneous function from $\mathcal{F}_{\mathrm{obj}}(L, \mu, 0, \tau, 1)$. Since $\mathcal{F}_{\mathrm{cmp}}(L, \mu, \nu, 0) \cup \mathcal{F}_{\mathrm{obj}}(L, \mu, 0, \tau, 1) \subset \mathcal{F}_{\mathrm{obj}}(L, \mu, \nu, \tau, \rho)$ for any $\rho \geq 1$, combining (8) and (9) for the $K \geq \max\{c_4 \kappa, \frac{MB}{N}\}$ case gives a lower bound $\Omega(\max\{\frac{\nu^2}{\mu M N K^2} + \frac{\nu^2 B}{\mu N^2 K^2}, \frac{\tau^2 B^2}{\mu N^2 K^2}\})$ that matches the large-epoch upper bound (5) in Theorem 2, up to a factor of $\kappa^2$. When $\kappa N \gtrsim MB$, $c_4 \kappa$ becomes the dominating term in the $\max$, in which case the threshold in (8) is $\Theta(\kappa)$. Tightening the $\kappa^2$ gap as well as removing additional requirements such as $\kappa \geq c_3$ and $\kappa N \gtrsim MB$ are left for future work.

**Using $B = \Theta(N)$ does not help, indeed.** In Section 3.2.1, we observed that if $B = \Theta(N)$, then even in the homogeneous case ($\tau = 0$), local RR converges at the rate of $\tilde{\mathcal{O}}(\frac{1}{NK^2})$. This is the same rate as single-machine RR, meaning the efforts by $M - 1$ machines become meaningless. Our lower bound (8) shows that $\tilde{\mathcal{O}}(\frac{1}{NK^2})$ is in fact the best we can hope for (treating $L$ and $\mu$ as constants). In order to make the best use of $M$ machines, $B$ should be smaller than $\Theta(N)$, as suggested in Section 3.2.1. In an existing work, Mishchenko et al. (2021) consider local RR with $B = N$ as a special case of a proximal algorithm. In Theorem 8 of Mishchenko et al. (2021), the authors claim "the convergence bound improves with the number of devices involved" because the bound has a factor of $M$ in the denominator. However, at least under our assumption, this is not the case; if we apply our Assumption 3 to upper-bound their $\sigma_*$, the term "$N\sigma_*^2$" in the numerator grows linearly with $M$. Hence, our bounds do not contradict Mishchenko et al. (2021); see Appendix A for details.

**Remark 2** (Small-epoch bound is likely loose). We note that while we focused on deriving a matching large-epoch lower bound, we did not try hard to tighten the small-epoch lower bound. Our small-epoch lower bound in (8) misses a term (such as $\frac{\nu^2 B}{\mu N^2 K^2}$) that corresponds to the error from local updates. We leave investigations on small-epoch lower and upper bounds for future work.

## 5 SYNCHRONIZED SHUFFLING: HOW TO BYPASS LOWER BOUNDS

Recall from the total complexity of minibatch RR (4) that the total cost shrinks with a factor of $\frac{1}{\sqrt{M}}$. Using $M$ machines, we are only getting a $\sqrt{M}$-factor speedup. Ideally, we hope to see a *linear speedup*, i.e., cost inverse proportional to $M$. Hence, Theorem 1 falls short of achieving this goal, and our lower bound in Theorem 3 confirms that linear speedup is indeed impossible.

In this section, we show that the desired linear speedup is possible, at least in some special cases. We consider the component-wise near-homogeneous case (i.e., Assumption 4 with small $\lambda$) and discuss how a simple modification to minibatch and local RR can let us "break" the lower bounds and achieve linear speedup. This comes at a cost of broadcasting permutations: at the beginning of the $k$-th epoch, the server samples $\sigma \sim \mathrm{Unif}(\mathcal{S}_N)$ and $\pi \sim \mathrm{Unif}(\mathcal{S}_M)$, and broadcasts them to the machines. Then, local machines choose their permutations $\sigma_k^m$ to be *shifted* versions of $\sigma$,[7] i.e., $\sigma_k^m(i) := \sigma\left(\left(i + \frac{N}{M}\pi(m)\right) \mod N\right)$. We call this trick **synchronized shuffling**, denoted as

---

[7]We assume for simplicity that $M$ divides $N$.

SYNCSHUF. Please revisit Algorithms 1 and 2 for the precise descriptions of the modified algorithms **local RR with SYNCSHUF** and **minibatch RR with SYNCSHUF**, respectively.

The intuition why this should help is simple. In the proof of RR, we aggregate the component gradients over an epoch (i.e., $N$ iterations) to write it as a full gradient plus noise. If we are in the component-wise homogeneous setting and permutations are synchronized, then instead of aggregating $N$ component gradients on a single machine, we can aggregate $\frac{N}{M}$ component gradients on $M$ machines to get a full gradient. This allows us to reduce the "noise" from without-replacement sampling. We emphasize here that we do not necessarily set $B = \frac{N}{M}$ to get a full gradient every time; our analysis works for arbitrary $B$ and $M$, as long as both divide $N$. See Appendix B for a detailed illustration of SYNCSHUF; also, see Appendix C for experiments showing its effectiveness.

The idea of synchronized shuffling is similar to approaches in distributed learning that shuffle and partition datasets and distribute them to local machines (see e.g., Lee et al. (2017); Meng et al. (2017)). In contrast, we do not communicate data, but communicate how to permute datasets stored in local machines. Meng et al. (2017, Theorem 3.3) provide an analysis for a distributed method similar to minibatch RR, but fail to show convergence to global minima in strongly convex cases. We also note that an independent concurrent result (Szlendak et al., 2021) uses the same idea as SYNCSHUF to build compressors for communication-efficient distributed optimization.

## 5.1 UPPER BOUNDS FOR MINIBATCH AND LOCAL RR WITH SYNCSHUF

With SYNCSHUF, we can show that the $M$'s appearing in the convergence rates ((3) and (5)) in Theorems 1 and 2 can be replaced with $M^2$, for a more stringent function class $\mathcal{F}_{\mathrm{cmp}}(\cdot)$ that requires bounded component-wise inter-machine deviation (Assumption 4).

**Theorem 6** (Upper bound for minibatch RR with SYNCSHUF). *Suppose that minibatch* RR *with* SYNCSHUF *has parameters satisfying Assumption 1. Additionally assume that $M$ divides $N$. For any $F \in \mathcal{F}_{\mathrm{cmp}}(L, \mu, \nu, \lambda)$, consider running the algorithm using step-size $\eta = \frac{B \log(M^2 N K^2)}{\mu N K}$ for epochs $K \geq 6\kappa \log(M^2 N K^2)$. Then, with probability at least $1 - \delta$,*

$$F(\boldsymbol{x}_{K, \frac{N}{B}}) - F^* \leq \frac{F(\boldsymbol{x}_0) - F^*}{M^2 N K^2} + \tilde{\mathcal{O}}\left(\frac{L^2}{\mu^3}\left(\frac{\nu^2}{M^2 N K^2} + \frac{\lambda^2}{M K^2}\right)\right). \tag{10}$$

The proof of Theorem 6 is presented in Appendix D.4. One can check that if the component-wise deviation constant $\lambda$ satisfies $\lambda \lesssim \frac{\nu}{\sqrt{MN}}$ (i.e., near-homogeneous), then the rate (10) becomes $\tilde{\mathcal{O}}(\frac{1}{M^2 N K^2})$. It is then easy to confirm that $M$ machines reduce total costs by $\frac{1}{M}$—a linear speedup.

A similar speedup can be shown for local RR. In Appendix D.5, we prove that

**Theorem 7** (Upper bound for local RR with SYNCSHUF). *Suppose that local* RR *with* SYNCSHUF *has parameters satisfying Assumption 1. Additionally assume that $M$ divides $N$. For any $F \in \mathcal{F}_{\mathrm{cmp}}(L, \mu, \nu, \lambda)$, consider running the algorithm with step-size $\eta = \frac{\log(M^2 N K^2)}{\mu N K}$ for epochs $K \geq 7\kappa \log(M^2 N K^2)$. Then, with probability at least $1 - \delta$,*

$$F(\boldsymbol{y}_{K, \frac{N}{B}}) - F^* \leq \frac{F(\boldsymbol{y}_0) - F^*}{M^2 N K^2} + \tilde{\mathcal{O}}\left(\frac{L^2}{\mu^3}\left(\frac{\nu^2}{M^2 N K^2} + \frac{\nu^2 B}{N^2 K^2} + \frac{\lambda^2 B^2}{N^2 K^2} + \frac{\lambda^2}{M K^2}\right)\right). \tag{11}$$

We can similarly check that if $B \lesssim \frac{N}{M^2}$ and $\lambda \lesssim \frac{\nu}{\sqrt{MN}}$, i.e., frequent communication and near-homogeneity, then the $\tilde{\mathcal{O}}(\frac{1}{M^2 N K^2})$ term dominates in (11), and hence gives a linear speedup that matches the best rate of minibatch RR with SYNCSHUF (10). Nevertheless, we note again that for local RR, such a small $B$ is favorable only when the communication cost $c_c$ is small (recall (6)).

## 6 CONCLUSION

We studied convergence bounds for local RR and minibatch RR, which are the practical without-replacement versions of local and minibatch SGD studied in the theory literature. For smooth functions satisfying the Polyak-Łojasiewicz condition, we showed large-epoch convergence bounds for minibatch and local RR that are faster than their with-replacement counterparts. We also proved matching lower bounds showing that our convergence analysis is tight. We also proposed a simple modification called synchronized shuffling that leads to convergence rates faster than our lower bounds in near-homogeneous settings. Immediate future research directions include extension to small-epoch regimes, as well as to general convex and nonconvex functions.

## ETHICS STATEMENT

This paper develops theoretical guarantees for popular distributed stochastic optimization algorithms. Therefore, the authors do not see any particular concerns related to its ethical aspects or future societal consequences.

## REPRODUCIBILITY STATEMENT

This paper is a theoretical work, without any experimental results. Definitions and assumptions are provided in Section 2. Our theoretical contributions as well as some additionally required assumptions are clearly stated in Sections 3, 4, and 5. Complete proofs of all the theorems are provided in the appendix.

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

## CONTENTS

## A   COMPARISONS WITH ASSUMPTIONS AND RATES IN EXISTING RESULTS

In this section, we compare our assumptions and convergence bounds against other existing results mentioned in the main text. Most existing results that study independent and unbiased gradient estimates state their assumptions in terms of the expectation over the randomness in the estimate; for such assumptions, we adapt them to our finite sum setting in order to make for easier comparison.

**Heterogeneity assumptions.**   We start by discussing our definition of objective-wise heterogeneity (Assumption 3), namely that there exist $\tau \geq 0$ and $\rho \geq 1$ such that

$$\frac{1}{M} \sum_{m=1}^{M} \|\nabla F^m(\boldsymbol{x})\| \leq \tau + \rho \|\nabla F(\boldsymbol{x})\|, \quad \text{for all } \boldsymbol{x} \in \mathbb{R}^d. \tag{12}$$

Perhaps the most relevant to this assumption is the $(G, B)$-BGD assumption that appears in Karim-ireddy et al. (2020): For all $\boldsymbol{x} \in \mathbb{R}^d$,

$$\frac{1}{M} \sum_{m=1}^{M} \|\nabla F^m(\boldsymbol{x})\|^2 \leq G^2 + B^2 \|\nabla F(\boldsymbol{x})\|^2. \tag{13}$$

Note that thanks to Jensen's inequality and $a^2 + b^2 \leq (a+b)^2$ for $a, b \geq 0$, (13) implies

$$\left(\frac{1}{M} \sum_{m=1}^{M} \|\nabla F^m(\boldsymbol{x})\|\right)^2 \leq \frac{1}{M} \sum_{m=1}^{M} \|\nabla F^m(\boldsymbol{x})\|^2 \leq G^2 + B^2 \|\nabla F(\boldsymbol{x})\|^2 \leq (G + B \|\nabla F(\boldsymbol{x})\|)^2,$$

and hence (12) with $\tau = G$ and $\rho = B$. Therefore, our Assumption 3 is weaker than the $(G, B)$-BGD assumption. Several papers (Haddadpour & Mahdavi, 2019; Li et al., 2020b) use the same

assumption (13), with $G = 0$; therefore, Assumption 3 also subsumes the heterogeneity assumption posed in these papers. Note that $G = 0$ implies that, the minima for $F$ are also the minima for $F^m$, for every $m$, and hence $G = 0$ results in a weak form of heterogeneity.

Some papers (Yu et al., 2019; Li et al., 2020c; Qu et al., 2020) assume bounded local gradients: for all $m \in [M]$ and $\boldsymbol{x} \in \mathbb{R}^d$,

$$\frac{1}{N} \sum_{i=1}^{N} \|\nabla f_i^m(\boldsymbol{x})\|^2 \leq G^2, \tag{14}$$

and this in fact implies the $(G, 0)$-BGD assumption (13). To see why, from Jensen's inequality

$$\|\nabla F^m(\boldsymbol{x})\|^2 := \left\| \frac{1}{N} \sum_{i=1}^{N} \nabla f_i^m(\boldsymbol{x}) \right\|^2 \leq \frac{1}{N} \sum_{i=1}^{N} \|\nabla f_i^m(\boldsymbol{x})\|^2 \leq G^2.$$

Therefore, (14) is a stronger assumption for objective-wise heterogeneity than Assumption 3.

In Theorem 3 of Woodworth et al. (2020b), the authors use the following assumption on heterogeneity: for all $\boldsymbol{x} \in \mathbb{R}^d$,

$$\frac{1}{M} \sum_{m=1}^{M} \|\nabla F^m(\boldsymbol{x}) - \nabla F(\boldsymbol{x})\|^2 \leq \bar{\zeta}^2. \tag{15}$$

Noting that $\frac{1}{M} \sum_{m=1}^{M} \|\nabla F^m(\boldsymbol{x}) - \nabla F(\boldsymbol{x})\|^2 = \frac{1}{M} \sum_{m=1}^{M} \|\nabla F^m(\boldsymbol{x})\|^2 - \|\nabla F(\boldsymbol{x})\|^2$, we can see that (15) implies (13) and hence Assumption 2 (12), with $\tau = \bar{\zeta}$ and $\rho = 1$.

Indeed, there are also some results that make weaker heterogeneity assumptions than ours (12), by requiring bounded deviation only at the global optimum $\boldsymbol{x}^*$. Given the global optimum $\boldsymbol{x}^*$ of $F$, Koloskova et al. (2020) define

$$\bar{\zeta}_*^2 := \frac{1}{M} \sum_{m=1}^{M} \|\nabla F^m(\boldsymbol{x}^*)\|^2, \tag{16}$$

and use this constant in their bounds. Khaled et al. (2020) also define a similar quantity that can capture heterogeneity, but does not provide a result on strongly convex functions in the heterogeneous setting. While assuming bounded $\bar{\zeta}_*^2$ (16) is weaker than Assumption 3 in the sense that only a bound at $\boldsymbol{x}^*$ is required, we note that these assumptions cannot be applied easily in nonconvex settings; in fact, for nonconvex (but not necessarily PŁ) functions, Koloskova et al. (2020) also use (13) as their heterogeneity assumption.

**Intra-machine variance assumptions.** We next consider our notion of intra-machine deviation (Assumption 2), namely that there exists $\nu \geq 0$ such that for all $m \in [M]$ and $i \in [N]$,

$$\|\nabla f_i^m(\boldsymbol{x}) - \nabla F^m(\boldsymbol{x})\| \leq \nu, \text{ for all } \boldsymbol{x} \in \mathbb{R}^d. \tag{17}$$

In many existing results using independent and unbiased gradient estimates (Karimireddy et al., 2020; Yu et al., 2019; Li et al., 2020c; Qu et al., 2020; Woodworth et al., 2020a;b; 2021), the bounded variance assumption is adopted: For all $m \in [M]$ and $\boldsymbol{x} \in \mathbb{R}^d$,

$$\frac{1}{N} \sum_{i=1}^{N} \|\nabla f_i^m(\boldsymbol{x}) - \nabla F^m(\boldsymbol{x})\|^2 \leq \sigma^2. \tag{18}$$

We note that the bounded local gradients assumption (14) also implies (18), by

$$\frac{1}{N} \sum_{i=1}^{N} \|\nabla f_i^m(\boldsymbol{x}) - \nabla F^m(\boldsymbol{x})\|^2 \leq \frac{1}{N} \sum_{i=1}^{N} \|\nabla f_i^m(\boldsymbol{x}) - \nabla F^m(\boldsymbol{x})\|^2 + \|\nabla F^m(\boldsymbol{x})\|^2$$

$$= \frac{1}{N} \sum_{i=1}^{N} \|\nabla f_i^m(\boldsymbol{x})\|^2 \leq G^2. \tag{19}$$

Some other papers (Haddadpour et al., 2019; Haddadpour & Mahdavi, 2019; Spiridonoff et al., 2020) consider a generalized version of (18), namely that for all $m \in [M]$ and $\boldsymbol{x} \in \mathbb{R}^d$,

$$\frac{1}{N} \sum_{i=1}^{N} \|\nabla f_i^m(\boldsymbol{x}) - \nabla F^m(\boldsymbol{x})\|^2 \leq c \|\nabla F^m(\boldsymbol{x})\|^2 + \sigma^2, \tag{20}$$

for $c, \sigma \geq 0$. There are other papers (Koloskova et al., 2020; Khaled et al., 2020) that use intra-machine variance at the global minimum $\boldsymbol{x}^*$ in their bounds. Koloskova et al. (2020) define

$$\bar{\sigma}_*^2 := \frac{1}{MN} \sum_{m=1}^{M} \sum_{i=1}^{N} \|\nabla f_i^m(\boldsymbol{x}^*) - \nabla F^m(\boldsymbol{x}^*)\|^2, \tag{21}$$

for the global minimum $\boldsymbol{x}^*$, and use it in their strong convexity and convexity bounds. Khaled et al. (2020) define

$$\sigma_{\mathrm{opt}}^2 := \frac{1}{MN} \sum_{m=1}^{M} \sum_{i=1}^{N} \|\nabla f_i^m(\boldsymbol{x}^*)\|^2, \tag{22}$$

which is used in their bound on strongly convex functions in homogeneous cases. Note that for homogeneous cases, $\nabla F^m(\boldsymbol{x}^*) = \nabla F(\boldsymbol{x}^*) = \boldsymbol{0}$, so $\bar{\sigma}_*^2 = \sigma_{\mathrm{opt}}^2$.

We note that in contrast to our discussion on inter-machine deviation (Assumption 3), the intra-machine variance assumptions in the existing literature are weaker than our Assumption 2. However, we utilize our stronger assumption to prove our *high-probability* upper bounds, which is a departure from *in-expectation* bounds in the literature.

**Existing upper bounds on local SGD.** In the discussion after Theorem 2, we mentioned some recent upper bounds on (with-replacement) local SGD. We make more detailed comparisons here. For the reader's convenience, we restate our theorem on local RR below.

**Theorem 2** (Upper bound for local RR). *Suppose that local* RR *has parameters satisfying Assumption 1. For any $F \in \mathcal{F}_{\mathrm{obj}}(L, \mu, \nu, \tau, \rho)$, consider running the algorithm using step-size $\eta = \frac{\log(MNK^2)}{\mu NK}$ for epochs $K \geq 7\rho\kappa \log(MNK^2)$. Then, with probability at least $1 - \delta$,*

$$F(\boldsymbol{y}_{K, \frac{N}{B}}) - F^* \leq \frac{F(\boldsymbol{y}_0) - F^*}{MNK^2} + \tilde{\mathcal{O}}\left(\frac{L^2}{\mu^3}\left(\frac{\nu^2}{MNK^2} + \frac{\nu^2 B}{N^2 K^2} + \frac{\tau^2 B^2}{N^2 K^2}\right)\right). \tag{5}$$

We start with last-iterate bounds in homogeneous cases. Theorem 3 and Corollary 3 of Khaled et al. (2020) consider local SGD on $\mu$-strongly convex and $L$-smooth $F$. Khaled et al. (2020) allow variable synchronization intervals, where the maximum interval is upper-bounded by $B$. In this setting, Corollary 3 of Khaled et al. (2020) shows that local SGD after $T$ total local update steps yield

$$\mathbb{E}\left[\|\bar{\boldsymbol{x}}_T - \boldsymbol{x}_*\|^2\right] = \tilde{\mathcal{O}}\left(\frac{\|\bar{\boldsymbol{x}}_0 - \boldsymbol{x}_*\|^2}{T^2} + \frac{\sigma_{\mathrm{opt}}^2}{\mu^2 MT} + \frac{L\sigma_{\mathrm{opt}}^2(B-1)}{\mu^3 T^2}\right), \tag{23}$$

where $\bar{\boldsymbol{x}}_i$ is the average over all $M$ machines' $i$-th local iterates, and $\sigma_{\mathrm{opt}}$ is from (22). Noting that $T$ corresponds to $NK$ in local SGD and $\sigma_{\mathrm{opt}}$ corresponds to $\nu$ in Assumption 2, (23) is comparable to an upper bound[8]

$$\mathbb{E}\left[F(\bar{\boldsymbol{x}}_T) - F^*\right] = \tilde{\mathcal{O}}\left(\frac{L\nu^2}{\mu^2 MNK} + \frac{L^2 \nu^2 B}{\mu^3 N^2 K^2}\right). \tag{24}$$

Here, we do not compare $\frac{F(\boldsymbol{y}_0) - F^*}{MNK^2}$ and $\frac{L\|\bar{\boldsymbol{x}}_0 - \boldsymbol{x}_*\|^2}{T^2}$ because in Theorem 2, the term $\frac{F(\boldsymbol{y}_0) - F^*}{MNK^2}$ can be made arbitrarily "small" (e.g., $\frac{F(\boldsymbol{y}_0) - F^*}{(MNK)^l}$ for any $l \in \mathbb{N}$) by changing the log factor in $\eta$.

A similar homogeneous, strongly convex, and smooth setting is considered in Spiridonoff et al. (2020), under a intra-machine variance assumption defined in (20). Theorem 1 of Spiridonoff et al. (2020) proves general theorem statement for arbitrary synchronization intervals. If we specialize to constant interval $B$, Corollary 1 of Spiridonoff et al. (2020) gives

$$\mathbb{E}\left[F(\bar{\boldsymbol{x}}_T) - F^*\right] = \tilde{\mathcal{O}}\left(\frac{\beta^2(F(\bar{\boldsymbol{x}}_0) - F^*)}{T^2} + \frac{L\sigma^2}{\mu^2 MT} + \frac{L^2 \sigma^2(B-1)}{\mu^3 T^2}\right), \tag{25}$$

---

[8]Note that an additional factor $L$ is due to conversion from squared distance to function value.

where $\beta \geq 2\kappa^2$ is a constant defined to choose algorithm parameters such as the step-size. Noting again that $T$ corresponds to $NK$ and $\sigma$ in (20) is comparable to $\nu$ in Assumption 2, (25) also translates to (24).

The next last-iterate bound we compare against is Qu et al. (2020). Theorem 1 of Qu et al. (2020) uses bounded intra-machine variance assumption (18) and bounded gradient assumption (14). Specializing Theorem 1 of Qu et al. (2020) to full device participation and uniform weight ($p_1 = \cdots = p_N = \frac{1}{N}$) case, their bound reads

$$\mathbb{E}\left[F(\bar{\boldsymbol{x}}_T) - F^*\right] = \tilde{\mathcal{O}}\left(\frac{L\sigma^2}{\mu^2 M T} + \frac{L^2 G^2 B^2}{\mu^3 T^2}\right). \tag{26}$$

Recalling that $T$ corresponds to $NK$, $\sigma$ to $\nu$ in Assumption 2, and $G$ to the heterogeneity constant $\tau$ in Assumption 3 and also $\nu$ (due to (19)), (26) translates to

$$\mathbb{E}\left[F(\bar{\boldsymbol{x}}_T) - F^*\right] = \tilde{\mathcal{O}}\left(\frac{L\nu^2}{\mu^2 M N K} + \frac{L^2(\nu^2 + \tau^2)B^2}{\mu^3 N^2 K^2}\right). \tag{27}$$

Comparing the local SGD last-iterate bounds (24) and (27) against our local RR bound (5) in Theorem 2, we can see that the last iterate of local RR satisfies a smaller upper bound as soon as $K \geq \kappa$. Admittedly, this is not a fully rigorous comparison given the differences in assumptions and types of bounds; nevertheless, we believe that the comparison at least provides some degree of evidence for faster convergence of local RR than local SGD. It is also interesting to see that the "error from local updates" terms match in with- and without-replacement bounds.

Next, we review existing average-iterate bounds mentioned in the main text, which are better than the last-iterate bounds. Koloskova et al. (2020) present a unifying framework for analyzing distributed optimization algorithms over networks, which can specialize to local SGD. For $\mu$-strongly convex and $L$-smooth $F$, Theorem 2 of Koloskova et al. (2020) shows that

$$\mathbb{E}\left[F(\hat{\boldsymbol{x}}) - F^*\right] = \tilde{\mathcal{O}}\left(LB\|\boldsymbol{x}_0 - \boldsymbol{x}^*\|^2 \exp\left(-\frac{\mu T}{LB}\right) + \frac{\bar{\sigma}_*^2}{\mu M T} + \frac{L\bar{\sigma}_*^2 B}{\mu^2 T^2} + \frac{L\bar{\zeta}_*^2 B^2}{\mu^2 T^2}\right), \tag{28}$$

where $\hat{\boldsymbol{x}}$ is some weighted average of iterates and $\bar{\zeta}_*^2$ and $\bar{\sigma}_*^2$ are defined in (16) and (21), respectively. Noting that $T$ corresponds to $NK$, $\bar{\zeta}_*$ to $\tau$ in Assumption 3, and $\bar{\sigma}_*$ to $\nu$ in Assumption 2 (although our assumptions are stronger), we can see that the bound (28) for large enough $T$ can be translated into

$$\mathbb{E}\left[F(\hat{\boldsymbol{x}}) - F^*\right] = \tilde{\mathcal{O}}\left(\frac{\nu^2}{\mu M N K} + \frac{L\nu^2 B}{\mu^2 N^2 K^2} + \frac{L\tau^2 B^2}{\mu^2 N^2 K^2}\right). \tag{29}$$

Notice that (29) is smaller than the last-iterate bounds (24) and (27) by a factor of $\kappa$. Similarly, Theorem 3 of Woodworth et al. (2020b) proves that for $\mu$-strongly convex and $L$-smooth $F$,

$$\mathbb{E}\left[F(\tilde{\boldsymbol{x}}) - F^*\right] = \tilde{\mathcal{O}}\left(\frac{L^2\|\boldsymbol{x}_0 - \boldsymbol{x}^*\|^2}{LT + \mu T^2} + \frac{\bar{\sigma}_*^2}{\mu M T} + \frac{L\sigma^2 B}{\mu^2 T^2} + \frac{L\bar{\zeta}^2 B^2}{\mu^2 T^2}\right), \tag{30}$$

for some weighted average of iterates $\tilde{\boldsymbol{x}}$. Here, $\bar{\sigma}_*^2$, $\sigma^2$, and $\bar{\zeta}^2$ are as defined in (21), (18), and (15), respectively. Substituting $\nu$ to its comparable constants $\bar{\sigma}_*$ and $\sigma$, and $\tau$ to $\bar{\zeta}$, we can similarly check that (30) can be converted to (29).

**Comparison to Mishchenko et al. (2021).** In Mishchenko et al. (2021), the authors study a proximal algorithm referred to as Proximal Random Reshuffling (ProxRR), and obtain a distributed optimization algorithm called FedRR as a special case. If we set $R \equiv 0$ in FedRR, the algorithm then is equal to local RR with $B = N$, i.e., the one that synchronizes only after one entire epoch.

Assuming that all component functions $f_i^m(\boldsymbol{x})$ are $\mu$-strongly convex[9] and $L$-smooth, and objective-wise homogeneity $F^1 = \cdots = F^m = F$, the authors obtain Theorem 8 (Mishchenko et al., 2021), which states that

$$\mathbb{E}\left[\|\boldsymbol{y}_K - \boldsymbol{x}^*\|^2\right] \leq (1 - \eta\mu)^{NK}\|\boldsymbol{y}_0 - \boldsymbol{x}^*\|^2 + \frac{\eta^2 LMN\sigma_{\text{opt}}^2}{\mu M}, \tag{31}$$

---

[9]This is in fact quite strong compared to this paper, because we only assume $F$ to be PŁ, not $F^m$ nor $f_i^m$.

where $\boldsymbol{y}_0$ and $\boldsymbol{y}_K$ are the initialization and last iterate of the algorithm, and $\sigma_{\text{opt}}^2$ was defined above in (22). The term $MN\sigma_{\text{opt}}^2$ in (31) corresponds to the term "$N\sigma_*^2$" as per the notation in Mishchenko et al. (2021). If we apply our Assumption 3 to bound $\sigma_{\text{opt}}^2$, we get $\sigma_{\text{opt}}^2 \leq \nu^2$, which reduces the last term in (31) to $\frac{\eta^2 L\nu^2 N}{\mu}$. If we substitute $\eta = \frac{\log(MNK^2)}{\mu NK}$ to the bound (31), we get

$$\mathbb{E}\left[\|\boldsymbol{y}_K - \boldsymbol{x}^*\|^2\right] \leq \frac{\|\boldsymbol{y}_0 - \boldsymbol{x}^*\|^2}{MNK^2} + \tilde{\mathcal{O}}\left(\frac{L\nu^2}{\mu^3 NK^2}\right), \tag{32}$$

which translates to the same convergence rate on $F(\boldsymbol{y}_K) - F^*$ as single-machine RR. For the heterogeneous setting, applying Lemma 3 of Mishchenko et al. (2021) to Theorem 2, we can obtain

$$\mathbb{E}\left[\|\boldsymbol{y}_K - \boldsymbol{x}^*\|^2\right] \leq (1 - \eta\mu)^{NK}\|\boldsymbol{y}_0 - \boldsymbol{x}^*\|^2$$
$$+ \frac{2\eta^2 L}{\mu M}\sum_{m=1}^M\left(N^2\|\nabla F^m(\boldsymbol{x}^*)\|^2 + \frac{1}{4}\sum_{i=1}^N\|\nabla f_i^m(\boldsymbol{x}^*) - \nabla F^m(\boldsymbol{x}^*)\|^2\right)$$
$$= (1 - \eta\mu)^{NK}\|\boldsymbol{y}_0 - \boldsymbol{x}^*\|^2 + \frac{2\eta^2 LN^2}{\mu M}\sum_{m=1}^M\|\nabla F^m(\boldsymbol{x}^*)\|^2 + \frac{\eta^2 L\bar{\sigma}_*^2 N}{2\mu}, \tag{33}$$

where $\bar{\sigma}_*^2$ was defined in (21). Note that in Lemma 3 (Mishchenko et al., 2021), the function "$F_m$" in the authors' notation is equal to $NF^m$ in our notation. Recall that the $(G, B)$-BGD assumption (13) is "comparable" to Assumption 3 (12). If we apply (13) to the bound (33), we get $\frac{1}{M}\sum_{m=1}^M\|\nabla F^m(\boldsymbol{x}^*)\|^2 \leq G^2$. Similarly, if we apply Assumption 3, we get $\bar{\sigma}_*^2 \leq \nu^2$. Substituting these upper bounds and $\eta = \frac{\log(MNK^2)}{\mu NK}$ to (33) gives

$$\mathbb{E}\left[\|\boldsymbol{y}_K - \boldsymbol{x}^*\|^2\right] \leq \frac{\|\boldsymbol{y}_0 - \boldsymbol{x}^*\|^2}{MNK^2} + \tilde{\mathcal{O}}\left(\frac{LG^2}{\mu^3 K^2} + \frac{L\nu^2}{\mu^3 NK^2}\right), \tag{34}$$

and after translating this into a bound on function value, we get an upper bound $\tilde{\mathcal{O}}(\frac{L^2\nu^2}{\mu^3 NK^2} + \frac{L^2 G^2}{\mu^3 K^2})$ which in fact matches our upper bound (5) in Theorem 2 when we set $B = N$.

The two upper bounds obtained for homogeneous (32) and heterogeneous (34) settings indicate that, at least under assumptions on intra- and inter-machine deviation such as ours, the claimed advantage that "the convergence bound improves with the number of devices involved" (Mishchenko et al., 2021) is not achievable. As our lower bound shows, one needs to choose $B$ smaller than $N$ in order to get the most out of parallelism. That being said, since Mishchenko et al. (2021) is free of uniform intra- and inter-machine deviation assumptions, there may still exist certain scenarios where multiple machines can speed up performance even with $B = N$.

## B    MORE DETAILED ILLUSTRATION OF SYNCHRONIZED SHUFFLING

In this section, we provide a more detailed explanation on synchronized shuffling that we introduced in Section 5. For the illustration, let us consider the component-wise homogeneous case. Component-wise homogeneity means that all the machines have the same set of components: $f_i^1 = f_i^2 = \cdots = f_i^M =: f_i$ for $i \in [N]$. Hence, we have $F = F^m$ for all $m \in [M]$ and our goal is to minimize $F = \frac{1}{N}\sum_{i=1}^N f_i$.

In the proof of Theorem 1 (presented in Appendix D.2; see Appendix D.1 for a sketch), we add the component gradients over an epoch and then use the following key identity: for any permutation $\sigma$,

$$\sum_{i=1}^N \nabla f_{\sigma(i)} = \sum_{i=1}^N \nabla f_i = N\nabla F. \tag{35}$$

We use this identity (35) to represent the per-epoch progress as "one big GD step plus noise." For the rest of the proof we bound the "noise" term, and the key to bounding it is to upper bound the norm of summations of the following form, for $i \in [N/B - 1]$ (see (41) and (43)):

$$\frac{1}{M}\sum_{m=1}^M\sum_{j=1}^{iB}\nabla f_{\sigma_k^m(j)}(\boldsymbol{x}_{k,0}). \tag{36}$$

To bound the norm, we decompose it into two terms using the triangle inequality

$$\left\|\frac{1}{M}\sum_{m=1}^{M}\sum_{j=1}^{iB}\nabla f_{\sigma_k^m(j)}(\boldsymbol{x}_{k,0})\right\| \leq \left\|\frac{1}{M}\sum_{m=1}^{M}\sum_{j=1}^{iB}\nabla f_{\sigma_k^m(j)}(\boldsymbol{x}_{k,0}) - iB\nabla F(\boldsymbol{x}_{k,0})\right\| + iB\left\|\nabla F(\boldsymbol{x}_{k,0})\right\|,$$

and we use concentration bounds (44) on the first term of the RHS, which gives a high-probability upper bound on the RHS of the form $\tilde{O}\left(\nu\sqrt{\frac{iB}{M}}\right) + iB\left\|\nabla F\right\|$ (45).

The key to proving fast convergence is to make the upper bound above as small as possible. To make the bound (45) even smaller, we wish to be able to apply the identity (35) to the summation (36). However, under the standard way of choosing permutations $\sigma_k^m$ independently over machines, one cannot apply the identity because we do not sum over all $j = 1, \dots, N$. This limitation motivates our proposed technique *synchronized shuffling*, a manipulation on the choices of $\sigma_k^m$ that lets us prove even faster convergence.

Recall the definition of synchronized shuffling. At the beginning of the $k$-th epoch, the server samples $\sigma \sim \text{Unif}(\mathcal{S}_N)$ and $\pi \sim \text{Unif}(\mathcal{S}_M)$, and broadcasts them to the machines. Then, local machines choose their permutations $\sigma_k^m$ to be *shifted* versions of $\sigma$: $\sigma_k^m(i) := \sigma\left(\left(i + \frac{N}{M}\pi(m)\right) \bmod N\right)$.

Now set $N = 6$ and $M = 3$. Assume for simplicity that the permutation $\pi \in \mathcal{S}_3$ of machines satisfies $\pi(m) = m$ for $m \in [3]$.[10] Suppose the server samples $\sigma = (\sigma(1), \sigma(2), \sigma(3), \sigma(4), \sigma(5), \sigma(6))$ and broadcasts it. Under synchronized shuffling, the local machines choose

$$\sigma_k^1 = (\sigma(3), \sigma(4), \sigma(5), \sigma(6), \sigma(1), \sigma(2)),$$
$$\sigma_k^2 = (\sigma(5), \sigma(6), \sigma(1), \sigma(2), \sigma(3), \sigma(4)),$$
$$\sigma_k^3 = (\sigma(1), \sigma(2), \sigma(3), \sigma(4), \sigma(5), \sigma(6)).$$

One can see that each permutation is a shifted version of $\sigma$, with an offset that is a multiple of $\frac{N}{M} = 2$.

Now consider adding $\nabla f_{\sigma_k^m(i)}$ over $m = 1, 2, 3$ and $j = 1, 2$ (in fact, any two consecutive $j$'s will do). By synchronized shuffling, we get a summation over all $N = 6$ component functions, which by (35) gives us the full gradient:

$$\sum_{m=1}^{3}\sum_{j=1}^{2}\nabla f_{\sigma_k^m(j)} = \sum_{i=1}^{6}\nabla f_i = 6\nabla F.$$

The point here is that the permutation identity (35) can be applied to the summations (36) to further reduce their norm bounds. This is in contrast to sampling independent $\sigma_k^m$'s where one cannot apply (35). For this reason, synchronized shuffling significantly reduces the noise that comes from without-replacement sampling, thus resulting in faster convergence rates in (near-)homogeneous cases.

## C  EXPERIMENTAL RESULTS

In this section, we present some simple numerical experiments that support our theoretical analysis. We evaluate the performance of the algorithms considered in this paper on the "hard instance" constructed in our lower bounds (Theorems 3 and 4).

Our hard instance $F \in \mathcal{F}_{\text{cmp}}(L, \mu, \nu, 0)$ is a function in the component-wise homogeneous setting, where all the machines have the same set of local component functions: $f_i^1 = f_i^2 = \cdots = f_i^M =: f_i$. In the proofs of Theorems 3 and 4, we construct the global objective $F(x) = \frac{1}{N}\sum_{i=1}^{N}f_i(x)$ as the following:

$$f_i(x) := (L\mathbb{1}_{x\leq 0} + \mu\mathbb{1}_{x>0})\frac{x^2}{2} + z_i\nu x, \quad z_i = \begin{cases} +1 & \text{if } 1 \leq i \leq N/2, \\ -1 & \text{if } N/2 < i \leq N. \end{cases}$$

---

[10]In fact, permuting the machines by $\pi$ is not required in the component-wise homogeneous setting (i.e., when $\lambda = 0$ in Assumption 4).

With this set of component functions, the global objective $F(x) = (L1_{x \leq 0} + \mu 1_{x > 0})\frac{x^2}{2}$ is $\mu$-strongly convex and $L$-smooth with a unique global minimizer at $x = 0$.

We compare the performance of the algorithms on this problem instance, with $L = 100$, $\mu = 1$, $\nu = 1$, $N = 768$, and $M = 16$, while varying the choice of $B \in \{1, 4, 16, 64, 256\}$ and $K \in \{1, 3, 5, 7, 10, 30, 50, 70, 100, 300, 500, 700, 1000\}$. For each value of $B$ and $K$, we run the algorithms for $K$ epochs ($KN/B$ communication rounds for with-replacement algorithms) starting at $x_0 = 0$ and return the values of $F$ evaluated at the last iterates. Note that the algorithms are not deterministic, because the sampling/shuffling schemes are random. In order to account for randomness, for each combination of (algorithm, $B$, $K$) we execute 20 independent runs of the algorithm and plot the mean, first quartile, and third quartile of the final objective values.

In the subsequent subsections, we compare the following seven algorithms with constant step-sizes.

- Minibatch RR, $\eta = \frac{B \log(MNK^2)}{\mu NK}$;

- Local RR, $\eta = \frac{\log(MNK^2)}{\mu NK}$;

- Minibatch RR with SYNCSHUF, $\eta = \frac{B \log(MNK^2)}{\mu NK}$;

- Local RR with SYNCSHUF, $\eta = \frac{\log(MNK^2)}{\mu NK}$;

- Single-machine RR with minibatch size $B$, $\eta = \frac{\log(NK^2)}{\mu NK}$;

- With-replacement minibatch SGD, $\eta = \frac{B \log(MNK^2)}{\mu NK}$;

- With-replacement local SGD, $\eta = \frac{\log(MNK^2)}{\mu NK}$.

## C.1   SYNCSHUF IMPROVES CONVERGENCE OF MINIBATCH/LOCAL RR

In Figure 1, we compare minibatch RR and local RR, with and without SYNCSHUF. Each plot in Figure 1 shows how the methods' performance changes with $K$, for a fixed value $B$. Each point on the curve is the mean of the final objective function values over 20 independent runs of the corresponding algorithm with the specific $B$ and $K$, and its error bar indicates the first and third quartiles.

Recall from our Theorems 1, 2, 6, and 7 that the four methods satisfy the following convergence bounds, in homogeneous settings (i.e., $\tau = \lambda = 0$):

- Minibatch RR: $\tilde{\mathcal{O}}\left(\frac{L^2}{\mu^3}\frac{\nu^2}{MNK^2}\right)$,

- Local RR: $\tilde{\mathcal{O}}\left(\frac{L^2}{\mu^3}\left(\frac{\nu^2}{MNK^2} + \frac{\nu^2 B}{N^2 K^2}\right)\right)$,

- Minibatch RR with SYNCSHUF: $\tilde{\mathcal{O}}\left(\frac{L^2}{\mu^3}\frac{\nu^2}{M^2 NK^2}\right)$,

- Local RR with SYNCSHUF: $\tilde{\mathcal{O}}\left(\frac{L^2}{\mu^3}\left(\frac{\nu^2}{M^2 NK^2} + \frac{\nu^2 B}{N^2 K^2}\right)\right)$.

In fact, if $B = 1$, local RR is identical to minibatch RR. Figure 1(a) confirms that this is indeed true, and also that the versions with SYNCSHUF outperforms the ones without SYNCSHUF. This corroborates the additional $M$ factor speedup in our bounds. In Figure 1(b) and 1(c), we can see that as $B$ increases, the performance of local RR with SYNCSHUF degrades and becomes closer to local RR without SYNCSHUF. This shows that the $\tilde{\mathcal{O}}\left(\frac{L^2}{\mu^3}\frac{\nu^2 B}{N^2 K^2}\right)$ term starts to dominate. Also, as we increase $B$ further, in Figure 1(c) and 1(d) we see that local RR (without SYNCSHUF) also starts to degrade and its gap between minibatch RR becomes larger. Again, this means that the $\tilde{\mathcal{O}}\left(\frac{L^2}{\mu^3}\frac{\nu^2 B}{N^2 K^2}\right)$ term becomes the dominant factor in the local RR bound. The performance of minibatch RR, with and without SYNCSHUF, stays relatively independent of $B$. One thing to note is that for large values of $B$, the small-epoch behavior of minibatch RR looks rather unstable. The choice of step-size $\eta = \frac{B \log(MNK^2)}{\mu NK}$ seems to cause overshooting when $B$ is large and $K$ is small. We

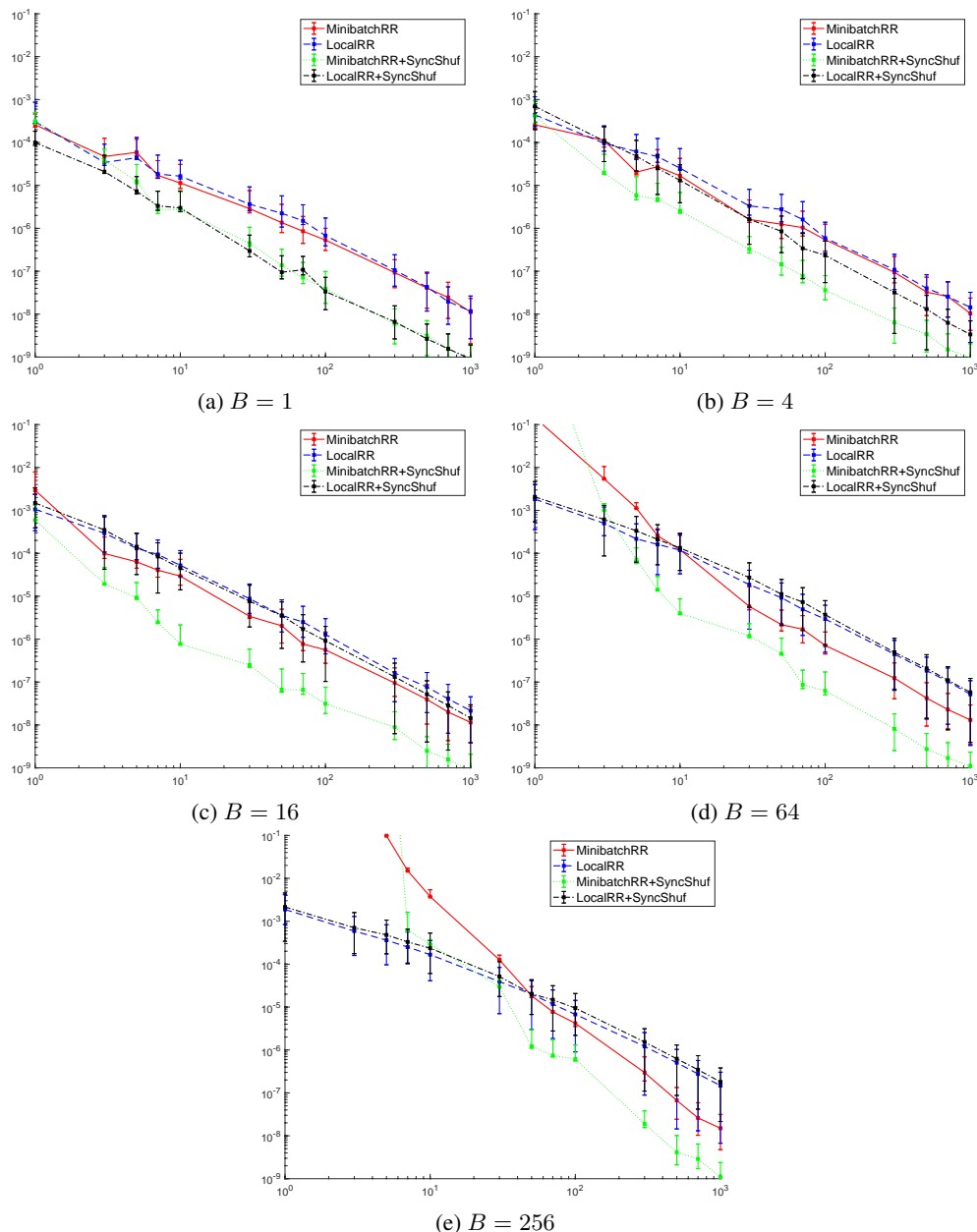

Figure 1: Comparison between minibatch RR and local RR, with and without SYNCSHUF. Best viewed in color. The algorithm versions with SYNCSHUF converge faster. Also note the performance degradation as $B$ increases, as expected by our theory.

note that this does not contradict our convergence analysis because our theorems only characterize the large-epoch behavior ($K$ above certain thresholds) of the algorithms. Perhaps in the small-epoch regime, our choice of $\eta$ is not necessarily optimal and a smaller $\eta$ is needed to prevent overshooting.

## C.2  LOCAL RR BECOMES CLOSER TO SINGLE-MACHINE RR AS $B \to N$

Our next set of plots presented in Figure 2 provides a comparison of minibatch RR, local RR, and single-machine RR (i.e., minibatch RR with $M = 1$). In Theorems 2 and 4, we showed that when $B = \Theta(N)$, then the convergence of local RR becomes just as fast as the single-machine RR.

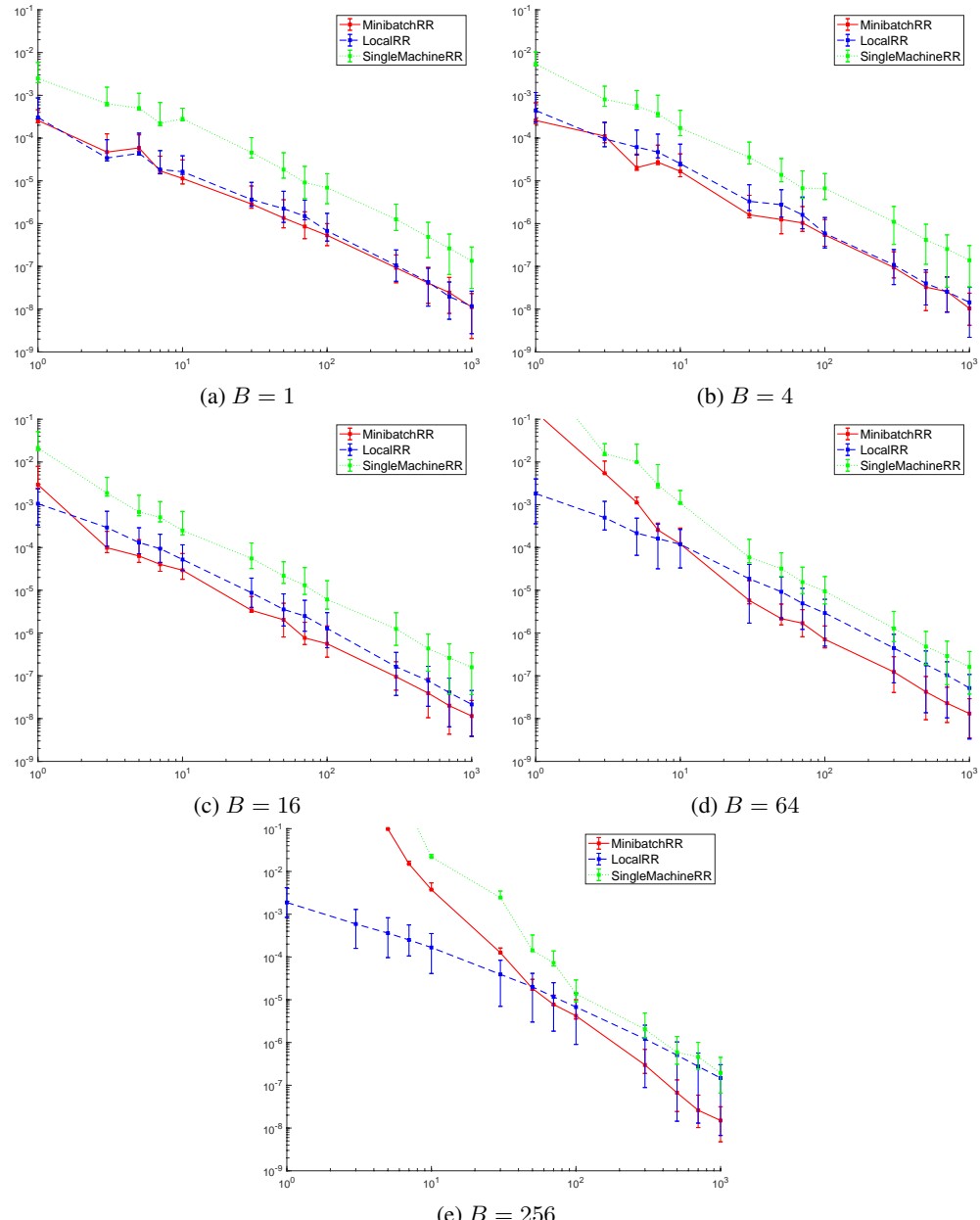

Figure 2: Comparison of minibatch RR, local RR, and single-machine RR. Best viewed in color. The large-epoch performance of local RR becomes similar to that of single-shuffle RR as $B$ becomes closer to $N$.

Indeed, we can observe from Figure 2 that this is really the case. As $B$ increases, the curve of local RR moves closer and closer to that of single-machine RR, especially in the large-epoch regime.

## C.3 WITH- VS. WITHOUT-REPLACEMENT SAMPLING

Lastly, in Figure 3 we compare the with-replacement and without-replacement versions of minibatch/local SGD. In all plots, we can see that the without-replacement versions outperform with-replacement ones, at least for our problem instance. It is also intriguing to note that the two versions perform very similarly in the small-epoch regime (for $K$ up to $\sim 10$), but without-replacement starts

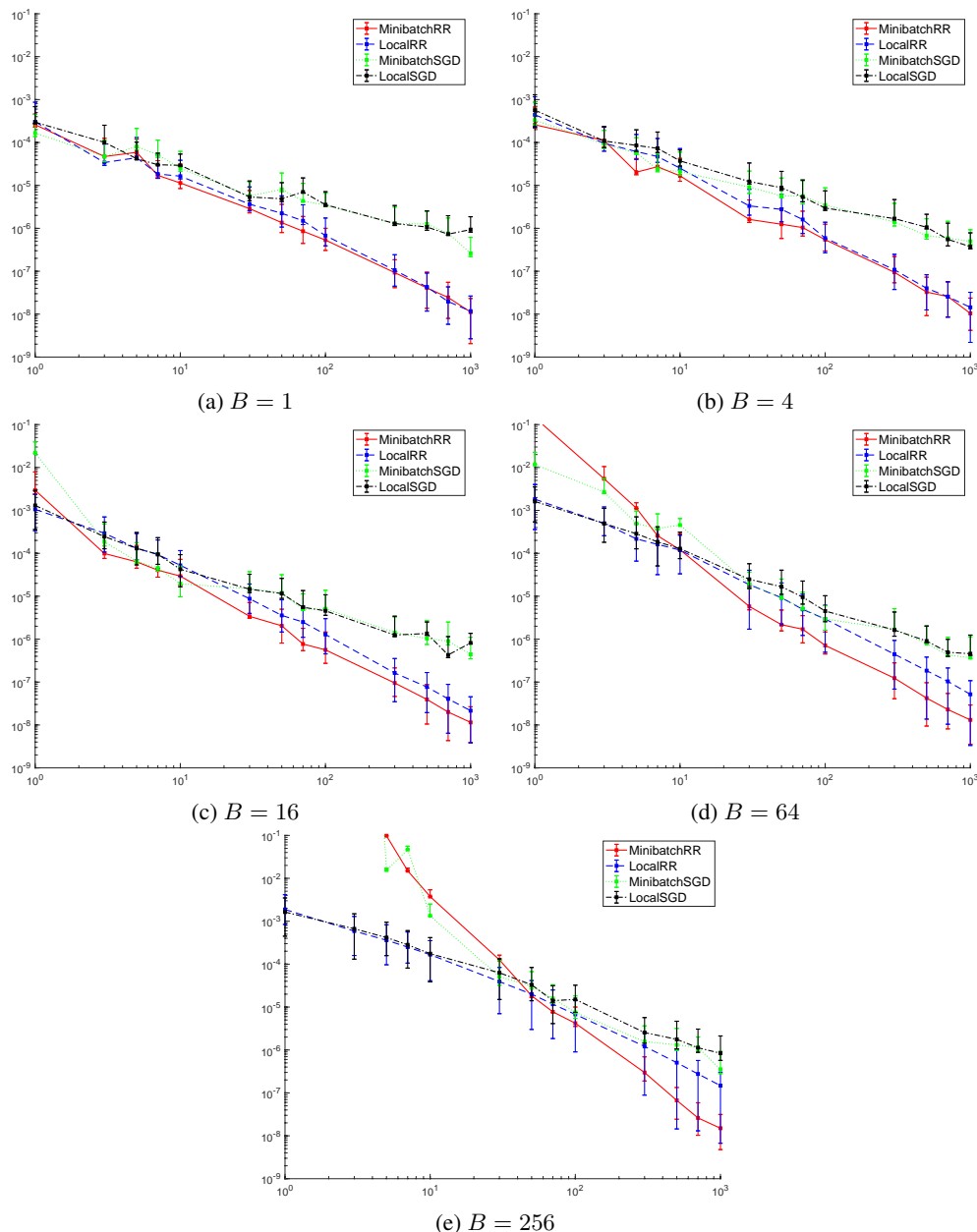

(a) $B = 1$

(b) $B = 4$

(c) $B = 16$

(d) $B = 64$

(e) $B = 256$

Figure 3: Comparison of with-replacement (SGD) and without-replacement (RR) versions. Best viewed in color. Without-replacement versions converge faster than with-replacement versions, at least in our problem instance. Also note that the two versions perform similarly in the small-epoch regime, which supports our theoretical findings.

to outperform for larger $K$'s. This observation supports our theoretical prediction from Theorem 3 that in the small-epoch regime, minibatch RR can at best perform as fast as minibatch SGD.

## D   PROOFS OF UPPER BOUNDS

In this section, we provide proofs of our upper bounds stated in Sections 3 and 5. We start by describing a high-level proof outline that we use for all the proofs presented in this section (Appendix D.1). In the subsequent subsections, we prove Theorems 1, 2, 6, and 7, in the order they appeared in the main text. The next subsection (Appendix D.6) states and proves a key lemma that gives concen-

tration bounds for the mean of multiple without-replacement sums of vectors. This general-purpose lemma can be of independent interest and it can prove useful in various other settings. Lastly, in Appendix D.7 we discuss how we can modify the theorem statements to remove the requirement in Assumptions 2, 3, and 4 that they must hold for the entire $\mathbb{R}^d$.

**Notation.** Throughout this section, we use the product notation $\prod$ in a slightly unconventional manner. For indices $i \le j$ and square matrices $\boldsymbol{A}_i, \boldsymbol{A}_{i+1}, \dots, \boldsymbol{A}_{j-1}, \boldsymbol{A}_j$, we use $\prod_{l=j}^{i} \boldsymbol{A}_l$ to denote the matrix product $\boldsymbol{A}_j \boldsymbol{A}_{j-1} \cdots \boldsymbol{A}_{i+1} \boldsymbol{A}_i$. If $i > j$, then $\prod_{l=j}^{i} \boldsymbol{A}_l = \boldsymbol{I}$.

## D.1 PROOF OUTLINE

The proofs of upper bounds follow a common structure, consisting of the following three steps:

1. writing one epoch as one step of GD plus noise;
2. getting a high-probability upper bound on the noise term using concentration inequalities;
3. obtaining the convergence rate using the bounds on the noise term.

We first unroll the update equations over an epoch, and write an epoch of the algorithms as one step of gradient descent plus noise:[11]

$$\boldsymbol{x}_{k+1,0} = \boldsymbol{x}_{k,0} - \eta N \nabla F(\boldsymbol{x}_{k,0}) + \eta^2 \boldsymbol{r}_k.$$

Substituting the above to the definition of $L$-smoothness of $F$ and arranging terms, we obtain

$$
\begin{aligned}
& F(\boldsymbol{x}_{k+1,0}) - F(\boldsymbol{x}_{k,0}) \\
& \le \langle \nabla F(\boldsymbol{x}_{k,0}), \boldsymbol{x}_{k+1,0} - \boldsymbol{x}_{k,0} \rangle + \frac{L}{2} \|\boldsymbol{x}_{k+1,0} - \boldsymbol{x}_{k,0}\|^2 \\
& \le -\eta N \|\nabla F(\boldsymbol{x}_{k,0})\|^2 + \eta^2 \|\nabla F(\boldsymbol{x}_{k,0})\| \|\boldsymbol{r}_k\| + \frac{\eta^2 L}{2} \|N \nabla F(\boldsymbol{x}_{k,0}) + \eta \boldsymbol{r}_k\|^2 \\
& \le (-\eta N + \eta^2 L N^2) \|\nabla F(\boldsymbol{x}_{k,0})\|^2 + \eta^2 \|\nabla F(\boldsymbol{x}_{k,0})\| \|\boldsymbol{r}_k\| + \eta^4 L \|\boldsymbol{r}_k\|^2.
\end{aligned}
\tag{37}
$$

The next step is to get high-probability upper bounds on the term $\|\boldsymbol{r}_k\|$. This is done by applying our concentration inequality lemma (Lemma 8) to partial without-replacement sums of component gradients. As a result, we will get upper bounds on $\|\boldsymbol{r}_k\|$ and $\|\boldsymbol{r}_k\|^2$, for $k = 1, \dots, K$, which hold with probability at least $1 - \delta$.

In the last part, we substitute the high-probability bounds to (37) and invoke the definition of PŁ functions to get a per-epoch progress bound. We then unroll the per-epoch inequality for all epochs $k = 1, \dots, K$. Arranging the terms in the resulting inequality gives our desired convergence bound that holds with probability at least $1 - \delta$.

## D.2 PROOF OF UPPER BOUND FOR MINIBATCH RR (THEOREM 1)

**One epoch as one step of GD plus noise.** To simplify the notation throughout the proof, we will prove the same convergence rate for a rescaled update rule and step-size:

$$\boldsymbol{x}_{k,i} := \boldsymbol{x}_{k,i-1} - \frac{\eta}{M} \sum_{m=1}^{M} \sum_{j=(i-1)B+1}^{iB} \nabla f_{\sigma_k^m(j)}^m (\boldsymbol{x}_{k,i-1}) \tag{38}$$

for $i \in [N/B]$ and $k \in [K]$, and $\eta = \frac{\log(MNK^2)}{\mu NK}$. Note that the gradient term is scaled up by $B$ and the step-size is scaled down by $B$. We will prove the convergence rate for this equivalent algorithm.

We start the proof by unrolling the update equations over an epoch and expressing the progress as

$$\boldsymbol{x}_{k+1,0} = \boldsymbol{x}_{k,0} - \eta N F(\boldsymbol{x}_{k,0}) + \eta^2 \boldsymbol{r}_k,$$

i.e., one step of full gradient descent plus some noise.

---

[11]In case of local RR, we replace $\boldsymbol{x}_{k,0}$ with $\boldsymbol{y}_{k,0}$.

To this end, we decompose the gradient $\nabla f^m_{\sigma^m_k(j)}(\boldsymbol{x}_{k,i-1})$ into the signal $\nabla f^m_{\sigma^m_k(j)}(\boldsymbol{x}_{k,0})$ and noise:

$$\nabla f^m_{\sigma^m_k(j)}(\boldsymbol{x}_{k,i-1}) = \nabla f^m_{\sigma^m_k(j)}(\boldsymbol{x}_{k,0}) + \nabla f^m_{\sigma^m_k(j)}(\boldsymbol{x}_{k,i-1}) - \nabla f^m_{\sigma^m_k(j)}(\boldsymbol{x}_{k,0})$$

$$= \nabla f^m_{\sigma^m_k(j)}(\boldsymbol{x}_{k,0}) + \left[\int_0^1 \nabla^2 f^m_{\sigma^m_k(j)}(\boldsymbol{x}_{k,0} + t(\boldsymbol{x}_{k,i-1} - \boldsymbol{x}_{k,0}))dt\right](\boldsymbol{x}_{k,i-1} - \boldsymbol{x}_{k,0}),$$

where $\nabla^2 f(\boldsymbol{x})$ denotes the Hessian of $f$ at $\boldsymbol{x}$, whenever it exists. We remark that the integral exists, due to the following reason. Since we assumed that each $f^m_{\sigma^m_k(j)}$ is differentiable and smooth, its gradient $\nabla f^m_{\sigma^m_k(j)}$ is Lipschitz continuous, and hence absolutely continuous. This means that $\nabla f^m_{\sigma^m_k(j)}$ is differentiable almost everywhere (i.e., $\nabla^2 f^m_{\sigma^m_k(j)}(\boldsymbol{x})$ exists a.e.), and the fundamental theorem of calculus for Lebesgue integral holds; hence the integral exists.

To simplify notation, we define the following for all $i \in [N/B]$:

$$\boldsymbol{g}_i := \frac{1}{M} \sum_{m=1}^M \sum_{j=(i-1)B+1}^{iB} \nabla f^m_{\sigma^m_k(j)}(\boldsymbol{x}_{k,0}),$$

$$\boldsymbol{H}_i := \frac{1}{M} \sum_{m=1}^M \sum_{j=(i-1)B+1}^{iB} \int_0^1 \nabla^2 f^m_{\sigma^m_k(j)}(\boldsymbol{x}_{k,0} + t(\boldsymbol{x}_{k,i-1} - \boldsymbol{x}_{k,0}))dt,$$

so that we can write (38) as

$$\boldsymbol{x}_{k,i} = \boldsymbol{x}_{k,i-1} - \eta \boldsymbol{g}_i - \eta \boldsymbol{H}_i(\boldsymbol{x}_{k,i-1} - \boldsymbol{x}_{k,0}). \tag{39}$$

From $L$-smoothness of $f^m_i$'s, it is straightforward to check that $\|\boldsymbol{H}_i\| \leq LB$. Unrolling (39) for $i = 1, \dots, N/B$, it turns out that we can write

$$\boldsymbol{x}_{k+1,0} = \boldsymbol{x}_{k,0} - \eta \sum_{i=1}^{N/B} \left(\prod_{j=N/B}^{i+1}(\boldsymbol{I} - \eta \boldsymbol{H}_j)\right)\boldsymbol{g}_i.$$

Due to summation by parts, the following identity holds:

$$\sum_{i=1}^{N/B} a_i b_i = a_{N/B} \sum_{j=1}^{N/B} b_j - \sum_{i=1}^{N/B-1}(a_{i+1} - a_i)\sum_{j=1}^i b_j.$$

We apply this to the last term, by substituting $a_i = \prod_{j=N/B}^{i+1}(\boldsymbol{I} - \eta \boldsymbol{H}_j)$ and $b_i = \boldsymbol{g}_i$:

$$\eta \sum_{i=1}^{N/B}\left(\prod_{j=N/B}^{i+1}(\boldsymbol{I} - \eta \boldsymbol{H}_j)\right)\boldsymbol{g}_i$$

$$= \eta \sum_{j=1}^{N/B}\boldsymbol{g}_j - \eta \sum_{i=1}^{N/B-1}\left(\prod_{t=N/B}^{i+2}(\boldsymbol{I} - \eta \boldsymbol{H}_t) - \prod_{t=N/B}^{i+1}(\boldsymbol{I} - \eta \boldsymbol{H}_t)\right)\sum_{j=1}^i \boldsymbol{g}_j$$

$$= \eta N \nabla F(\boldsymbol{x}_{k,0}) - \eta^2 \underbrace{\sum_{i=1}^{N/B-1}\left(\prod_{t=N/B}^{i+2}(\boldsymbol{I} - \eta \boldsymbol{H}_t)\right)\boldsymbol{H}_{i+1}\sum_{j=1}^i \boldsymbol{g}_j}_{=: \boldsymbol{r}_k}.$$

With the "noise" $\boldsymbol{r}_k$ defined as above, we can write $\boldsymbol{x}_{k+1,0} = \boldsymbol{x}_{k,0} - \eta N \nabla F(\boldsymbol{x}_{k,0}) + \eta^2 \boldsymbol{r}_k$, as desired. Next, it follows from $L$-smoothness of $F$ that

$$F(\boldsymbol{x}_{k+1,0}) - F(\boldsymbol{x}_{k,0})$$

$$\leq \langle \nabla F(\boldsymbol{x}_{k,0}), \boldsymbol{x}_{k+1,0} - \boldsymbol{x}_{k,0}\rangle + \frac{L}{2}\|\boldsymbol{x}_{k+1,0} - \boldsymbol{x}_{k,0}\|^2$$

$$\leq -\eta N \|\nabla F(\boldsymbol{x}_{k,0})\|^2 + \eta^2 \|\nabla F(\boldsymbol{x}_{k,0})\|\|\boldsymbol{r}_k\| + \frac{\eta^2 L}{2}\|N \nabla F(\boldsymbol{x}_{k,0}) + \eta \boldsymbol{r}_k\|^2$$

$$\leq (-\eta N + \eta^2 L N^2)\|\nabla F(\boldsymbol{x}_{k,0})\|^2 + \eta^2 \|\nabla F(\boldsymbol{x}_{k,0})\|\|\boldsymbol{r}_k\| + \eta^4 L \|\boldsymbol{r}_k\|^2, \tag{40}$$

where the last inequality used $\|\boldsymbol{a} + \boldsymbol{b}\|^2 \leq 2\|\boldsymbol{a}\|^2 + 2\|\boldsymbol{b}\|^2$.

**Bounding noise term using concentration.** It is left to bound $\|\boldsymbol{r}_k\|$. We have

$$
\begin{aligned}
\|\boldsymbol{r}_k\| &= \left\| \sum_{i=1}^{N/B-1} \left( \prod_{t=N/B}^{i+2} (\boldsymbol{I} - \eta \boldsymbol{H}_t) \right) \boldsymbol{H}_{i+1} \sum_{j=1}^{i} \boldsymbol{g}_j \right\| \\
&\leq \sum_{i=1}^{N/B-1} \left\| \left( \prod_{t=N/B}^{i+2} (\boldsymbol{I} - \eta \boldsymbol{H}_t) \right) \boldsymbol{H}_{i+1} \sum_{j=1}^{i} \boldsymbol{g}_j \right\| \\
&\leq LB(1 + \eta LB)^{N/B} \sum_{i=1}^{N/B-1} \left\| \sum_{j=1}^{i} \boldsymbol{g}_j \right\|,
\end{aligned}
\tag{41}
$$

where the last step used $\|\boldsymbol{H}_i\| \leq LB$ for $i \in [N/B]$. Recall from the theorem statement that $K \geq 6\kappa \log(MNK^2)$ and $\eta = \frac{\log(MNK^2)}{\mu NK}$. This means that

$$
(1 + \eta LB)^{N/B} = \left( 1 + \frac{\kappa B \log(MNK^2)}{NK} \right)^{N/B} \leq \left( 1 + \frac{B}{6N} \right)^{N/B} \leq e^{1/6}.
\tag{42}
$$

Now, we use Lemma 8 to bound the norm of

$$
\sum_{j=1}^{i} \boldsymbol{g}_j = \frac{1}{M} \sum_{m=1}^{M} \sum_{j=1}^{iB} \nabla f_{\sigma_k^m(j)}^m (\boldsymbol{x}_{k,0}).
\tag{43}
$$

Note that for any epoch $k$, the permutations $\sigma_k^1, \ldots, \sigma_k^M$ are independent of the first iterate $\boldsymbol{x}_{k,0}$ of the epoch, and hence independent of all $\nabla f_i^m(\boldsymbol{x}_{k,0})$. Therefore, we can apply Lemma 8 to the partial sum (43), with $\boldsymbol{v}_i^m \leftarrow \nabla f_i^m(\boldsymbol{x}_{k,0})$, $n \leftarrow iB$, and $\delta \leftarrow \frac{B\delta}{NK}$. By Lemma 8, with probability at least $1 - \frac{B\delta}{NK}$, we have

$$
\left\| \frac{1}{iBM} \sum_{m=1}^{M} \sum_{j=1}^{iB} \nabla f_{\sigma_k^m(j)}^m (\boldsymbol{x}_{k,0}) - \nabla F(\boldsymbol{x}_{k,0}) \right\| \leq \nu \sqrt{\frac{8 \log \frac{2NK}{B\delta}}{iBM}}.
\tag{44}
$$

Using this concentration bound, with probability at least $1 - \frac{B\delta}{NK}$ we have

$$
\left\| \sum_{j=1}^{i} \boldsymbol{g}_j \right\| \leq iB\nu \sqrt{\frac{8 \log \frac{2NK}{B\delta}}{iBM}} + iB \|\nabla F(\boldsymbol{x}_{k,0})\| = \nu \sqrt{\frac{8iB \log \frac{2NK}{B\delta}}{M}} + iB \|\nabla F(\boldsymbol{x}_{k,0})\|.
\tag{45}
$$

We can now substitute (42) and (45) to (41) to get

$$
\begin{aligned}
\|\boldsymbol{r}_k\| &\leq e^{1/6} LB \sum_{i=1}^{N/B-1} \left( \nu \sqrt{\frac{8iB \log \frac{2NK}{B\delta}}{M}} + iB \|\nabla F(\boldsymbol{x}_{k,0})\| \right) \\
&\leq \frac{e^{1/6} \sqrt{8} L\nu B^{3/2}}{M^{1/2}} \int_1^{N/B} \sqrt{t} \, dt \sqrt{\log \frac{2NK}{B\delta}} + e^{1/6} LB^2 \|\nabla F(\boldsymbol{x}_{k,0})\| \sum_{i=1}^{N/B-1} i \\
&\leq \frac{5L\nu (N^{3/2} - B^{3/2})}{2M^{1/2}} \sqrt{\log \frac{2NK}{B\delta}} + LN(N - B) \|\nabla F(\boldsymbol{x}_{k,0})\|,
\end{aligned}
\tag{46}
$$

which holds with probability at least $1 - \frac{\delta}{K}$, due to the union bound over $i = 1, \ldots, N/B - 1$. The bound (46) holds for all $k \in [K]$ with probability $1 - \delta$ if we apply the union bound over $k = 1, \ldots, K$. Next, by $(a + b)^2 \leq 2a^2 + 2b^2$, we have

$$
\|\boldsymbol{r}_k\|^2 \leq \frac{25L^2\nu^2(N^{3/2} - B^{3/2})^2}{2M} \log \frac{2NK}{B\delta} + 2L^2N^2(N - B)^2 \|\nabla F(\boldsymbol{x}_{k,0})\|^2,
\tag{47}
$$

which also holds for all $k \in [K]$ with probability at least $1 - \delta$.

**Getting a high-probability convergence rate.** Given our high-probability bounds (46) and (47), we can substitute them to (40) and get

$$
\begin{aligned}
&F(\boldsymbol{x}_{k+1,0}) - F(\boldsymbol{x}_{k,0}) \\
&\leq \left(-\eta N + \eta^2 L N^2\right) \|\nabla F(\boldsymbol{x}_{k,0})\|^2 + \eta^2 \|\nabla F(\boldsymbol{x}_{k,0})\| \|\boldsymbol{r}_k\| + \eta^4 L \|\boldsymbol{r}_k\|^2 \\
&\leq \left(-\eta N + \eta^2 L N^2 + \eta^2 L N (N-B) + 2\eta^4 L^3 N^2 (N-B)^2\right) \|\nabla F(\boldsymbol{x}_{k,0})\|^2 \\
&\quad + \frac{5\eta^2 L \nu (N^{3/2} - B^{3/2})}{2M^{1/2}} \sqrt{\log \frac{2NK}{B\delta}} \|\nabla F(\boldsymbol{x}_{k,0})\| \\
&\quad + \frac{25\eta^4 L^3 \nu^2 (N^{3/2} - B^{3/2})^2}{2M} \log \frac{2NK}{B\delta}.
\end{aligned}
\tag{48}
$$

The second term in the RHS of (48) can be bounded using $ab \leq \frac{a^2}{2} + \frac{b^2}{2}$:

$$
\begin{aligned}
&\frac{5\eta^2 L \nu (N^{3/2} - B^{3/2})}{2M^{1/2}} \sqrt{\log \frac{2NK}{B\delta}} \|\nabla F(\boldsymbol{x}_{k,0})\| \\
&= \left(\frac{\eta^{1/2} N^{1/2}}{2} \|\nabla F(\boldsymbol{x}_{k,0})\|\right) \left(\frac{5\eta^{3/2} L \nu (N^{3/2} - B^{3/2})}{M^{1/2} N^{1/2}} \sqrt{\log \frac{2NK}{B\delta}}\right) \\
&\leq \frac{\eta N}{8} \|\nabla F(\boldsymbol{x}_{k,0})\|^2 + \frac{25\eta^3 L^2 \nu^2 (N^{3/2} - B^{3/2})^2}{2MN} \log \frac{2NK}{B\delta}.
\end{aligned}
$$

Putting this inequality to (48) and noting $N - B \leq N$ gives

$$
\begin{aligned}
F(\boldsymbol{x}_{k+1,0}) - F(\boldsymbol{x}_{k,0}) &\leq \left(-\frac{7}{8}\eta N + 2\eta^2 L N^2 + 2\eta^4 L^3 N^4\right) \|\nabla F(\boldsymbol{x}_{k,0})\|^2 \\
&\quad + \frac{25\eta^3 L^2 (1 + \eta L N) \nu^2 (N^{3/2} - B^{3/2})^2}{2MN} \log \frac{2NK}{B\delta}.
\end{aligned}
\tag{49}
$$

Recall from $K \geq 6\kappa \log(MNK^2)$ and $\eta = \frac{\log(MNK^2)}{\mu NK}$ that $\eta LN \leq \frac{1}{6}$. Since the inequality $-\frac{7}{8}z + 2z^2 + 2z^4 \leq -\frac{1}{2}z$ holds on $z \in [0, \frac{1}{6}]$, we have

$$
-\frac{7}{8}\eta N + 2\eta^2 L N^2 + 2\eta^4 L^3 N^4 \leq -\frac{1}{2}\eta N.
$$

Applying this bound to (49) results in

$$
F(\boldsymbol{x}_{k+1,0}) - F(\boldsymbol{x}_{k,0}) \leq -\frac{\eta N}{2} \|\nabla F(\boldsymbol{x}_{k,0})\|^2 + \frac{15\eta^3 L^2 \nu^2 (N^{3/2} - B^{3/2})^2}{MN} \log \frac{2NK}{B\delta}.
$$

We now recall that $F$ is $\mu$-PŁ, so $\|\nabla F(\boldsymbol{x}_{k,0})\|^2 \geq 2\mu(F(\boldsymbol{x}_{k,0}) - F^*)$:

$$
F(\boldsymbol{x}_{k+1,0}) - F^* \leq (1 - \eta\mu N)(F(\boldsymbol{x}_{k,0}) - F^*) + \frac{15\eta^3 L^2 \nu^2 (N^{3/2} - B^{3/2})^2}{MN} \log \frac{2NK}{B\delta}.
\tag{50}
$$

Recall that (50) holds for all $k \in [K]$, with probability $1 - \delta$. Therefore, by unrolling the inequality,

$$
\begin{aligned}
F(\boldsymbol{x}_{K,\frac{N}{B}}) - F^* &\leq (1 - \eta\mu N)^K (F(\boldsymbol{x}_0) - F^*) \\
&\quad + \frac{15\eta^3 L^2 \nu^2 (N^{3/2} - B^{3/2})^2}{MN} \log \frac{2NK}{B\delta} \sum_{k=0}^{K-1} (1 - \eta\mu N)^k \\
&\leq (1 - \eta\mu N)^K (F(\boldsymbol{x}_0) - F^*) + \frac{15\eta^2 L^2 \nu^2 (N^{3/2} - B^{3/2})^2}{\mu MN^2} \log \frac{2NK}{B\delta}.
\end{aligned}
\tag{51}
$$

Lastly, substituting $\eta = \frac{\log(MNK^2)}{\mu NK}$ gives

$$
\begin{aligned}
F(\boldsymbol{x}_{K,\frac{N}{B}}) - F^* &\leq \frac{F(\boldsymbol{x}_0) - F^*}{MNK^2} + \frac{15 L^2 \nu^2 (N^{3/2} - B^{3/2})^2 \log \frac{2NK}{B\delta} \log^2(MNK^2)}{\mu^3 MN^4 K^2} \\
&= \frac{F(\boldsymbol{x}_0) - F^*}{MNK^2} + \tilde{\mathcal{O}}\left(\frac{L^2 \nu^2}{\mu^3} \frac{1}{MNK^2}\right).
\end{aligned}
\tag{52}
$$

**Getting an in-expectation bound from the high-probability bound.** We conclude this subsection by briefly describing how we can obtain an in-expectation bound from the high-probability bound we just proved. Recall that the bound we proved above holds under the event $E$ that all the concentration bounds used throughout the proof hold. The key to proving an in-expectation bound is to obtain an upper bound under its complement $E^c$, i.e., conditioned on the event that at least one of our concentration bounds does not hold. We do so by repeating the same proof *without* ever using the Hoeffding-Serfling bounds (Lemma 8). Of course, this leads to a much looser bound, but we can choose $\delta$ to be small enough so that the desired bound $\tilde{\mathcal{O}}\left(\frac{L^2\nu^2}{\mu^3}\frac{1}{MNK^2}\right)$ holds in expectation.

For the version without concentration bounds, the proof proceeds in the same way until it starts diverge at (44). Instead of applying concentration inequalities, we loosely bound the quantity as the following:

$$
\left\| \frac{1}{iBM}\sum_{m=1}^{M}\sum_{j=1}^{iB}\nabla f_{\sigma_k^m(j)}^m(\boldsymbol{x}_{k,0}) - \nabla F(\boldsymbol{x}_{k,0}) \right\|
$$

$$
= \left\| \frac{1}{iBM}\sum_{m=1}^{M}\sum_{j=1}^{iB}\nabla f_{\sigma_k^m(j)}^m(\boldsymbol{x}_{k,0}) - \frac{1}{M}\sum_{m=1}^{M}F^m(\boldsymbol{x}_{k,0}) \right\|
$$

$$
\le \frac{1}{M}\sum_{m=1}^{M}\left\| \frac{1}{iB}\sum_{j=1}^{iB}\nabla f_{\sigma_k^m(j)}^m(\boldsymbol{x}_{k,0}) - F^m(\boldsymbol{x}_{k,0}) \right\| \le \nu.
$$

With this bound, the RHS of the upper bound (45) on $\|\sum_{j=1}^{i}\boldsymbol{g}_j\|$ becomes $iB\nu + iB\|\nabla F(\boldsymbol{x}_{k,0})\|$. This results in the bounds on $\|\boldsymbol{r}_k\|$ and $\|\boldsymbol{r}_k\|^2$ (corresponding to (46) and (47)) that read

$$
\|\boldsymbol{r}_k\| \le L\nu N(N-B) + LN(N-B)\|\nabla F(\boldsymbol{x}_{k,0})\|,
$$
$$
\|\boldsymbol{r}_k\| \le 2L^2\nu^2 N^2(N-B)^2 + 2L^2 N^2(N-B)^2\|\nabla F(\boldsymbol{x}_{k,0})\|^2.
$$

The rest is substituting the bounds above to (40), and going through the same steps to obtain the final bound. The resulting bound that corresponds to (51) is

$$
F(\boldsymbol{x}_{K,\frac{N}{B}}) - F^* \le (1-\eta\mu N)^K(F(\boldsymbol{x}_0)-F^*) + \frac{7\eta^2 L^2\nu^2(N-B)^2}{3\mu},
$$

which, by substituting $\eta = \frac{\log(MNK^2)}{\mu NK}$, yields

$$
F(\boldsymbol{x}_{K,\frac{N}{B}}) - F^* \le \frac{F(\boldsymbol{x}_0)-F^*}{MNK^2} + \tilde{\mathcal{O}}\left(\frac{L^2\nu^2}{\mu^3}\frac{1}{K^2}\right). \tag{53}
$$

To finish the proof of in-expectation bound, choose $\delta = \frac{1}{MN}$. Recall that the probabilistic event $E$ occurs when all our concentration bounds hold. Conditioned on $E$, which occurs with probability at least $1 - \frac{1}{MN}$, the tighter bound (52) holds, with $\log\frac{1}{\delta}$ replaced by $\log(MN)$. The complement event $E^c$ occurs with probability at most $\frac{1}{MN}$, under which the looser bound (53) is true. Thus, in expectation,

$$
\mathbb{E}\left[F(\boldsymbol{x}_{K,\frac{N}{B}})-F^*\right] = \mathbb{P}(E)\mathbb{E}\left[F(\boldsymbol{x}_{K,\frac{N}{B}})-F^*\mid E\right] + \mathbb{P}(E^c)\mathbb{E}\left[F(\boldsymbol{x}_{K,\frac{N}{B}})-F^*\mid E^c\right]
$$

$$
\le \mathbb{E}\left[F(\boldsymbol{x}_{K,\frac{N}{B}})-F^*\mid E\right] + \frac{1}{MN}\mathbb{E}\left[F(\boldsymbol{x}_{K,\frac{N}{B}})-F^*\mid E^c\right]
$$

$$
\le \frac{3(F(\boldsymbol{x}_0)-F^*)}{2MNK^2} + \tilde{\mathcal{O}}\left(\frac{L^2\nu^2}{\mu^3}\frac{1}{MNK^2}\right).
$$

For the remaining high-probability upper bounds proved in the paper, we can similarly follow this process to obtain matching (up to log factors) in-expectation upper bounds.

### D.3  PROOF OF UPPER BOUND FOR LOCAL RR (THEOREM 2)

**One epoch as one step of GD plus noise.**  The update rule of local RR can be written as the following. For $k \in [K]$, $i \in [N]$, and $m \in [M]$,

$$\boldsymbol{x}_{k,i}^m := \begin{cases} \boldsymbol{x}_{k,i-1}^m - \eta \nabla f_{\sigma_k^m(i)}^m(\boldsymbol{x}_{k,i-1}^m) & \text{if } B \text{ does not divide } i, \\ \frac{1}{M} \sum_{m=1}^M (\boldsymbol{x}_{k,i-1}^m - \eta \nabla f_{\sigma_k^m(i)}^m(\boldsymbol{x}_{k,i-1}^m)) =: \boldsymbol{y}_{k,i/B} & \text{if } B \text{ divides } i. \end{cases} \tag{54}$$

Recall that $\boldsymbol{x}_{k,0}^m$'s are the initial points of an epoch and they all the same regardless of the machine $m$. We define $\boldsymbol{y}_{k,0} := \boldsymbol{x}_{k,0}^1$. For any $i$ in the range of $(l-1)B + 1 \le i \le lB$ for some $l \in [M]$, we will use $\boldsymbol{y}_{k,l-1}$ as the "pivot" and decompose the gradients into the ones evaluated at $\boldsymbol{y}_{k,l-1}$ plus noise terms.

$$\nabla f_{\sigma_k^m(i)}^m(\boldsymbol{x}_{k,i-1}^m) = \nabla f_{\sigma_k^m(i)}^m(\boldsymbol{y}_{k,l-1}) + \nabla f_{\sigma_k^m(i)}^m(\boldsymbol{x}_{k,i-1}^m) - \nabla f_{\sigma_k^m(i)}^m(\boldsymbol{y}_{k,l-1})$$

$$= \nabla f_{\sigma_k^m(i)}^m(\boldsymbol{y}_{k,l-1}) + \underbrace{\left[ \int_0^1 \nabla^2 f_{\sigma_k^m(i)}^m(\boldsymbol{y}_{k,l-1} + t(\boldsymbol{x}_{k,i-1}^m - \boldsymbol{y}_{k,l-1})) dt \right]}_{=:\boldsymbol{H}_i^m} (\boldsymbol{x}_{k,i-1}^m - \boldsymbol{y}_{k,l-1}),$$

where the integral $\boldsymbol{H}_i^m$ exists due to the reason discussed in Appendix D.2. Also note from $L$-smoothness of $f_i^m$'s that $\|\boldsymbol{H}_i^m\| \le L$. Using the decomposition, one can unroll the updates (54) and write $\boldsymbol{y}_{k,l}$ in terms of $\boldsymbol{y}_{k,l-1}$ in the following way:

$$\boldsymbol{y}_{k,l} = \boldsymbol{y}_{k,l-1} - \frac{\eta}{M} \sum_{m=1}^M \sum_{i=(l-1)B+1}^{lB} \left( \prod_{j=lB}^{i+1} (\boldsymbol{I} - \eta \boldsymbol{H}_j^m) \right) \nabla f_{\sigma_k^m(i)}^m(\boldsymbol{y}_{k,l-1}), \tag{55}$$

for $l = 1, \ldots, N/B$. Next, we again decompose the gradient $\nabla f_{\sigma_k^m(i)}^m(\boldsymbol{y}_{k,l-1})$, this time using $\boldsymbol{y}_{k,0}$ as the pivot:

$$\nabla f_{\sigma_k^m(i)}^m(\boldsymbol{y}_{k,l-1}) = \nabla f_{\sigma_k^m(i)}^m(\boldsymbol{y}_{k,0}) + \nabla f_{\sigma_k^m(i)}^m(\boldsymbol{y}_{k,l-1}) - \nabla f_{\sigma_k^m(i)}^m(\boldsymbol{y}_{k,0})$$

$$= \nabla f_{\sigma_k^m(i)}^m(\boldsymbol{y}_{k,0}) + \underbrace{\left[ \int_0^1 \nabla^2 f_{\sigma_k^m(i)}^m(\boldsymbol{y}_{k,0} + t(\boldsymbol{y}_{k,l-1} - \boldsymbol{y}_{k,0})) dt \right]}_{=:\tilde{\boldsymbol{H}}_i^m} (\boldsymbol{y}_{k,l-1} - \boldsymbol{y}_{k,0}).$$

This decomposition allows us to rewrite (55) in the following form

$$\boldsymbol{y}_{k,l} = \boldsymbol{y}_{k,l-1} - \eta \boldsymbol{t}_l - \eta \boldsymbol{S}_l(\boldsymbol{y}_{k,l-1} - \boldsymbol{y}_{k,0}), \tag{56}$$

where

$$\boldsymbol{t}_l := \frac{1}{M} \sum_{m=1}^M \sum_{i=(l-1)B+1}^{lB} \left( \prod_{j=lB}^{i+1} (\boldsymbol{I} - \eta \boldsymbol{H}_j^m) \right) \nabla f_{\sigma_k^m(i)}^m(\boldsymbol{y}_{k,0}), \tag{57}$$

$$\boldsymbol{S}_l := \frac{1}{M} \sum_{m=1}^M \sum_{i=(l-1)B+1}^{lB} \left( \prod_{j=lB}^{i+1} (\boldsymbol{I} - \eta \boldsymbol{H}_j^m) \right) \tilde{\boldsymbol{H}}_i^m. \tag{58}$$

Unrolling (56) for $l = 1, \ldots, N/B$ then gives the progress over an epoch:

$$\boldsymbol{y}_{k+1,0} = \boldsymbol{y}_{k,0} - \eta \sum_{l=1}^{N/B} \left( \prod_{j=N/B}^{l+1} (\boldsymbol{I} - \eta \boldsymbol{S}_j) \right) \boldsymbol{t}_l. \tag{59}$$

As done in the proof of Theorem 1 (Appendix D.2), we will express (59) as one step of GD on $F$ plus some noise. Of course, the noise terms here will be more complicated to handle than they were in Theorem 1. Due to summation by parts, the following identity holds:

$$\sum_{l=1}^{N/B} a_l b_l = a_{N/B} \sum_{j=1}^{N/B} b_j - \sum_{l=1}^{N/B-1} (a_{l+1} - a_l) \sum_{j=1}^l b_j.$$

We apply this to the last term of (59), by substituting $a_i = \prod_{j=N/B}^{l+1}(\boldsymbol{I} - \eta\boldsymbol{S}_j)$ and $b_l = \boldsymbol{t}_l$:

$$\eta \sum_{l=1}^{N/B}\left(\prod_{j=N/B}^{l+1}(\boldsymbol{I} - \eta\boldsymbol{S}_j)\right)\boldsymbol{t}_l = \eta\sum_{l=1}^{N/B}\boldsymbol{t}_l - \eta^2\sum_{l=1}^{N/B-1}\left(\prod_{j=N/B}^{l+2}(\boldsymbol{I} - \eta\boldsymbol{S}_j)\right)\boldsymbol{S}_{l+1}\sum_{j=1}^{l}\boldsymbol{t}_j. \quad (60)$$

We also apply the summation by parts to the inner summation of $\boldsymbol{t}_l$'s (57):

$$\boldsymbol{t}_l := \frac{1}{M}\sum_{m=1}^{M}\sum_{i=(l-1)B+1}^{lB}\left(\prod_{j=lB}^{i+1}(\boldsymbol{I} - \eta\boldsymbol{H}_j^m)\right)\nabla f_{\sigma_k^m(i)}^m(\boldsymbol{y}_{k,0})$$

$$= \frac{1}{M}\sum_{m=1}^{M}\sum_{i=(l-1)B+1}^{lB}\nabla f_{\sigma_k^m(i)}^m(\boldsymbol{y}_{k,0})$$

$$- \frac{\eta}{M}\sum_{m=1}^{M}\sum_{i=(l-1)B+1}^{lB-1}\left(\prod_{t=lB}^{i+2}(\boldsymbol{I} - \eta\boldsymbol{H}_t^m)\right)\boldsymbol{H}_{i+1}^m\sum_{j=(l-1)B+1}^{i}\nabla f_{\sigma_k^m(j)}^m(\boldsymbol{y}_{k,0}) \quad (61)$$

Substituting (61) to (60) gives

$$\boldsymbol{y}_{k+1,0} = \boldsymbol{y}_{k,0} - \eta N\nabla F(\boldsymbol{y}_{k,0}) + \eta^2\boldsymbol{r}_{k,1} + \eta^2\boldsymbol{r}_{k,2} - \eta^3\boldsymbol{r}_{k,3},$$

where $\boldsymbol{r}_{k,1}$, $\boldsymbol{r}_{k,2}$, and $\boldsymbol{r}_{k,3}$ are noise terms defined as

$$\boldsymbol{r}_{k,1} := \frac{1}{M}\sum_{l=1}^{N/B}\sum_{m=1}^{M}\sum_{i=(l-1)B+1}^{lB-1}\left(\prod_{t=lB}^{i+2}(\boldsymbol{I} - \eta\boldsymbol{H}_t^m)\right)\boldsymbol{H}_{i+1}^m\sum_{t=(l-1)B+1}^{i}\nabla f_{\sigma_k^m(t)}^m(\boldsymbol{y}_{k,0}),$$

$$\boldsymbol{r}_{k,2} := \frac{1}{M}\sum_{l=1}^{N/B-1}\left(\prod_{j=N/B}^{l+2}(\boldsymbol{I} - \eta\boldsymbol{S}_j)\right)\boldsymbol{S}_{l+1}\sum_{m=1}^{M}\sum_{t=1}^{lB}\nabla f_{\sigma_k^m(t)}^m(\boldsymbol{y}_{k,0}),$$

$$\boldsymbol{r}_{k,3} := \frac{1}{M}\sum_{l=1}^{N/B-1}\left(\prod_{j=N/B}^{l+2}(\boldsymbol{I} - \eta\boldsymbol{S}_j)\right)\boldsymbol{S}_{l+1}\times$$

$$\sum_{j=1}^{l}\sum_{m=1}^{M}\sum_{i=(j-1)B+1}^{jB-1}\left(\prod_{t=jB}^{i+2}(\boldsymbol{I} - \eta\boldsymbol{H}_t^m)\right)\boldsymbol{H}_{i+1}^m\sum_{t=(j-1)B+1}^{i}\nabla f_{\sigma_k^m(t)}^m(\boldsymbol{y}_{k,0}).$$

Defining $\boldsymbol{r}_k := \boldsymbol{r}_{k,1} + \boldsymbol{r}_{k,2} - \eta\boldsymbol{r}_{k,3}$, it follows from $L$-smoothness of $F$ that

$$F(\boldsymbol{y}_{k+1,0}) - F(\boldsymbol{y}_{k,0})$$

$$\leq \langle\nabla F(\boldsymbol{y}_{k,0}), \boldsymbol{y}_{k+1,0} - \boldsymbol{y}_{k,0}\rangle + \frac{L}{2}\|\boldsymbol{y}_{k+1,0} - \boldsymbol{y}_{k,0}\|^2$$

$$\leq -\eta N\|\nabla F(\boldsymbol{y}_{k,0})\|^2 + \eta^2\|\nabla F(\boldsymbol{y}_{k,0})\|\|\boldsymbol{r}_k\| + \frac{\eta^2 L}{2}\|N\nabla F(\boldsymbol{y}_{k,0}) + \eta\boldsymbol{r}_k\|^2$$

$$\leq (-\eta N + \eta^2 LN^2)\|\nabla F(\boldsymbol{y}_{k,0})\|^2 + \eta^2\|\nabla F(\boldsymbol{y}_{k,0})\|\|\boldsymbol{r}_k\| + \eta^4 L\|\boldsymbol{r}_k\|^2. \quad (62)$$

**Bounding noise terms using concentration.** We next bound $\|\boldsymbol{r}_k\|$ by bounding each $\|\boldsymbol{r}_{k,1}\|$, $\|\boldsymbol{r}_{k,2}\|$, and $\|\boldsymbol{r}_{k,3}\|$. From this point on, we write $\boldsymbol{g}_t^m := \nabla f_{\sigma_k^m(t)}^m(\boldsymbol{y}_{k,0})$ to simplify notation.

$$\|\boldsymbol{r}_{k,1}\| = \left\|\frac{1}{M}\sum_{l=1}^{N/B}\sum_{m=1}^{M}\sum_{i=(l-1)B+1}^{lB-1}\left(\prod_{t=lB}^{i+2}(\boldsymbol{I} - \eta\boldsymbol{H}_t^m)\right)\boldsymbol{H}_{i+1}^m\sum_{t=(l-1)B+1}^{i}\boldsymbol{g}_t^m\right\|$$

$$\leq \frac{1}{M}\sum_{l=1}^{N/B}\sum_{m=1}^{M}\sum_{i=(l-1)B+1}^{lB-1}\left\|\left(\prod_{t=lB}^{i+2}(\boldsymbol{I} - \eta\boldsymbol{H}_t^m)\right)\boldsymbol{H}_{i+1}^m\sum_{t=(l-1)B+1}^{i}\boldsymbol{g}_t^m\right\|$$

$$\leq \frac{L(1+\eta L)^B}{M}\sum_{l=1}^{N/B}\sum_{m=1}^{M}\sum_{i=(l-1)B+1}^{lB-1}\left\|\sum_{t=(l-1)B+1}^{i}\boldsymbol{g}_t^m\right\|, \quad (63)$$

where we used $\|\boldsymbol{H}_i^m\| \leq L$. Recall from the theorem statement that $K \geq 7\rho\kappa\log(MNK^2)$ and $\eta = \frac{\log(MNK^2)}{\mu NK}$. This means that

$$(1+\eta L)^B = \left(1 + \frac{\kappa\log(MNK^2)}{NK}\right)^B \leq \left(1 + \frac{1}{7\rho N}\right)^B \leq \left(1 + \frac{1}{7B}\right)^B \leq e^{1/7}. \tag{64}$$

Also note from the definition of $\boldsymbol{S}_l$ (58) that $\|\boldsymbol{S}_l\| \leq LB(1+\eta L)^B \leq e^{1/7}LB$, which we use to get similar bounds for the next two terms $\boldsymbol{r}_{k,2}$ and $\boldsymbol{r}_{k,3}$.

$$\|\boldsymbol{r}_{k,2}\| = \left\|\frac{1}{M}\sum_{l=1}^{N/B-1}\left(\prod_{j=N/B}^{l+2}(\boldsymbol{I}-\eta\boldsymbol{S}_j)\right)\boldsymbol{S}_{l+1}\sum_{m=1}^{M}\sum_{t=1}^{lB}\boldsymbol{g}_t^m\right\|$$

$$\leq \frac{e^{1/7}LB(1+e^{1/7}\eta LB)^{N/B}}{M}\sum_{l=1}^{N/B-1}\left\|\sum_{m=1}^{M}\sum_{t=1}^{lB}\boldsymbol{g}_t^m\right\|, \tag{65}$$

and we can bound

$$(1+e^{1/7}\eta LB)^{N/B} = \left(1 + \frac{e^{1/7}\kappa B\log(MNK^2)}{NK}\right)^{N/B}$$

$$\leq \left(1 + \frac{e^{1/7}B}{7\rho N}\right)^{N/B} \leq \exp\left(\frac{e^{1/7}}{7}\right). \tag{66}$$

We similarly bound the norm of the last noise term $\boldsymbol{r}_{k,3}$:

$$\|\boldsymbol{r}_{k,3}\| \leq \frac{e^{1/7}LB(1+e^{1/7}\eta LB)^{N/B}}{M}\times$$

$$\sum_{l=1}^{N/B-1}\left\|\sum_{j=1}^{l}\sum_{m=1}^{M}\sum_{i=(j-1)B+1}^{jB-1}\left(\prod_{t=jB}^{i+2}(\boldsymbol{I}-\eta\boldsymbol{H}_t^m)\right)\boldsymbol{H}_{i+1}^m\sum_{t=(j-1)B+1}^{i}\boldsymbol{g}_t^m\right\|$$

$$\leq \frac{e^{1/7}L^2B(1+e^{1/7}\eta LB)^{N/B}(1+\eta L)^B}{M}\sum_{l=1}^{N/B-1}\sum_{j=1}^{l}\sum_{m=1}^{M}\sum_{i=(j-1)B+1}^{jB-1}\left\|\sum_{t=(j-1)B+1}^{i}\boldsymbol{g}_t^m\right\|$$

$$\leq \frac{11L^2B}{7M}\sum_{l=1}^{N/B-1}\sum_{j=1}^{l}\sum_{m=1}^{M}\sum_{i=(j-1)B+1}^{jB-1}\left\|\sum_{t=(j-1)B+1}^{i}\boldsymbol{g}_t^m\right\|, \tag{67}$$

where the last inequality used (64), (66), and $e^{2/7}\exp(e^{1/7}/7) \leq 11/7$.

Given the bounds (63), (65), and (67), we now use Lemma 8 to get high-probability bounds for the partial sums of $\boldsymbol{g}_t^m$ that appear in the bounds. For any $i$ satisfying $(j-1)B+1 \leq i \leq jB-1$, where $j \in [N/B]$, and for any $m \in [M]$, the following bound holds with probability at least $1 - \frac{\delta}{2MNK}$:

$$\left\|\frac{1}{i-(j-1)B}\sum_{t=(l-1)B+1}^{i}\boldsymbol{g}_t^m - \frac{1}{N}\sum_{t=1}^{N}\boldsymbol{g}_t^m\right\|$$

$$= \left\|\frac{1}{i-(j-1)B}\sum_{t=(l-1)B+1}^{i}\nabla f_{\sigma_k^m(t)}^m(\boldsymbol{y}_{k,0}) - \nabla F^m(\boldsymbol{y}_{k,0})\right\| \leq \nu\sqrt{\frac{8\log\frac{4MNK}{\delta}}{i-(j-1)B}}.$$

From this, with probability at least $1 - \frac{\delta}{2MNK}$ we have

$$\left\|\sum_{t=(j-1)B+1}^{i}\boldsymbol{g}_t^m\right\| \leq \nu\sqrt{8(i-(j-1)B)\log\frac{4MNK}{\delta}} + (i-(j-1)B)\|\nabla F^m(\boldsymbol{y}_{k,0})\|. \tag{68}$$

Similarly, for $l \in [N/B-1]$, the following bound holds with probability at least $1 - \frac{B\delta}{2NK}$:

$$\left\|\frac{1}{lBM}\sum_{m=1}^{M}\sum_{t=1}^{lB}\boldsymbol{g}_t^m - \frac{1}{MN}\sum_{m=1}^{M}\sum_{t=1}^{N}\boldsymbol{g}_t^m\right\|$$

$$= \left\| \frac{1}{lBM} \sum_{m=1}^{M} \sum_{t=1}^{lB} \nabla f_{\sigma_k^m(t)}^m(\boldsymbol{y}_{k,0}) - \nabla F(\boldsymbol{y}_{k,0}) \right\| \leq \nu \sqrt{\frac{8 \log \frac{4NK}{B\delta}}{lBM}},$$

which gives us

$$\left\| \sum_{m=1}^{M} \sum_{t=1}^{lB} \boldsymbol{g}_t^m \right\| \leq \nu \sqrt{8lBM \log \frac{4NK}{B\delta}} + lBM \, \|\nabla F(\boldsymbol{y}_{k,0})\| \tag{69}$$

By applying the union bound, with probability at least $1 - \frac{\delta}{K}$, the bound (68) holds for all $m \in [M]$ and $i \in \bigcup_{j=1}^{N/B}[(j-1)B + 1 : jB - 1]$, and the bound (69) holds for all $l \in [N/B - 1]$.

We now substitute the bounds (68) and (69) to (63), (65), and (67) to get upper bounds for $\|\boldsymbol{r}_{k,1}\|$, $\|\boldsymbol{r}_{k,2}\|$, and $\|\boldsymbol{r}_{k,3}\|$, respectively. First,

$$\|\boldsymbol{r}_{k,1}\|$$

$$\leq \frac{e^{1/7}L}{M} \sum_{l=1}^{N/B} \sum_{m=1}^{M} \sum_{i=(l-1)B+1}^{lB-1} \left( \nu \sqrt{8(i - (l-1)B) \log \frac{4MNK}{\delta}} + (i - (l-1)B) \, \|\nabla F^m(\boldsymbol{y}_{k,0})\| \right)$$

$$= \frac{e^{1/7}\sqrt{8}L\nu N}{B} \left( \sum_{i=1}^{B-1} \sqrt{i} \right) \sqrt{\log \frac{4MNK}{\delta}} + \frac{e^{1/7}LN}{B} \left( \sum_{i=1}^{B-1} i \right) \left( \frac{1}{M} \sum_{m=1}^{M} \|\nabla F^m(\boldsymbol{y}_{k,0})\| \right)$$

$$\leq \frac{24L\nu N(B^{3/2} - 1)}{11B} \sqrt{\log \frac{4MNK}{\delta}} + \frac{3LN(B-1)}{5} (\tau + \rho \, \|\nabla F(\boldsymbol{y}_{k,0})\|), \tag{70}$$

where the last inequality used $\sum_{i=1}^{B-1} \sqrt{i} \leq \int_1^B \sqrt{z} dz = \frac{2}{3}(B^{3/2} - 1)$ and Assumption 3. For the next noise term, we have $e^{1/7}(1 + e^{1/7}\eta LB)^{N/B} \leq 7/5$, so

$$\|\boldsymbol{r}_{k,2}\| \leq \frac{7LB}{5M} \sum_{l=1}^{N/B-1} \left( \nu \sqrt{8lBM \log \frac{4NK}{B\delta}} + lBM \, \|\nabla F(\boldsymbol{y}_{k,0})\| \right)$$

$$= \frac{7\sqrt{8}L\nu B^{3/2}}{5M^{1/2}} \left( \sum_{l=1}^{N/B-1} \sqrt{l} \right) \sqrt{\log \frac{4NK}{B\delta}} + \frac{7LB^2}{5} \left( \sum_{l=1}^{N/B-1} l \right) \|\nabla F(\boldsymbol{y}_{k,0})\|$$

$$\leq \frac{8L\nu(N^{3/2} - B^{3/2})}{3M^{1/2}} \sqrt{\log \frac{4NK}{B\delta}} + \frac{7LN(N-B)}{10} \|\nabla F(\boldsymbol{y}_{k,0})\|. \tag{71}$$

Lastly,

$$\|\boldsymbol{r}_{k,3}\| \leq \frac{11L^2 B}{7M} \sum_{l=1}^{N/B-1} \sum_{j=1}^{l} \sum_{m=1}^{M} \sum_{i=(j-1)B+1}^{jB-1} \left( \nu \sqrt{8(i - (j-1)B) \log \frac{4MNK}{\delta}} \right.$$

$$\left. + (i - (j-1)B) \, \|\nabla F^m(\boldsymbol{y}_{k,0})\| \right)$$

$$= \frac{11\sqrt{8}L^2 \nu B}{7} \left( \sum_{l=1}^{N/B-1} l \right) \left( \sum_{i=1}^{B-1} \sqrt{i} \right) \sqrt{\log \frac{4MNK}{\delta}}$$

$$+ \frac{11L^2 B}{7} \left( \sum_{l=1}^{N/B-1} l \right) \left( \sum_{i=1}^{B-1} i \right) \left( \frac{1}{M} \sum_{m=1}^{M} \|\nabla F^m(\boldsymbol{y}_{k,0})\| \right)$$

$$\leq \frac{3L^2 \nu N(N-B)(B^{3/2} - 1)}{2B} \sqrt{\log \frac{4MNK}{\delta}}$$

$$+ \frac{2L^2 N(N-B)(B-1)}{5} (\tau + \rho \, \|\nabla F(\boldsymbol{y}_{k,0})\|). \tag{72}$$

Recalling the definition $r_k := r_{k,1} + r_{k,2} - \eta r_{k,3}$, we get an upper bound for $\|r_k\|$ from (70), (71), and (72):

$$
\begin{aligned}
\|r_k\| &\leq \|r_{k,1}\| + \|r_{k,2}\| + \eta \|r_{k,3}\| \\
&\leq L\nu\sqrt{\log\frac{4MNK}{\delta}}\left(\frac{24N(B^{3/2}-1)}{11B} + \frac{8(N^{3/2}-B^{3/2})}{3M^{1/2}} + \frac{3\eta LN(N-B)(B^{3/2}-1)}{2B}\right) \\
&\quad + L\tau\left(\frac{3N(B-1)}{5} + \frac{2\eta LN(N-B)(B-1)}{5}\right) \\
&\quad + L\|\nabla F(y_{k,0})\|\left(\frac{3\rho N(B-1)}{5} + \frac{7N(N-B)}{10} + \frac{2\eta L\rho N(N-B)(B-1)}{5}\right). \quad (73)
\end{aligned}
$$

Recall again that we have $K \geq 7\rho\kappa\log(MNK^2)$ and $\eta = \frac{\log(MNK^2)}{\mu NK}$, so $\eta LN \leq 1/7$. Using this and $N - B \leq N$, we can further simplify (73).

$$
\begin{aligned}
\|r_k\| &\leq L\nu\sqrt{\log\frac{4MNK}{\delta}}\underbrace{\left(\frac{12N(B^{3/2}-1)}{5B} + \frac{8(N^{3/2}-B^{3/2})}{3M^{1/2}}\right)}_{=:\Phi} + \frac{2L\tau N(B-1)}{3} \\
&\quad + L\|\nabla F(y_{k,0})\|\left(\frac{2\rho N(B-1)}{3} + \frac{7N(N-B)}{10}\right) \\
&\leq L\nu\Phi\sqrt{\log\frac{4MNK}{\delta}} + \frac{2L\tau N(B-1)}{3} + \frac{7L\rho N^2}{5}\|\nabla F(y_{k,0})\|, \quad (74)
\end{aligned}
$$

which holds with probability at least $1 - \frac{\delta}{K}$. The bound (74) holds for all $k \in [K]$ with probability $1 - \delta$ if we apply the union bound over $k = 1, \dots, K$. Next, by $(a + b + c)^2 \leq 3a^2 + 3b^2 + 3c^2$, we have

$$
\|r_k\|^2 \leq 3L^2\nu^2\Phi^2\log\frac{4MNK}{\delta} + \frac{4L^2\tau^2 N^2(B-1)^2}{3} + \frac{147L^2\rho^2 N^4}{25}\|\nabla F(y_{k,0})\|^2, \quad (75)
$$

which also holds for all $k \in [K]$ with probability at least $1 - \delta$.

**Getting a high-probability convergence rate.** Given our high-probability bounds (74) and (75), we can substitute them to (62) and get

$$
\begin{aligned}
F(y_{k+1,0}) &- F(y_{k,0}) \\
&\leq (-\eta N + \eta^2 LN^2)\|\nabla F(y_{k,0})\|^2 + \eta^2\|\nabla F(y_{k,0})\|\,\|r_k\| + \eta^4 L\|r_k\|^2 \\
&\leq \left(-\eta N + \eta^2 LN^2 + \frac{7\eta^2 L\rho N^2}{5} + \frac{147\eta^4 L^3\rho^2 N^4}{25}\right)\|\nabla F(y_{k,0})\|^2 \\
&\quad + \eta^2 L\nu\Phi\sqrt{\log\frac{4MNK}{\delta}}\|\nabla F(y_{k,0})\| + 3\eta^4 L^3\nu^2\Phi^2\log\frac{4MNK}{\delta} \\
&\quad + \frac{2\eta^2 L\tau N(B-1)}{3}\|\nabla F(y_{k,0})\| + \frac{4\eta^4 L^3\tau^2 N^2(B-1)^2}{3}. \quad (76)
\end{aligned}
$$

The following terms in the RHS of (76) can be bounded using $ab \leq \frac{a^2}{2} + \frac{b^2}{2}$:

$$
\begin{aligned}
&\eta^2 L\nu\Phi\sqrt{\log\frac{4MNK}{\delta}}\|\nabla F(y_{k,0})\| \\
&= \left(\frac{\eta^{1/2} N^{1/2}}{\sqrt{8}}\|\nabla F(y_{k,0})\|\right)\left(\frac{\sqrt{8}\eta^{3/2} L\nu\Phi}{N^{1/2}}\sqrt{\log\frac{4MNK}{\delta}}\right) \\
&\leq \frac{\eta N}{16}\|\nabla F(y_{k,0})\|^2 + \frac{4\eta^3 L^2\nu^2\Phi^2}{N}\log\frac{4MNK}{\delta}, \quad (77) \\
&\frac{2\eta^2 L\tau N(B-1)}{3}\|\nabla F(y_{k,0})\|
\end{aligned}
$$

$$= \left( \frac{\eta^{1/2} N^{1/2}}{\sqrt{8}} \|\nabla F(\boldsymbol{y}_{k,0})\| \right) \left( \frac{2\sqrt{8}\eta^{3/2} L\tau N^{1/2}(B-1)}{3} \right)$$

$$\leq \frac{\eta N}{16} \|\nabla F(\boldsymbol{y}_{k,0})\|^2 + \frac{16\eta^3 L^2 \tau^2 N(B-1)^2}{9}. \tag{78}$$

Substituting (77) and (78) to (76) results in

$$F(\boldsymbol{y}_{k+1,0}) - F(\boldsymbol{y}_{k,0})$$

$$\leq \left( -\frac{7}{8}\eta N + \eta^2 LN^2 + \frac{7\eta^2 L\rho N^2}{5} + \frac{147\eta^4 L^3 \rho^2 N^4}{25} \right) \|\nabla F(\boldsymbol{y}_{k,0})\|^2$$

$$+ \frac{\eta^3 L^2 \nu^2 (4+3\eta LN)\Phi^2}{N} \log \frac{4MNK}{\delta} + \frac{4\eta^3 L^2 \tau^2 (4+3\eta LN)N(B-1)^2}{9}. \tag{79}$$

Again, we have $K \geq 7\rho\kappa \log(MNK^2)$ and $\eta = \frac{\log(MNK^2)}{\mu NK}$, so $\eta L\rho N \leq \frac{1}{7}$. Since the inequality $-\frac{7}{8}z + \frac{12}{5}z^2 + \frac{147}{25}z^4 \leq -\frac{1}{2}z$ holds on $z \in [0, \frac{1}{7}]$, we have

$$-\frac{7}{8}\eta N + \eta^2 LN^2 + \frac{7\eta^2 L\rho N^2}{5} + \frac{147\eta^4 L^3 \rho^2 N^4}{25}$$

$$\leq -\frac{7}{8}\eta N + \frac{12\eta^2 L\rho N^2}{5} + \frac{147\eta^4 L^3 \rho^3 N^4}{25} \leq -\frac{1}{2}\eta N.$$

Substituting this inequality to (79), together with $4 + 3\eta LN \leq \frac{31}{7} < \frac{9}{2}$, yields

$$F(\boldsymbol{y}_{k+1,0}) - F(\boldsymbol{y}_{k,0}) \leq -\frac{\eta N}{2} \|\nabla F(\boldsymbol{y}_{k,0})\|^2 + \frac{9\eta^3 L^2 \nu^2 \Phi^2}{2N} \log \frac{4MNK}{\delta} + 2\eta^3 L^2 \tau^2 N(B-1)^2.$$

We now recall that $F$ is $\mu$-PŁ, so $\|\nabla F(\boldsymbol{y}_{k,0})\|^2 \geq 2\mu(F(\boldsymbol{y}_{k,0}) - F^*)$:

$$F(\boldsymbol{y}_{k+1,0}) - F^* \leq (1-\eta\mu N)(F(\boldsymbol{y}_{k,0}) - F^*)$$

$$+ \frac{9\eta^3 L^2 \nu^2 \Phi^2}{2N} \log \frac{4MNK}{\delta} + 2\eta^3 L^2 \tau^2 N(B-1)^2. \tag{80}$$

Recall that (80) holds for all $k \in [K]$, with probability $1 - \delta$. Therefore, by unrolling the inequality,

$$F(\boldsymbol{y}_{K,\frac{N}{B}}) - F^* \leq (1-\eta\mu N)^K (F(\boldsymbol{y}_0) - F^*)$$

$$+ \left( \frac{9\eta^3 L^2 \nu^2 \Phi^2}{2N} \log \frac{4MNK}{\delta} + 2\eta^3 L^2 \tau^2 N(B-1)^2 \right) \sum_{k=0}^{K-1} (1-\eta\mu N)^k$$

$$\leq (1-\eta\mu N)^K (F(\boldsymbol{y}_0) - F^*)$$

$$+ \frac{9\eta^2 L^2 \nu^2 \Phi^2}{2\mu N^2} \log \frac{4MNK}{\delta} + \frac{2\eta^2 L^2 \tau^2 (B-1)^2}{\mu}. \tag{81}$$

Recall that $\Phi := \frac{12N(B^{3/2}-1)}{5B} + \frac{8(N^{3/2}-B^{3/2})}{3M^{1/2}}$, hence

$$\Phi^2 \leq \frac{288N^2(B^{3/2}-1)^2}{25B^2} + \frac{128(N^{3/2}-B^{3/2})^2}{9M}.$$

Substituting this inequality and also $\eta = \frac{\log(MNK^2)}{\mu NK}$ gives

$$F(\boldsymbol{y}_{K,\frac{N}{B}}) - F^*$$

$$\leq \frac{F(\boldsymbol{y}_0) - F^*}{MNK^2} + \frac{2L^2 \tau^2 (B-1)^2}{\mu^3 N^2 K^2} \log^2(MNK^2)$$

$$+ \frac{9L^2 \nu^2}{2\mu^3 N^4 K^2} \log \frac{4MNK}{\delta} \log^2(MNK^2) \left( \frac{288N^2(B^{3/2}-1)^2}{25B^2} + \frac{128(N^{3/2}-B^{3/2})^2}{9M} \right)$$

$$= \frac{F(\boldsymbol{y}_0) - F^*}{MNK^2} + \tilde{\mathcal{O}} \left( \frac{L^2 \tau^2}{\mu^3} \frac{B^2}{N^2 K^2} \right) + \tilde{\mathcal{O}} \left( \frac{L^2 \nu^2}{\mu^3} \frac{B}{N^2 K^2} \right) + \tilde{\mathcal{O}} \left( \frac{L^2 \nu^2}{\mu^3} \frac{1}{MNK^2} \right),$$

with probability at least $1 - \delta$. This finishes the proof.

### D.4 PROOF OF UPPER BOUND FOR MINIBATCH RR WITH SYNCSHUF (THEOREM 6)

The first part ("One epoch as one step of GD plus noise") of the proof is identical to that of Theorem 1. We start from the second part.

**Bounding noise term using concentration.** It is left to bound $\|\boldsymbol{r}_k\|$. As seen in (41), we have

$$\|\boldsymbol{r}_k\| \leq LB(1+\eta LB)^{N/B} \sum_{i=1}^{N/B-1} \left\| \sum_{j=1}^{i} \boldsymbol{g}_j \right\|. \tag{82}$$

Recall from the theorem statement that $K \geq 6\kappa \log(MNK^2)$ and $\eta = \frac{\log(MNK^2)}{\mu NK}$. This means that

$$(1+\eta LB)^{N/B} = \left(1 + \frac{\kappa B \log(MNK^2)}{NK}\right)^{N/B} \leq \left(1 + \frac{B}{6N}\right)^{N/B} \leq e^{1/6}. \tag{83}$$

Next, we bound the norm of

$$\sum_{j=1}^{i} \boldsymbol{g}_j = \frac{1}{M} \sum_{m=1}^{M} \sum_{j=1}^{iB} \nabla f_{\sigma_k^m(j)}^m(\boldsymbol{x}_{k,0}), \tag{84}$$

exploiting our modification SYNCSHUF as well as Lemma 8. For each $\nabla f_{\sigma_k^m(j)}^m(\boldsymbol{x}_{k,0})$, we first add and subtract its corresponding $\nabla \bar{f}_{\sigma_k^m(j)}(\boldsymbol{x}_{k,0})$, where $\bar{f}_i := \frac{1}{M}\sum_{m=1}^{M} f_i^m$ as defined in Assumption 4. This way, (84) can be decomposed into two sums $\sum_{j=1}^{i} \boldsymbol{g}_j = \frac{1}{M}(\boldsymbol{p}_i + \boldsymbol{q}_i)$, where

$$\boldsymbol{p}_i := \sum_{m=1}^{M} \sum_{j=1}^{iB} \nabla \bar{f}_{\sigma_k^m(j)}(\boldsymbol{x}_{k,0}),$$

$$\boldsymbol{q}_i := \sum_{m=1}^{M} \sum_{j=1}^{iB} \nabla f_{\sigma_k^m(j)}^m(\boldsymbol{x}_{k,0}) - \nabla \bar{f}_{\sigma_k^m(j)}(\boldsymbol{x}_{k,0}).$$

Using this decomposition, we will derive high-probability bounds for $\|\boldsymbol{p}_i\|$ and $\|\boldsymbol{q}_i\|$.

To simplify expressions to follow, we decompose $iBM$ (i.e., the total number of component gradients that are summed up) into a multiple of $N$ and the remainder. Let

$$\alpha(i) := \left\lfloor \frac{iBM}{N} \right\rfloor, \quad \beta(i) := iBM - N\alpha(i),$$

so that $iBM$ is decomposed into $N\alpha(i)$ and the remainder $0 \leq \beta(i) < N$. Using this new notation, we can write $\boldsymbol{p}_i$ as

$$\boldsymbol{p}_i = \sum_{m=1}^{M} \sum_{j=1}^{\frac{N\alpha(i)}{M}} \nabla \bar{f}_{\sigma_k^m(j)}(\boldsymbol{x}_{k,0}) + \sum_{m=1}^{M} \sum_{j=\frac{N\alpha(i)}{M}+1}^{iB} \nabla \bar{f}_{\sigma_k^m(j)}(\boldsymbol{x}_{k,0}). \tag{85}$$

Here, recall that with SYNCSHUF, we defined $\sigma_k^m(j) := \sigma((j + \frac{N}{M}\pi(m)) \bmod N)$. With this choice of "shifted" permutations, one can notice that $\{\sigma_k^m(j)\}_{m=1,j=1}^{M,\frac{N}{M}} = [N]$, meaning that adding $\bar{f}_{\sigma_k^m(j)}$ for $m \in [M]$ and $j \in [N/M]$ results in the sum of all $N$ $\bar{f}_i$'s. In fact, this happens if we sum over $m \in [M]$ and any $N/M$ consecutive $j$'s. From this observation and $F = \frac{1}{N}\sum_{i=1}^{N} \bar{f}_i$, (85) can be written as

$$\boldsymbol{p}_i = N\alpha(i)\nabla F(\boldsymbol{x}_{k,0}) + \sum_{m=1}^{M} \sum_{j=\frac{N\alpha(i)}{M}+1}^{iB} \nabla \bar{f}_{\sigma((j+\frac{Nm}{M}) \bmod N)}(\boldsymbol{x}_{k,0}). \tag{86}$$

Assume for now that $\beta(i) > 0$, i.e., $N\alpha(i) < iBM$. The summation in the second term of RHS in (86) is a without-replacement sum (note that the indices $j + \frac{Nm}{M}$ do not overlap) of $\beta(i)$ terms.

Hence, it is equal in distribution to $\sum_{j=1}^{\beta(i)} \nabla \bar{f}_{\sigma(j)}(\boldsymbol{x}_{k,0})$. Also, from Assumption 2, it can be easily checked that for any $i \in [N]$

$$\left\| \nabla \bar{f}_i(\boldsymbol{x}_{k,0}) - \nabla F(\boldsymbol{x}_{k,0}) \right\| \leq \nu.$$

These observations mean that we can apply Lemma 8 to get a concentration bound, with $M \leftarrow 1$, $\boldsymbol{v}_i^1 \leftarrow \nabla \bar{f}_i(\boldsymbol{x}_{k,0})$, $n \leftarrow \beta(i)$, and $\delta \leftarrow \frac{B\delta}{2NK}$. By Lemma 8, with probability at least $1 - \frac{B\delta}{2NK}$ (over the randomness in $\sigma$), we have

$$\left\| \frac{1}{\beta(i)} \sum_{m=1}^{M} \sum_{j=\frac{N\alpha(i)}{M}+1}^{iB} \nabla \bar{f}_{\sigma((j+\frac{Nm}{M}) \bmod N)}(\boldsymbol{x}_{k,0}) - \nabla F(\boldsymbol{x}_{k,0}) \right\| \leq \nu \sqrt{\frac{8 \log \frac{4NK}{B\delta}}{\beta(i)}}. \tag{87}$$

Combining (86) and (87), we get the following upper bound on $\|\boldsymbol{p}_i\|$, which holds with probability at least $1 - \frac{B\delta}{2NK}$.

$$\|\boldsymbol{p}_i\| \leq \|iBM \nabla F(\boldsymbol{x}_{k,0})\| + \left\| \sum_{m=1}^{M} \sum_{j=\frac{N\alpha(i)}{M}+1}^{iB} \nabla \bar{f}_{\sigma((j+\frac{Nm}{M}) \bmod N)}(\boldsymbol{x}_{k,0}) - \beta(i)\nabla F(\boldsymbol{x}_{k,0}) \right\|$$

$$\leq iBM \|\nabla F(\boldsymbol{x}_{k,0})\| + \nu \sqrt{8\beta(i) \log \frac{4NK}{B\delta}}$$

$$\leq iBM \|\nabla F(\boldsymbol{x}_{k,0})\| + \nu \sqrt{N} \sqrt{8 \log \frac{4NK}{B\delta}}, \tag{88}$$

where the last inequality used $\beta(i) < N$. Also recall that we assumed $\beta(i) > 0$ in order to use Lemma 8 and derive (88). However, note that even with $\beta(i) = 0$, the bound (88) trivially holds.

We next bound $\|\boldsymbol{q}_i\|$. This time, we will apply Lemma 8 to the permutation $\pi$ over the local machines. To do this, we will condition on a fixed instantiation of the permutation $\sigma$ and derive a high-probability bound that holds with conditional probability at least $1 - \frac{B\delta}{2NK}$. The conditional probability is at least $1 - \frac{B\delta}{2NK}$ irrespective of the choice of $\sigma$, so we can conclude that the (unconditional) probability that our bound holds is also at least $1 - \frac{B\delta}{2NK}$.

Without loss of generality, choose the instantiation $\sigma(l) = l$ for all $l \in [N]$. With this $\sigma$, we have $\sigma_k^m(j) := (j + \frac{N}{M}\pi(m)) \bmod N$, so the vector $\boldsymbol{q}_i$ reads

$$\boldsymbol{q}_i = \sum_{m=1}^{M} \sum_{j=1}^{iB} \nabla f_{(j+\frac{N}{M}\pi(m)) \bmod N}^m(\boldsymbol{x}_{k,0}) - \nabla \bar{f}_{(j+\frac{N}{M}\pi(m)) \bmod N}(\boldsymbol{x}_{k,0}). \tag{89}$$

Let us consider rewriting this summation as the sum over $l \in [N]$, where $l$ appears in the subscript of the component functions. One can check that

$$l = \left( j + \frac{N}{M}\pi(m) \right) \bmod N \quad \Leftrightarrow \quad \frac{N}{M} \mid (l - j) \text{ and } \pi(m) = (l-j)\frac{M}{N} \bmod M,$$

where $a \mid b$ denotes "$a$ divides $b$." From this, we can rewrite (89) as

$$\boldsymbol{q}_i = \sum_{l=1}^{N} \underbrace{\sum_{\substack{j \in [iB] \\ \frac{N}{M} \mid (l-j)}} \left( \nabla f_l^{\pi^{-1}\left((l-j)\frac{M}{N} \bmod M\right)}(\boldsymbol{x}_{k,0}) - \nabla \bar{f}_l(\boldsymbol{x}_{k,0}) \right)}_{=:\boldsymbol{q}_{i,l}}. \tag{90}$$

By the same reasoning above and below (86), we can see from (90) that for a given index $l \in [N]$, the cardinality of the set $\mathcal{J}_l := \{ j \in [iB] : \frac{N}{M} \mid (l - j) \}$ is either $\alpha(i)$ or $\alpha(i) + 1$. From this, we notice that each $\boldsymbol{q}_{i,l}$ is a without-replacement sum of $\alpha(i)$ or $\alpha(i) + 1$ terms. For now, suppose $\alpha(i) > 0$. For each $\boldsymbol{q}_{i,l}$, we can apply Lemma 8 to it, and show that with probability (conditioned on the instantiation $\sigma(l) = l$) at least $1 - \frac{B\delta}{2N^2K}$, we have

$$\|\boldsymbol{q}_{i,l}\| \leq \begin{cases} \lambda \sqrt{8(\alpha(i)) \log \frac{4N^2K}{B\delta}} & \text{if } |\mathcal{J}_l| = \alpha(i), \\ \lambda \sqrt{8(\alpha(i)+1) \log \frac{4N^2K}{B\delta}} & \text{if } |\mathcal{J}_l| = \alpha(i) + 1. \end{cases}$$

Note that cases in the RHS are all bounded from above by $\lambda\sqrt{16\alpha(i)\log\frac{4N^2K}{B\delta}}$. Applying union bound on all $l \in [N]$, we get that with probability at least $1 - \frac{B\delta}{2NK}$, we have

$$\|\boldsymbol{q}_i\| \le \lambda N\sqrt{16\alpha(i)\log\frac{4N^2K}{B\delta}} \le \lambda N\sqrt{\frac{16iBM}{N}\log\frac{4N^2K}{B\delta}} = 4\lambda\sqrt{iBMN\log\frac{4N^2K}{B\delta}}. \quad (91)$$

Now consider the case $\alpha(i) = 0$. Recall from the definition $\alpha(i) := \lfloor\frac{iBM}{N}\rfloor$ that $\alpha(i) = 0$ implies $iBM < N$. In this case, $\boldsymbol{q}_{i,l} = \boldsymbol{0}$ for $N - iBM$ indices $l$ satisfying $|\mathcal{J}_l| = 0$, and $\|\boldsymbol{q}_{i,l}\| \le \lambda$ for the remaining $iBM$ $l$'s satisfying $|\mathcal{J}_l| = 1$. Summing up, $\|\boldsymbol{q}_i\|$ is bounded from above by $\lambda iBM$, which is in fact less than the upper bound in (91). Therefore, the bound (91) holds even for $\alpha(i) = 0$.

Recall that our goal was to find a bound on the norm of $\sum_{j=1}^{i}\boldsymbol{g}_j = \frac{1}{M}(\boldsymbol{p}_i + \boldsymbol{q}_i)$. From the high-probability bounds obtained in (88) and (91), with probability at least $1 - \frac{B\delta}{NK}$,

$$\left\|\sum_{j=1}^{i}\boldsymbol{g}_j\right\| \le iB\|\nabla F(\boldsymbol{x}_{k,0})\| + \frac{\nu\sqrt{N}}{M}\sqrt{8\log\frac{4NK}{B\delta}} + 4\lambda\sqrt{\frac{iBN}{M}}\sqrt{\log\frac{4N^2K}{B\delta}}. \quad (92)$$

We can now substitute (83) and (92) to (82) to get

$$
\begin{aligned}
\|\boldsymbol{r}_k\| &\le e^{1/6}LB\sum_{i=1}^{N/B-1}\left(iB\|\nabla F(\boldsymbol{x}_{k,0})\| + \frac{\nu\sqrt{N}}{M}\sqrt{8\log\frac{4NK}{B\delta}} + 4\lambda\sqrt{\frac{iBN}{M}}\sqrt{\log\frac{4N^2K}{B\delta}}\right) \\
&\le e^{1/6}LB^2\|\nabla F(\boldsymbol{x}_{k,0})\|\sum_{i=1}^{N/B-1}i + \frac{e^{1/6}\sqrt{8}L\nu N^{1/2}(N-B)}{M}\sqrt{\log\frac{4NK}{B\delta}} \\
&\quad + \frac{4e^{1/6}L\lambda B^{3/2}N^{1/2}}{M^{1/2}}\int_{1}^{N/B}\sqrt{t}dt\sqrt{\log\frac{4N^2K}{B\delta}} \\
&\le LN(N-B)\|\nabla F(\boldsymbol{x}_{k,0})\| + \frac{7L\nu N^{1/2}(N-B)}{2M}\sqrt{\log\frac{4NK}{B\delta}} \\
&\quad + \frac{7L\lambda N^{1/2}(N^{3/2}-B^{3/2})}{2M^{1/2}}\sqrt{\log\frac{4N^2K}{B\delta}},
\end{aligned}
\quad (93)
$$

which holds with probability at least $1 - \frac{\delta}{K}$, due to the union bound over $i = 1,\ldots,N/B-1$. The bound (93) holds for all $k \in [K]$ with probability $1 - \delta$ if we apply the union bound over $k = 1,\ldots,K$. Next, by $(a+b+c)^2 \le 3a^2 + 3b^2 + 3c^2$, we have

$$
\begin{aligned}
\|\boldsymbol{r}_k\|^2 &\le 3L^2N^2(N-B)^2\|\nabla F(\boldsymbol{x}_{k,0})\|^2 + \frac{147L^2\nu^2N(N-B)^2}{4M^2}\log\frac{4NK}{B\delta} \\
&\quad + \frac{147L^2\lambda^2N(N^{3/2}-B^{3/2})^2}{4M}\log\frac{4N^2K}{B\delta},
\end{aligned}
\quad (94)
$$

which also holds for all $k \in [K]$ with probability at least $1 - \delta$.

**Getting a high-probability convergence rate.** Given our high-probability bounds (93) and (94), we can substitute them to (40) and get

$$
\begin{aligned}
&F(\boldsymbol{x}_{k+1,0}) - F(\boldsymbol{x}_{k,0}) \\
&\le (-\eta N + \eta^2LN^2)\|\nabla F(\boldsymbol{x}_{k,0})\|^2 + \eta^2\|\nabla F(\boldsymbol{x}_{k,0})\|\|\boldsymbol{r}_k\| + \eta^4L\|\boldsymbol{r}_k\|^2 \\
&\le \left(-\eta N + \eta^2LN^2 + \eta^2LN(N-B) + 3\eta^4L^3N^2(N-B)^2\right)\|\nabla F(\boldsymbol{x}_{k,0})\|^2 \\
&\quad + \frac{7\eta^2L\nu N^{1/2}(N-B)}{2M}\sqrt{\log\frac{4NK}{B\delta}}\|\nabla F(\boldsymbol{x}_{k,0})\| + \frac{147\eta^4L^3\nu^2N(N-B)^2}{4M^2}\log\frac{4NK}{B\delta} \\
&\quad + \frac{7\eta^2L\lambda N^{1/2}(N^{3/2}-B^{3/2})}{2M^{1/2}}\sqrt{\log\frac{4N^2K}{B\delta}}\|\nabla F(\boldsymbol{x}_{k,0})\| \\
&\quad + \frac{147\eta^4L^3\lambda^2N(N^{3/2}-B^{3/2})^2}{4M}\log\frac{4N^2K}{B\delta}.
\end{aligned}
\quad (95)
$$

Two terms in the RHS of (95) can be bounded using $ab \leq \frac{a^2}{2} + \frac{b^2}{2}$:

$$\frac{7\eta^2 L\nu N^{1/2}(N-B)}{2M}\sqrt{\log\frac{4NK}{B\delta}}\|\nabla F(\boldsymbol{x}_{k,0})\|$$

$$= \left(\frac{\eta^{1/2}N^{1/2}}{2\sqrt{2}}\|\nabla F(\boldsymbol{x}_{k,0})\|\right)\left(\frac{7\sqrt{2}\eta^{3/2}L\nu(N-B)}{M}\sqrt{\log\frac{4NK}{B\delta}}\right)$$

$$\leq \frac{\eta N}{16}\|\nabla F(\boldsymbol{x}_{k,0})\|^2 + \frac{49\eta^3 L^2\nu^2(N-B)^2}{M^2}\log\frac{4NK}{B\delta}, \tag{96}$$

$$\frac{7\eta^2 L\lambda N^{1/2}(N^{3/2}-B^{3/2})}{2M^{1/2}}\sqrt{\log\frac{4N^2K}{B\delta}}\|\nabla F(\boldsymbol{x}_{k,0})\|$$

$$= \left(\frac{\eta^{1/2}N^{1/2}}{2\sqrt{2}}\|\nabla F(\boldsymbol{x}_{k,0})\|\right)\left(\frac{7\sqrt{2}\eta^{3/2}L\lambda(N^{3/2}-B^{3/2})}{M^{1/2}}\sqrt{\log\frac{4N^2K}{B\delta}}\right)$$

$$\leq \frac{\eta N}{16}\|\nabla F(\boldsymbol{x}_{k,0})\|^2 + \frac{49\eta^3 L^2\lambda^2(N^{3/2}-B^{3/2})^2}{M}\log\frac{4N^2K}{B\delta}, \tag{97}$$

Putting inequalities (96) and (97) to (95) and noting $N - B \leq N$ gives

$$F(\boldsymbol{x}_{k+1,0}) - F(\boldsymbol{x}_{k,0}) \leq \left(-\frac{7}{8}\eta N + 2\eta^2 LN^2 + 3\eta^4 L^3 N^4\right)\|\nabla F(\boldsymbol{x}_{k,0})\|^2$$

$$+ \frac{49\eta^3 L^2(4+3\eta LN)\nu^2(N-B)^2}{4M^2}\log\frac{4NK}{B\delta}$$

$$+ \frac{49\eta^3 L^2(4+3\eta LN)\lambda^2(N^{3/2}-B^{3/2})^2}{4M}\log\frac{4N^2K}{B\delta}. \tag{98}$$

Recall from $K \geq 6\kappa\log(M^2 NK^2)$ and $\eta = \frac{\log(M^2 NK^2)}{\mu NK}$ that $\eta LN \leq \frac{1}{6}$. Since the inequality $-\frac{7}{8}z + 2z^2 + 3z^4 \leq -\frac{1}{2}z$ holds on $z \in [0, \frac{1}{6}]$, we have

$$-\frac{7}{8}\eta N + 2\eta^2 LN^2 + 3\eta^4 L^3 N^4 \leq -\frac{1}{2}\eta N.$$

Applying this bound to (98) results in

$$F(\boldsymbol{x}_{k+1,0}) - F(\boldsymbol{x}_{k,0}) \leq -\frac{\eta N}{2}\|\nabla F(\boldsymbol{x}_{k,0})\|^2 + \frac{56\eta^3 L^2\nu^2(N-B)^2}{M^2}\log\frac{4NK}{B\delta}$$

$$+ \frac{56\eta^3 L^2\lambda^2(N^{3/2}-B^{3/2})^2}{M}\log\frac{4N^2K}{B\delta}$$

We now recall that $F$ is $\mu$-PŁ, so $\|\nabla F(\boldsymbol{x}_{k,0})\|^2 \geq 2\mu(F(\boldsymbol{x}_{k,0}) - F^*)$:

$$F(\boldsymbol{x}_{k+1,0}) - F^* \leq (1-\eta\mu N)(F(\boldsymbol{x}_{k,0}) - F^*) + \frac{56\eta^3 L^2\nu^2(N-B)^2}{M^2}\log\frac{4NK}{B\delta}$$

$$+ \frac{56\eta^3 L^2\lambda^2(N^{3/2}-B^{3/2})^2}{M}\log\frac{4N^2K}{B\delta} \tag{99}$$

Recall that (99) holds for all $k \in [K]$, with probability $1 - \delta$. Therefore, by unrolling the inequality, and using $\sum_{k=0}^{K-1}(1-\eta\mu N)^k \leq \frac{1}{\eta\mu N}$, we get

$$F(\boldsymbol{x}_{K,\frac{N}{B}}) - F^* \leq (1-\eta\mu N)^K(F(\boldsymbol{x}_0) - F^*) + \frac{56\eta^2 L^2\nu^2(N-B)^2}{\mu M^2 N}\log\frac{4NK}{B\delta}$$

$$+ \frac{56\eta^2 L^2\lambda^2(N^{3/2}-B^{3/2})^2}{\mu MN}\log\frac{4N^2K}{B\delta} \tag{100}$$

Lastly, substituting $\eta = \frac{\log(M^2 NK^2)}{\mu NK}$ to (100) gives

$$F(\boldsymbol{x}_{K,\frac{N}{B}}) - F^* \leq \frac{F(\boldsymbol{x}_0) - F^*}{M^2 NK^2} + \frac{56L^2\nu^2(N-B)^2\log\frac{4NK}{B\delta}\log^2(M^2 NK^2)}{\mu^3 M^2 N^3 K^2}$$

$$+ \frac{56L^2\lambda^2(N^{3/2} - B^{3/2})^2 \log \frac{4N^2K}{B\delta} \log^2(M^2NK^2)}{\mu^3 MN^3K^2}$$

$$= \frac{F(\boldsymbol{x}_0) - F^*}{M^2NK^2} + \tilde{\mathcal{O}}\left(\frac{L^2}{\mu^3}\left(\frac{\nu^2}{M^2NK^2} + \frac{\lambda^2}{MK^2}\right)\right).$$

### D.5 PROOF OF UPPER BOUND FOR LOCAL RR WITH SYNCSHUF (THEOREM 7)

The first part ("One epoch as one step of GD plus noise") of the proof is identical to that of Theorem 2. The first part defines our "noise" $\boldsymbol{r}_k$ as the sum of three terms $\boldsymbol{r}_k := \boldsymbol{r}_{k,1} + \boldsymbol{r}_{k,2} - \eta\boldsymbol{r}_{k,3}$. We start from the second part.

**Bounding noise terms using concentration.** We next bound $\|\boldsymbol{r}_k\|$ by bounding each $\|\boldsymbol{r}_{k,1}\|$, $\|\boldsymbol{r}_{k,2}\|$, and $\|\boldsymbol{r}_{k,3}\|$. We have already seen from (63), (64), (65), (66), and (67) in Appendix D.3 that

$$\|\boldsymbol{r}_{k,1}\| \le \frac{e^{1/7}L}{M} \sum_{l=1}^{N/B} \sum_{m=1}^{M} \sum_{i=(l-1)B+1}^{lB-1} \left\| \sum_{t=(l-1)B+1}^{i} \nabla f_{\sigma_k^m(t)}^m(\boldsymbol{y}_{k,0}) \right\|, \tag{101}$$

$$\|\boldsymbol{r}_{k,2}\| \le \frac{7LB}{5M} \sum_{l=1}^{N/B-1} \left\| \sum_{m=1}^{M} \sum_{t=1}^{lB} \nabla f_{\sigma_k^m(t)}^m(\boldsymbol{y}_{k,0}) \right\|, \tag{102}$$

$$\|\boldsymbol{r}_{k,3}\| \le \frac{11L^2B}{7M} \sum_{l=1}^{N/B-1} \sum_{j=1}^{l} \sum_{m=1}^{M} \sum_{i=(j-1)B+1}^{jB-1} \left\| \sum_{t=(j-1)B+1}^{i} \nabla f_{\sigma_k^m(t)}^m(\boldsymbol{y}_{k,0}) \right\|. \tag{103}$$

As in the previous subsections, the key is to bound the norm of the partial sums of $\nabla f_{\sigma_k^m(t)}^m(\boldsymbol{y}_{k,0})$ using Lemma 8. For the summations appearing in (101) and (103), we apply Lemma 8 in the same way as (68). For any $i$ satisfying $(j-1)B + 1 \le i \le jB - 1$, where $j \in [N/B]$, and for any $m \in [M]$, the following bound holds with probability at least $1 - \frac{\delta}{3MNK}$:

$$\left\| \sum_{t=(j-1)B+1}^{i} \nabla f_{\sigma_k^m(t)}^m(\boldsymbol{y}_{k,0}) \right\| \le \nu\sqrt{8(i - (j-1)B)\log\frac{6MNK}{\delta}}$$

$$+ (i - (j-1)B)\|\nabla F^m(\boldsymbol{y}_{k,0})\|. \tag{104}$$

For the summation that appear in (102), we use the techniques from Appendix D.4. For each $l \in [N/B - 1]$, we can similarly decompose $\sum_{m=1}^{M} \sum_{t=1}^{lB} \nabla f_{\sigma_k^m(t)}^m(\boldsymbol{y}_{k,0})$ into $\boldsymbol{p}_l + \boldsymbol{q}_l$, where

$$\boldsymbol{p}_l := \sum_{m=1}^{M} \sum_{t=1}^{lB} \nabla \bar{f}_{\sigma_k^m(t)}^m(\boldsymbol{y}_{k,0}),$$

$$\boldsymbol{q}_l := \sum_{m=1}^{M} \sum_{t=1}^{lB} \nabla f_{\sigma_k^m(t)}^m(\boldsymbol{y}_{k,0}) - \nabla \bar{f}_{\sigma_k^m(t)}^m(\boldsymbol{y}_{k,0}).$$

As done in Appendix D.4, we can follow the same steps and show a high-probability bound, which is a slightly different version of (88): with probability at least $1 - \frac{B\delta}{3NK}$,

$$\|\boldsymbol{p}_l\| \le lBM\|\nabla F(\boldsymbol{y}_{k,0})\| + \nu\sqrt{N}\sqrt{8\log\frac{6NK}{B\delta}}. \tag{105}$$

Similarly, for $\|\boldsymbol{q}_l\|$ we can show a slight modification of (91):

$$\|\boldsymbol{q}_l\| \le 4\lambda\sqrt{lBMN\log\frac{6N^2K}{B\delta}}, \tag{106}$$

which holds with probability at least $1 - \frac{B\delta}{3NK}$. Combining (105) and (106), with probability at least $1 - \frac{2B\delta}{3NK}$, we have

$$\left\| \sum_{m=1}^{M} \sum_{t=1}^{lB} \nabla f_{\sigma_k^m(t)}^m(\boldsymbol{y}_{k,0}) \right\| \le lBM\|\nabla F(\boldsymbol{y}_{k,0})\| + \nu\sqrt{N}\sqrt{8\log\frac{6NK}{B\delta}}$$

$$+ 4\lambda\sqrt{lBMN \log \frac{6N^2K}{B\delta}}. \tag{107}$$

By applying the union bound, with probability at least $1 - \frac{\delta}{K}$, the bound (104) holds for all $m \in [M]$ and $i \in \bigcup_{j=1}^{N/B}[(j-1)B + 1 : jB - 1]$, and the bound (107) holds for all $l \in [N/B - 1]$.

We now substitute the bounds (104) and (107) to (101), (102), and (103) to get upper bounds for $\|\boldsymbol{r}_{k,1}\|$, $\|\boldsymbol{r}_{k,2}\|$, and $\|\boldsymbol{r}_{k,3}\|$, respectively. For $\|\boldsymbol{r}_{k,1}\|$ and $\|\boldsymbol{r}_{k,3}\|$, we can apply the same calculations as in (70) and (72), modulo the fact that Assumption 3 is now implied by Assumption 4, with constants $\tau = \lambda$ and $\rho = 1$. We obtain

$$\|\boldsymbol{r}_{k,1}\| \leq \frac{24L\nu N(B^{3/2} - 1)}{11B}\sqrt{\log \frac{6MNK}{\delta}} + \frac{3LN(B-1)}{5}(\lambda + \|\nabla F(\boldsymbol{y}_{k,0})\|), \tag{108}$$

$$\|\boldsymbol{r}_{k,3}\| \leq \frac{3L^2\nu N(N-B)(B^{3/2}-1)}{2B}\sqrt{\log \frac{6MNK}{\delta}}$$
$$+ \frac{2L^2N(N-B)(B-1)}{5}(\lambda + \|\nabla F(\boldsymbol{y}_{k,0})\|). \tag{109}$$

For $\|\boldsymbol{r}_{k,2}\|$, we have

$$\|\boldsymbol{r}_{k,2}\| \leq \frac{7LB}{5M}\sum_{l=1}^{N/B-1}\left(lBM\|\nabla F(\boldsymbol{y}_{k,0})\| + \nu\sqrt{N}\sqrt{8\log \frac{6NK}{B\delta}} + 4\lambda\sqrt{lBMN\log \frac{6N^2K}{B\delta}}\right)$$

$$= \frac{7LB^2}{5}\left(\sum_{l=1}^{N/B-1}l\right)\|\nabla F(\boldsymbol{y}_{k,0})\| + \frac{7\sqrt{8}L\nu N^{1/2}(N-B)}{5M}\sqrt{\log \frac{6NK}{B\delta}}$$

$$+ \frac{28L\lambda B^{3/2}N^{1/2}}{5M^{1/2}}\left(\sum_{l=1}^{N/B-1}\sqrt{l}\right)\sqrt{\log \frac{6N^2K}{B\delta}}$$

$$\leq \frac{7LN(N-B)}{10}\|\nabla F(\boldsymbol{y}_{k,0})\| + \frac{4L\nu N^{1/2}(N-B)}{M}\sqrt{\log \frac{6NK}{B\delta}}$$

$$+ \frac{15L\lambda N^{1/2}(N^{3/2} - B^{3/2})}{4M^{1/2}}\sqrt{\log \frac{6N^2K}{B\delta}}. \tag{110}$$

Recalling the definition $\boldsymbol{r}_k := \boldsymbol{r}_{k,1} + \boldsymbol{r}_{k,2} - \eta\boldsymbol{r}_{k,3}$, we get an upper bound for $\|\boldsymbol{r}_k\|$ from (108), (109), and (110):

$$\|\boldsymbol{r}_k\| \leq \|\boldsymbol{r}_{k,1}\| + \|\boldsymbol{r}_{k,2}\| + \eta\|\boldsymbol{r}_{k,3}\|$$

$$\leq L\nu\sqrt{\log \frac{6MNK}{\delta}}\left(\frac{24N(B^{3/2}-1)}{11B} + \frac{4N^{1/2}(N-B)}{M} + \frac{3\eta LN(N-B)(B^{3/2}-1)}{2B}\right)$$

$$+ L\lambda\left(\frac{3N(B-1)}{5} + \frac{15N^{1/2}(N^{3/2}-B^{3/2})}{4M^{1/2}}\sqrt{\log \frac{6N^2K}{B\delta}} + \frac{2\eta LN(N-B)(B-1)}{5}\right)$$

$$+ L\|\nabla F(\boldsymbol{y}_{k,0})\|\left(\frac{3N(B-1)}{5} + \frac{7N(N-B)}{10} + \frac{2\eta LN(N-B)(B-1)}{5}\right). \tag{111}$$

Recall again that we have $K \geq 7\kappa\log(M^2NK^2)$ and $\eta = \frac{\log(M^2NK^2)}{\mu NK}$, so $\eta LN \leq 1/7$. Using this and $N - B \leq N$, we can further simplify (111).

$$\|\boldsymbol{r}_k\| \leq L\nu\sqrt{\log \frac{6MNK}{\delta}}\underbrace{\left(\frac{12N(B^{3/2}-1)}{5B} + \frac{4N^{1/2}(N-B)}{M}\right)}_{=:\Phi_\nu}$$

$$+ L\lambda\sqrt{\log \frac{6N^2K}{B\delta}}\underbrace{\left(\frac{2N(B-1)}{3} + \frac{15N^{1/2}(N^{3/2}-B^{3/2})}{4M^{1/2}}\right)}_{=:\Phi_\lambda}$$

$$+ L \left\| \nabla F(\boldsymbol{y}_{k,0}) \right\| \left( \frac{2N(B-1)}{3} + \frac{7N(N-B)}{10} \right)$$

$$\leq L\nu\Phi_\nu \sqrt{\log \frac{6MNK}{\delta}} + L\lambda\Phi_\lambda \sqrt{\log \frac{6N^2K}{B\delta}} + \frac{7LN^2}{5} \left\| \nabla F(\boldsymbol{y}_{k,0}) \right\|, \qquad (112)$$

which holds with probability at least $1 - \frac{\delta}{K}$. The bound (112) holds for all $k \in [K]$ with probability $1 - \delta$ if we apply the union bound over $k = 1, \ldots, K$. Next, by $(a + b + c)^2 \leq 3a^2 + 3b^2 + 3c^2$, we have

$$\left\| \boldsymbol{r}_k \right\|^2 \leq 3L^2\nu^2\Phi_\nu^2 \log \frac{6MNK}{\delta} + 3L^2\lambda^2\Phi_\lambda^2 \log \frac{6N^2K}{B\delta} + \frac{147L^2N^4}{25} \left\| \nabla F(\boldsymbol{y}_{k,0}) \right\|^2, \qquad (113)$$

which also holds for all $k \in [K]$ with probability at least $1 - \delta$.

**Getting a high-probability convergence rate.** Given our high-probability bounds (112) and (113), we can substitute them to (62) and get

$$F(\boldsymbol{y}_{k+1,0}) - F(\boldsymbol{y}_{k,0})$$
$$\leq (-\eta N + \eta^2 LN^2) \left\| \nabla F(\boldsymbol{y}_{k,0}) \right\|^2 + \eta^2 \left\| \nabla F(\boldsymbol{y}_{k,0}) \right\| \left\| \boldsymbol{r}_k \right\| + \eta^4 L \left\| \boldsymbol{r}_k \right\|^2$$
$$\leq \left( -\eta N + \eta^2 LN^2 + \frac{7\eta^2 LN^2}{5} + \frac{147\eta^4 L^3 N^4}{25} \right) \left\| \nabla F(\boldsymbol{y}_{k,0}) \right\|^2$$
$$+ \eta^2 L\nu\Phi_\nu \sqrt{\log \frac{6MNK}{\delta}} \left\| \nabla F(\boldsymbol{y}_{k,0}) \right\| + 3\eta^4 L^3 \nu^2 \Phi_\nu^2 \log \frac{6MNK}{\delta}$$
$$+ \eta^2 L\lambda\Phi_\lambda \sqrt{\log \frac{6N^2K}{B\delta}} \left\| \nabla F(\boldsymbol{y}_{k,0}) \right\| + 3\eta^4 L^3 \lambda^2 \Phi_\lambda^2 \log \frac{6N^2K}{B\delta}. \qquad (114)$$

The following terms in the RHS of (114) can be bounded using $ab \leq \frac{a^2}{2} + \frac{b^2}{2}$:

$$\eta^2 L\nu\Phi_\nu \sqrt{\log \frac{6MNK}{\delta}} \left\| \nabla F(\boldsymbol{y}_{k,0}) \right\|$$
$$= \left( \frac{\eta^{1/2} N^{1/2}}{\sqrt{8}} \left\| \nabla F(\boldsymbol{y}_{k,0}) \right\| \right) \left( \frac{\sqrt{8}\eta^{3/2} L\nu\Phi_\nu}{N^{1/2}} \sqrt{\log \frac{6MNK}{\delta}} \right)$$
$$\leq \frac{\eta N}{16} \left\| \nabla F(\boldsymbol{y}_{k,0}) \right\|^2 + \frac{4\eta^3 L^2 \nu^2 \Phi_\nu^2}{N} \log \frac{6MNK}{\delta}, \qquad (115)$$

$$\eta^2 L\lambda\Phi_\lambda \sqrt{\log \frac{6N^2K}{B\delta}} \left\| \nabla F(\boldsymbol{y}_{k,0}) \right\|$$
$$= \left( \frac{\eta^{1/2} N^{1/2}}{\sqrt{8}} \left\| \nabla F(\boldsymbol{y}_{k,0}) \right\| \right) \left( \frac{\sqrt{8}\eta^{3/2} L\lambda\Phi_\lambda}{N^{1/2}} \sqrt{\log \frac{6N^2K}{B\delta}} \right)$$
$$\leq \frac{\eta N}{16} \left\| \nabla F(\boldsymbol{y}_{k,0}) \right\|^2 + \frac{4\eta^3 L^2 \lambda^2 \Phi_\lambda^2}{N} \log \frac{6N^2K}{B\delta}. \qquad (116)$$

Substituting (115) and (116) to (114) results in

$$F(\boldsymbol{y}_{k+1,0}) - F(\boldsymbol{y}_{k,0})$$
$$\leq \left( -\frac{7}{8}\eta N + \frac{12\eta^2 LN^2}{5} + \frac{147\eta^4 L^3 \rho^2 N^4}{25} \right) \left\| \nabla F(\boldsymbol{y}_{k,0}) \right\|^2$$
$$+ \frac{\eta^3 L^2 \nu^2 (4 + 3\eta LN) \Phi_\nu^2}{N} \log \frac{6MNK}{\delta} + \frac{\eta^3 L^2 \lambda^2 (4 + 3\eta LN) \Phi_\lambda^2}{N} \log \frac{6N^2K}{B\delta}. \qquad (117)$$

Again, we have $K \geq 7\kappa \log(M^2 N K^2)$ and $\eta = \frac{\log(M^2 N K^2)}{\mu N K}$, so $\eta LN \leq \frac{1}{7}$. Since the inequality $-\frac{7}{8}z + \frac{12}{5}z^2 + \frac{147}{25}z^4 \leq -\frac{1}{2}z$ holds on $z \in [0, \frac{1}{7}]$, we have

$$-\frac{7}{8}\eta N + \frac{12\eta^2 L\rho N^2}{5} + \frac{147\eta^4 L^3 \rho^3 N^4}{25} \leq -\frac{1}{2}\eta N.$$

Substituting this inequality to (117), together with $4 + 3\eta LN \leq \frac{31}{7} < \frac{9}{2}$, yields

$$F(\boldsymbol{y}_{k+1,0}) - F(\boldsymbol{y}_{k,0}) \leq -\frac{\eta N}{2}\|\nabla F(\boldsymbol{y}_{k,0})\|^2 + \frac{9\eta^3 L^2 \nu^2 \Phi_\nu^2}{2N}\log\frac{6MNK}{\delta}$$
$$+ \frac{9\eta^3 L^2 \lambda^2 \Phi_\lambda^2}{2N}\log\frac{6N^2K}{B\delta}.$$

We now recall that $F$ is $\mu$-PŁ, so $\|\nabla F(\boldsymbol{y}_{k,0})\|^2 \geq 2\mu(F(\boldsymbol{y}_{k,0}) - F^*)$:

$$F(\boldsymbol{y}_{k+1,0}) - F^* \leq (1 - \eta\mu N)(F(\boldsymbol{y}_{k,0}) - F^*) + \frac{9\eta^3 L^2 \nu^2 \Phi_\nu^2}{2N}\log\frac{6MNK}{\delta}$$
$$+ \frac{9\eta^3 L^2 \lambda^2 \Phi_\lambda^2}{2N}\log\frac{6N^2K}{B\delta}. \tag{118}$$

Recall that (118) holds for all $k \in [K]$, with probability $1 - \delta$. Therefore, by unrolling the inequality, and using $\sum_{k=0}^{K-1}(1 - \eta\mu N)^k \leq \frac{1}{\eta\mu N}$, we get

$$F(\boldsymbol{y}_{K,\frac{N}{B}}) - F^* \leq (1 - \eta\mu N)^K(F(\boldsymbol{y}_0) - F^*) + \frac{9\eta^2 L^2 \nu^2 \Phi_\nu^2}{2\mu N^2}\log\frac{6MNK}{\delta}$$
$$+ \frac{9\eta^2 L^2 \lambda^2 \Phi_\lambda^2}{2\mu N^2}\log\frac{6N^2K}{B\delta}. \tag{119}$$

Recall that $\Phi_\nu := \frac{12N(B^{3/2}-1)}{5B} + \frac{4N^{1/2}(N-B)}{M}$ and $\Phi_\lambda := \frac{2N(B-1)}{3} + \frac{15N^{1/2}(N^{3/2}-B^{3/2})}{4M^{1/2}}$, hence

$$\Phi_\nu^2 \leq \frac{288N^2(B^{3/2}-1)^2}{25B^2} + \frac{32N(N-B)^2}{M^2},$$
$$\Phi_\lambda^2 \leq \frac{8N^2(B-1)^2}{9} + \frac{225N(N^{3/2}-B^{3/2})^2}{2M}.$$

Substituting these inequalities and also $\eta = \frac{\log(M^2NK^2)}{\mu NK}$ gives

$$F(\boldsymbol{y}_{K,\frac{N}{B}}) - F^*$$
$$\leq \frac{F(\boldsymbol{y}_0) - F^*}{M^2NK^2} + \frac{9L^2\nu^2 \log\frac{6MNK}{\delta}\log^2(M^2NK^2)}{2\mu^3 N^4 K^2}\left(\frac{288N^2(B^{3/2}-1)^2}{25B^2} + \frac{32N(N-B)^2}{M^2}\right)$$
$$+ \frac{9L^2\lambda^2 \log\frac{6N^2K}{B\delta}\log^2(M^2NK^2)}{2\mu^3 N^4 K^2}\left(\frac{8N^2(B-1)^2}{9} + \frac{225N(N^{3/2}-B^{3/2})^2}{2M}\right)$$
$$= \frac{F(\boldsymbol{y}_0) - F^*}{M^2NK^2} + \tilde{\mathcal{O}}\left(\frac{L^2}{\mu^3}\left(\frac{\nu^2 B}{N^2 K^2} + \frac{\nu^2}{M^2 NK^2} + \frac{\lambda^2 B^2}{N^2 K^2} + \frac{\lambda^2}{MK^2}\right)\right),$$

with probability at least $1 - \delta$. This finishes the proof.

### D.6 A GENERALIZED VECTOR-VALUED HOEFFDING-SERFLING INEQUALITY

We extend the vector-valued Hoeffding-Serfling inequality proved in Schneider (2016) to account for the mean of multiple independent without-replacement sums.

**Lemma 8.** *Suppose there are $MN$ vectors $\{\boldsymbol{v}_i^m\}_{m=1,i=1}^{M,N} \in \mathbb{R}^d$ that satisfy $\|\boldsymbol{v}_i^m - \bar{\boldsymbol{v}}^m\| \leq \nu$ for $m \in [M]$, where $\bar{\boldsymbol{v}}^m := \frac{1}{N}\sum_{i=1}^{N}\boldsymbol{v}_i^m$. Consider $M$ independently and uniformly sampled permutations $\sigma_1, \ldots, \sigma_M \sim \text{Unif}(\mathcal{S}_N)$. For any $n \leq N-1$, with probability at least $1 - \delta$, we have*

$$\left\|\frac{1}{Mn}\sum_{m=1}^{M}\sum_{i=1}^{n}\boldsymbol{v}_{\sigma_m(i)}^m - \frac{1}{M}\sum_{m=1}^{M}\bar{\boldsymbol{v}}^m\right\| \leq \nu\sqrt{\frac{8(1 - \frac{n-1}{N})\log\frac{2}{\delta}}{Mn}}. \tag{120}$$

*Proof.* The proof is an extension of Theorem 2 of Schneider (2016) which proves the $M = 1$ case for vectors in smooth separable Banach spaces. We prove our extended concentration inequality for $\mathbb{R}^d$, but we note that the proof technique can be applied directly to general smooth separable Banach spaces, as done in Schneider (2016). Below, we state a special case of Theorem 3 of Pinelis (1992) and Theorem 3.5 of Pinelis (1994), because this $\mathbb{R}^d$ case serves our purpose.

**Lemma 9** (Pinelis (1992; 1994))**.** *Suppose that a sequence of random variables $\{\mathbf{x}_j\}_{j \geq 0}$ is a martingale taking values in $\mathbb{R}^d$, and $\sum_{j=1}^{\infty} \operatorname{ess\,sup} \|\mathbf{x}_j - \mathbf{x}_{j-1}\|^2 \leq c^2$ for some $c > 0$. Then, for $\lambda > 0$,*

$$\mathbb{P}\big(\sup\{\|\mathbf{x}_j\| : j \geq 0\} \geq \lambda\big) \leq 2\exp\left(-\frac{\lambda^2}{2c^2}\right).$$

The proof of Lemma 8 proceeds by defining a sequence of random variables $\{\mathbf{x}_j\}$, showing that it is a martingale, and applying Lemma 9 to prove our concentration bound. For $m \in [M]$, define index functions $k_m : \mathbb{N} \cup \{0\} \to [0:n]$ in the following way:

$$k_m(j) := \max\{0, \min\{n, j - (m-1)n\}\} = \begin{cases} 0 & \text{if } j \leq (m-1)n, \\ j - (m-1)n & \text{if } (m-1)n + 1 \leq j \leq mn, \\ n & \text{if } j \geq mn + 1. \end{cases}$$

Using these index functions, we introduce the following sequence of random variables $\{\mathbf{x}_j\}$:

$$\mathbf{x}_j := \sum_{m=1}^{M} \frac{1}{N - k_m(j)} \sum_{i=1}^{k_m(j)} (\boldsymbol{v}_{\sigma_m(i)}^m - \bar{\boldsymbol{v}}^m),$$

and show that this is a martingale, i.e.,

$$\mathbb{E}[\mathbf{x}_j \mid \mathbf{x}_1, \ldots, \mathbf{x}_{j-1}] = \mathbf{x}_{j-1} \tag{121}$$

for all $j \geq 1$. Notice first that by definition of $k_m$'s we have $\mathbf{x}_{Mn} = \mathbf{x}_{Mn+1} = \mathbf{x}_{Mn+2} = \ldots$, so (121) is trivially satisfied for all $j > Mn$. Next, for any $j$ satisfying $(l-1)n + 1 \leq j \leq ln$ where $l \in [M]$, we have

$$\mathbf{x}_j = \sum_{m=1}^{l-1} \frac{1}{N-n} \sum_{i=1}^{n} (\boldsymbol{v}_{\sigma_m(i)}^m - \bar{\boldsymbol{v}}^m) + \frac{1}{N - k_l(j)} \sum_{i=1}^{k_l(j)} (\boldsymbol{v}_{\sigma_l(i)}^l - \bar{\boldsymbol{v}}^l)$$

$$= \mathbf{x}_{j-1} + \left(\frac{1}{N - k_l(j)} - \frac{1}{N - k_l(j) + 1}\right) \sum_{i=1}^{k_l(j)-1} (\boldsymbol{v}_{\sigma_l(i)}^l - \bar{\boldsymbol{v}}^l)$$

$$+ \frac{1}{N - k_l(j)} (\boldsymbol{v}_{\sigma_l(k_l(j))}^l - \bar{\boldsymbol{v}}^l) \tag{122}$$

Now note that for any $k \in [N-1]$, we have

$$\mathbb{E}[\boldsymbol{v}_{\sigma_l(k)}^l - \bar{\boldsymbol{v}}^l \mid \sigma_l(1), \ldots, \sigma_l(k-1)] = \frac{1}{N - k + 1} \sum_{i=k}^{N} (\boldsymbol{v}_{\sigma_l(i)}^l - \bar{\boldsymbol{v}}^l)$$

$$= -\frac{1}{N - k + 1} \sum_{i=1}^{k-1} (\boldsymbol{v}_{\sigma_l(i)}^l - \bar{\boldsymbol{v}}^l),$$

where the last equality used $\sum_{i=1}^{N} (\boldsymbol{v}_{\sigma_l(i)}^l - \bar{\boldsymbol{v}}^l) = \mathbf{0}$. Using applying this fact to (122) and noting $\frac{1}{N - k_l(j)} - \frac{1}{N - k_l(j) + 1} = \frac{1}{(N - k_l(j))(N - k_l(j) + 1)}$,

$$\mathbb{E}[\mathbf{x}_j \mid \mathbf{x}_1, \ldots, \mathbf{x}_{j-1}] = \mathbf{x}_{j-1} + \frac{1}{(N - k_l(j))(N - k_l(j) + 1)} \sum_{i=1}^{k_l(j)-1} (\boldsymbol{v}_{\sigma_l(i)}^l - \bar{\boldsymbol{v}}^l)$$

$$+ \frac{1}{N - k_l(j)} \mathbb{E}[\boldsymbol{v}_{\sigma_l(k_l(j))}^l - \bar{\boldsymbol{v}}^l \mid \mathbf{x}_1, \ldots, \mathbf{x}_{j-1}]$$

$$= \mathbf{x}_{j-1},$$

hence proving that $\{\mathbf{x}_j\}_{j \geq 0}$ is a martingale. We now apply Lemma 9 to our $\{\mathbf{x}_j\}$. For $j$ such that $(l-1)n + 1 \leq j \leq ln$, notice from (122) that

$$(N - k_l(j))(\mathbf{x}_j - \mathbf{x}_{j-1}) = \frac{1}{(N - k_l(j) + 1)} \left[\sum_{i=1}^{k_l(j)-1} (\boldsymbol{v}_{\sigma_l(i)}^l - \bar{\boldsymbol{v}}^l)\right] + (\boldsymbol{v}_{\sigma_l(k_l(j))}^l - \bar{\boldsymbol{v}}^l)$$

$$= -\frac{1}{(N - k_l(j) + 1)} \left[ \sum_{i=k_l(j)}^{N} (\boldsymbol{v}^l_{\sigma_l(i)} - \bar{\boldsymbol{v}}^l) \right] + (\boldsymbol{v}^l_{\sigma_l(k_l(j))} - \bar{\boldsymbol{v}}^l),$$

which leads to

$$(N - k_l(j)) \|\mathbf{x}_j - \mathbf{x}_{j-1}\| \leq \frac{\nu \min\{k_l(j) - 1, N - k_l(j) + 1\}}{N - k_l(j) + 1} + \nu \leq 2\nu,$$

by the triangle inequality. From this, we get the bound $c^2$ in the statement of Lemma 9:

$$\sum_{j=1}^{\infty} \operatorname{ess\,sup} \|\mathbf{x}_j - \mathbf{x}_{j-1}\|^2 = \sum_{j=1}^{Mn} \operatorname{ess\,sup} \|\mathbf{x}_j - \mathbf{x}_{j-1}\|^2 \leq M \sum_{k=1}^{n} \frac{4\nu^2}{(N - k)^2}$$

$$= \frac{4\nu^2 M}{(N - n)^2} + 4\nu^2 M \sum_{k=N-n+1}^{N-1} \frac{1}{k^2} \leq \frac{4\nu^2 M}{(N - n)^2} + \frac{4\nu^2 M(n - 1)}{(N - n)N} = \frac{4\nu^2 Mn}{(N - n)^2} \left( 1 - \frac{n - 1}{N} \right),$$

where the second inequality used the inequality that $\sum_{k=a+1}^{b} \frac{1}{k^2} \leq \frac{b-a}{a(b+1)}$ (Serfling, 1974, Lemma 2.1). Now, applying Lemma 9 to $\{\mathbf{x}_j\}$ with $c^2 = \frac{4\nu^2 Mn}{(N-n)^2} \left( 1 - \frac{n-1}{N} \right)$ gives

$$\mathbb{P}\big( \|\mathbf{x}_{Mn}\| \geq \lambda \big) \leq \mathbb{P}\big( \sup\{\|\mathbf{x}_j\| : j \geq 0\} \geq \lambda \big) \leq 2\exp\left( -\frac{\lambda^2 (N - n)^2}{8\nu^2 Mn \left( 1 - \frac{n-1}{N} \right)} \right). \quad (123)$$

Recall from the definition of $\{\mathbf{x}_j\}$ that

$$\mathbf{x}_{Mn} = \frac{1}{N - n} \sum_{m=1}^{M} \sum_{i=1}^{n} (\boldsymbol{v}^m_{\sigma_m(i)} - \bar{\boldsymbol{v}}^m).$$

Substituting $\lambda = \frac{Mn\epsilon}{N-n}$ to (123) gives

$$\mathbb{P}\left( \left\| \frac{1}{Mn} \sum_{m=1}^{M} \sum_{i=1}^{n} (\boldsymbol{v}^m_{\sigma_m(i)} - \bar{\boldsymbol{v}}^m) \right\| \geq \epsilon \right) \leq 2\exp\left( -\frac{Mn\epsilon^2}{8\nu^2 \left( 1 - \frac{n-1}{N} \right)} \right),$$

which finishes the proof. $\qquad\square$

### D.7 HOW CAN WE AVOID UNIFORM BOUNDS OVER $\mathbb{R}^d$ IN OUR ASSUMPTIONS?

In Section 2, we introduced Assumptions 2, 3, and 4 on the intra- and inter-machine deviation. The assumptions required that inequalities such as $\|\nabla f^m_i(\boldsymbol{x}) - \nabla F^m(\boldsymbol{x})\| \leq \nu$ hold for all $\boldsymbol{x} \in \mathbb{R}^d$. In this subsection, we discuss more on this strong requirement "entire $\mathbb{R}^d$."

In fact, the entire-$\mathbb{R}^d$ requirement is posed in our assumptions to simplify the exposition of the main results, and is not strictly necessary. One can easily check from our proofs that the assumptions are only applied to the beginning iterates $\boldsymbol{x}_{k,0}$ (for minibatch RR) or $\boldsymbol{y}_{k,0}$ (for local RR) of epochs. Hence, if these iterates lie in a bounded set, then the constants $\nu$, $\tau$, $\rho$, and $\lambda$ may become much smaller, depending on problem instances. Actually, if we explicitly assume that the iterates lie in a compact set $\mathbb{S}$,[12] then Assumptions 2–4 are even *guaranteed* to hold for some constants; e.g., for Assumption 2, we can choose

$$\nu := \max_{m \in [M], i \in [N], \boldsymbol{x} \in \mathbb{S}} \|\nabla f^m_i(\boldsymbol{x}) - \nabla F^m(\boldsymbol{x})\|,$$

since the maximum always exists.

However, assuming that the iterates lie in a specific set $\mathbb{S}$ can be problematic because the distance that the iterates travel depend on the objective functions. One cannot know a priori if all iterates will stay in a fixed set $\mathbb{S}$; hence, explicitly assuming bounded iterates should be avoided.

Then, a natural question is whether we can *prove* bounded iterates under some reasonable conditions, instead of assuming it. We point out that this can be done by applying the technique developed in Ahn et al. (2020, Theorem 1) to our upper bound theorems. Using the technique, a modified version of our Theorem 1 can be written as follows:

---

[12] This *bounded iterates assumption* is indeed used in some existing results such as Haochen & Sra (2019); Nagaraj et al. (2019); Rajput et al. (2020); Ahn et al. (2020).

**Theorem 10** (Best-iterate version of Theorem 1). *Suppose that minibatch* RR *has parameters satisfying Assumption 1. Assume that all local component functions $f_i^m$ are L-smooth, the global objective function F is $\mu$-PŁ, and the set of global minima of F is nonempty and compact. Consider running the algorithm using step-size $\eta = \frac{B \log(MNK^2)}{\mu NK}$ and initialization $\boldsymbol{x}_{1,0} := \boldsymbol{x}_0$, for epochs $K \geq 6\kappa \log(MNK^2)$. Then, with probability at least $1 - \delta$,*

$$\min_{k \in [K+1]} F(\boldsymbol{x}_{k,0}) - F^* \leq \frac{F(\boldsymbol{x}_0) - F^*}{MNK^2} + \tilde{\mathcal{O}}\left(\frac{L^2}{\mu^3} \frac{\nu^2}{MNK^2}\right),$$

*where the constant $\nu < \infty$ is defined as*

$$\nu := \sup_{\boldsymbol{x}: F(\boldsymbol{x}) \leq F(\boldsymbol{x}_0)} \max_{i \in [N]} \max_{m \in [M]} \|\nabla f_i^m(\boldsymbol{x}) - \nabla F^m(\boldsymbol{x})\|. \tag{124}$$

In the theorem, we used $\boldsymbol{x}_{K+1,0}$ to denote the last iterate of the algorithm $\boldsymbol{x}_{K,\frac{N}{B}}$. Theorem 10 differs from Theorem 1 in three aspects: 1) it considers the *best-iterate*, not the *last-iterate*; 2) it additionally assumes that the set of global minima of F is nonempty and compact, which always holds if F is strongly convex; and 3) it does not rely on Assumption 2, but instead "proves" it for the $F(\boldsymbol{x}_0)$-sublevel set of F (124). Note that the constant $\nu$ (124) can be much smaller than the uniform bound required to make Assumption 2 hold for the entire $\mathbb{R}^d$. For Theorems 2, 6, and 7, we can also apply similar techniques to prove best-iterate bounds with smaller intra- and inter-machine deviation constants $\nu$, $\tau$, $\rho$, and $\lambda$; we omit the precise statements.

We conclude this subsection with the proof of Theorem 10.

*Proof.* The proof follows that of Ahn et al. (2020, Theorem 1).

**Existence of $\nu$.** We first show the existence of $\nu < \infty$ (124). The global objective function F is $\mu$-PŁ. If we denote the set of global minima of F as $\mathbb{X}^*$, the set $\mathbb{X}^*$ is nonempty and compact by assumption. Then, by Karimi et al. (2016, Theorem 2) $\mu$-PŁ functions satisfy quadratic growth, i.e., denoting by $\boldsymbol{x}^*$ the closest global minimum in $\mathbb{X}^*$ to the point $\boldsymbol{x}$,

$$F(\boldsymbol{x}) - F^* \geq 2\mu \|\boldsymbol{x} - \boldsymbol{x}^*\|^2.$$

Define the sublevel set $\mathbb{S} := \{\boldsymbol{x} \mid F(\boldsymbol{x}) \leq F(\boldsymbol{x}_0)\}$. Due to the quadratic growth property, we have $F(\boldsymbol{x}_0) - F^* \geq F(\boldsymbol{x}) - F^* \geq 2\mu \|\boldsymbol{x} - \boldsymbol{x}^*\|^2$ for all $\boldsymbol{x} \in \mathbb{S}$. This implies that

$$\mathbb{S} := \{\boldsymbol{x} \in \mathbb{R}^d \mid F(\boldsymbol{x}) \leq F(\boldsymbol{x}_0)\} \subset \left\{\boldsymbol{x} \in \mathbb{R}^d \mid \|\boldsymbol{x} - \boldsymbol{x}^*\|^2 \leq \frac{F(\boldsymbol{x}_0) - F^*}{2\mu}\right\}.$$

Since we assumed that $\mathbb{X}^*$ is compact, $\mathbb{S}$ is also bounded, and hence compact. Now, for any $m \in [M]$ and $i \in [N]$, $\|\nabla f_i^m(\boldsymbol{x}) - \nabla F^m(\boldsymbol{x})\|$ is a continuous function on a compact set $\mathbb{S}$, so there must exist a constant $\nu_i^m < \infty$ such that $\|\nabla f_i^m(\boldsymbol{x}) - \nabla F^m(\boldsymbol{x})\| \leq \nu_i^m$ for all $\boldsymbol{x} \in \mathbb{S}$. Taking the maximum of $\nu_i^m$ over all $m$ and $i$ gives $\nu$.

**Proving the best-iterate bound.** With the constant $\nu$ (124), if all the iterates $\{\boldsymbol{x}_{k,0}\}_{k \in [K+1]}$ stay within the sublevel set $\mathbb{S} := \{\boldsymbol{x} \mid F(\boldsymbol{x}) \leq F(\boldsymbol{x}_0)\}$, one can consider Assumption 2 to be true with constant $\nu$. From this observation, we consider two cases:

1. All the iterates $\{\boldsymbol{x}_{k,0}\}_{k \in [K+1]}$ stay in the sublevel set $\mathbb{S}$.

2. There exists an iterate $\boldsymbol{x}_{k,0} \notin \mathbb{S}$.

In fact, the first case can be proven by exactly the same steps as Theorem 1, described in Appendix D.2.

For the second case, suppose that there exists an iterate $\boldsymbol{x}_{k,0}$ that escapes the sublevel set $\mathbb{S}$. Let $k' \in \{2, \ldots, K+1\}$ be the first such $k$. Then, since $\boldsymbol{x}_{k'-1,0}$ is still in $\mathbb{S}$, it follows from (50) in Appendix D.2 that we have

$$F(\boldsymbol{x}_{k',0}) - F^* \leq (1 - \eta\mu N)(F(\boldsymbol{x}_{k'-1,0}) - F^*) + \frac{15\eta^3 L^2 \nu^2 (N^{3/2} - B^{3/2})^2}{MN} \log \frac{2NK}{B\delta}. \tag{125}$$

However, the fact that $\boldsymbol{x}_{k',0} \notin \mathbb{S}$ and $\boldsymbol{x}_{k'-1,0} \in \mathbb{S}$ implies

$$F(\boldsymbol{x}_{k',0}) > F(\boldsymbol{x}_0) \geq F(\boldsymbol{x}_{k'-1,0}). \tag{126}$$

Combining the two bounds (125) and (126), we get

$$0 < -\eta\mu N(F(\boldsymbol{x}_{k'-1,0}) - F^*) + \frac{15\eta^3 L^2 \nu^2 (N^{3/2} - B^{3/2})^2}{MN} \log \frac{2NK}{B\delta},$$

which implies

$$\min_{k \in [K+1]} F(\boldsymbol{x}_{k,0}) - F^* \leq F(\boldsymbol{x}_{k'-1,0}) - F^* < \frac{15\eta^2 L^2 \nu^2 (N^{3/2} - B^{3/2})^2}{\mu M N^2} \log \frac{2NK}{B\delta}.$$

Substituting $\eta = \frac{\log(MNK^2)}{\mu NK}$[13] gives the desired bound and finishes the proof. $\qquad\square$

## E  PROOF OF LOWER BOUND FOR MINIBATCH RR (THEOREM 3)

For Theorem 3, we consider three step-size ranges and do case analysis for each of them. We construct functions for each corresponding step-size regime such that the convergence of minibatch RR is "slow" for the functions on their corresponding step-size regime. The final lower bound is the minimum among the lower bounds obtained for the three regimes. More concretely, we will construct three one-dimensional functions $F_1(x)$, $F_2(x)$, and $F_3(x)$ satisfying $L$-smoothness (1), $\mu$-PŁ condition (2), and Assumption 2 such that[14]

- Minibatch RR on $F_1(x)$ with $\eta \leq \frac{B}{\mu NK}$ and initialization $x_0 = \frac{\nu}{\mu}$ results in

$$\mathbb{E}[F_1(x_{K,\frac{N}{B}})] = \Omega\left(\frac{\nu^2}{\mu}\right).$$

- Minibatch RR on $F_2(x)$ with $\eta \geq \frac{B}{\mu NK}$ and $\eta \leq \frac{B}{513LN}$ and initialization $x_0 = 0$ results in

$$\mathbb{E}[F_2(x_{K,\frac{N}{B}})] = \Omega\left(\frac{\nu^2}{\mu MNK^2}\right).$$

  Note that the step-size range requires $K \geq 513\kappa$, hence this lower bound occurs only in the "large-epoch" regime, i.e., $K \gtrsim \kappa$.

- Minibatch RR on $F_3(x)$ with $\eta \geq \frac{B}{\mu NK}$ and $\eta \geq \frac{B}{513LN}$ and initialization $x_0 = 0$ results in

$$\mathbb{E}[F_3(x_{K,\frac{N}{B}})] = \Omega\left(\frac{\nu^2}{\mu MNK}\right).$$

Then, the three dimensional function $F([x,y,z]^\top) = F_1(x) + F_2(y) + F_3(z)$ will show bad convergence in any step-size regime. Furthermore,

$$\mu\boldsymbol{I} \preceq \min(\nabla^2 F_1, \nabla^2 F_2, \nabla^2 F_3)\boldsymbol{I} \preceq \nabla^2 F \preceq \max(\nabla^2 F_1, \nabla^2 F_2, \nabla^2 F_3)\boldsymbol{I} \preceq L\boldsymbol{I},$$

that is, if $F_1$, $F_2$ and $F_3$ are $\mu$-strongly convex and $L$-smooth, then so is $F$. Moreover, since the component functions in each coordinate are designed to satisfy Assumption 2 with $\nu$, the resulting three dimensional function $F$ also satisfies Assumption 2 with $\sqrt{3}\nu$.

Since the final lower bound is the minimum among the lower bounds obtained in the step-size ranges, the lower bound becomes $\Omega\left(\frac{\nu^2}{\mu MNK^2}\right)$ if $K \geq 513\kappa$, and $\Omega\left(\frac{\nu^2}{\mu MNK}\right)$ if $K < 513\kappa$ (in which case the second step-size range does not exist).

In the subsequent subsections, we prove the lower bounds for $F_1$, $F_2$, and $F_3$ separately.

---

[13]Recall that this is different from $\eta = \frac{B\log(MNK^2)}{\mu NK}$ in the theorem statement, because for the proofs, we consider an equivalent "rescaled" version of minibatch RR defined in the beginning of Appendix D.2.

[14]In fact, the functions constructed in this theorem are $\mu$-strongly convex, which is stronger than $\mu$-PL required in Definition 1.

## E.1 Lower bound for $\eta \leq \frac{B}{\mu NK}$

Consider the case where every function at every machine is the same: for all $i \in [N]$ and $m \in [M]$, $f_i^m(x) := \frac{\mu x^2}{2}$. Hence, $F_1(x) = \frac{\mu x^2}{2}$.

Let $x_{k,0}$ and $x_{k,\frac{N}{B}}$ denote the iterates where the $k$-th epoch starts and ends respectively. Then,

$$x_{k+1,0} = x_{k,\frac{N}{B}} = (1-\eta\mu)^{\frac{N}{B}} x_{k,0}.$$

Initializing at $x_{1,0} = \frac{\nu}{\mu}$ and unrolling this for $K$ epochs, we get

$$x_{K,\frac{N}{B}} = (1-\eta\mu)^{\frac{NK}{B}} \cdot \frac{\nu}{\mu} \geq \left(1 - \frac{B}{NK}\right)^{\frac{NK}{B}} \cdot \frac{\nu}{\mu} \geq \frac{\nu}{4\mu},$$

since $N \geq 2$, $K \geq 1$, and $B$ divides $N$. Hence, $F_1(x_{K,\frac{N}{B}}) = \Omega(\frac{\nu^2}{\mu})$.

## E.2 Lower bound for $\eta \geq \frac{B}{\mu NK}$ and $\eta \leq \frac{B}{513LN}$

For most part of this subsection, we consider iterates within a single epoch, and hence we will omit the subscripts denoting epochs. Let $x_0$ denote the iterate at the beginning of the epoch, and $x_i$ denote the iterate after the $i$-th communication round in that epoch. In our construction, each machine will have the same set of component functions, that is, there will be no inter-machine deviation. We therefore omit the superscript $m$ from the local component functions $f_i^m$. The function we construct for the lower bound and its component functions are as follows:

$$F_2(x) := \frac{1}{N} \left( \sum_{i=1}^{\frac{N}{2}} f_{+1}(x) + \sum_{i=\frac{N}{2}+1}^{N} f_{-1}(x) \right), \text{ where}$$

$$f_{+1}(x) := (L1_{x \leq 0} + \mu 1_{x>0})\frac{x^2}{2} + \nu x, \text{ and}$$

$$f_{-1}(x) := (L1_{x \leq 0} + \mu 1_{x>0})\frac{x^2}{2} - \nu x$$

Note that the function $F_2(x) = (L1_{x \leq 0} + \mu 1_{x>0})\frac{x^2}{2}$ is $\mu$-strongly convex and $L$-smooth with minimizer at 0, and also satisfies Assumption 2.

Let $\sigma^m$ be a random permutation of $\frac{N}{2}$ +1's and $\frac{N}{2}$ −1's. Then, machine $m$ computes gradients on $f_{-1}$ and $f_{+1}$ in the order given by $\sigma^m$. Let $\sigma_j^m$ denote the $j$-th ordered element of $\sigma^m$. Then,

$$\nabla f_{\sigma_j^m}(x) = (L1_{x \leq 0} + \mu 1_{x>0})x + \nu\sigma_j^m.$$

Hence, the last iterate of an epoch, $x_{\frac{N}{B}}$, is given by

$$\begin{aligned}
x_{\frac{N}{B}} - x_0 &= \sum_{i=0}^{\frac{N}{B}-1} \left( -\frac{\eta}{MB} \sum_{m=1}^{M} \sum_{j=iB+1}^{(i+1)B} \nabla f_{\sigma_j^m}(x_i) \right) \\
&= \sum_{i=0}^{\frac{N}{B}-1} \left( -\frac{\eta}{MB} \sum_{m=1}^{M} \sum_{j=iB+1}^{(i+1)B} ((L1_{x_i \leq 0} + \mu 1_{x_i>0})x_i + \nu\sigma_j^m) \right) \\
&= \sum_{i=0}^{\frac{N}{B}-1} \left( -\frac{\eta}{MB} \sum_{m=1}^{M} \sum_{j=iB+1}^{(i+1)B} (L1_{x_i \leq 0} + \mu 1_{x_i>0})x_i \right) \qquad \text{(Since } \sum_{j=1}^{N} \sigma_j^m = 0) \\
&= -\eta \sum_{i=0}^{\frac{N}{B}-1} (L1_{x_i \leq 0} + \mu 1_{x_i>0})x_i. \qquad (127)
\end{aligned}$$

Thus, $\mathbb{E}[x_{\frac{N}{B}} - x_0] = -\eta \sum_{i=0}^{\frac{N}{B}-1} \mathbb{E}[(L1_{x_i \leq 0} + \mu 1_{x_i > 0})x_i]$. We want to prove that $\mathbb{E}[x_{\frac{N}{B}}]$ keeps increasing over an epoch, that is $\mathbb{E}[x_{\frac{N}{B}} - x_0] > 0$ when $x_0$ is close enough to the minimizer 0.

For this, we first consider the case where the first iterate $x_0$ of the epoch satisfies $x_0 \geq 0$. The $x_0 < 0$ case will be considered later. For the case $x_0 \geq 0$, we will show that whenever $x_0$ is small, the expected amount of update made in the $(i+1)$-th iteration, $\mathbb{E}[(L1_{x_i \leq 0} + \mu 1_{x_i > 0})x_i]$, is negative in the first half of the epoch and not too big in the second half.

We use the following lemmas, proven in Appendices F.1 and F.2, respectively.

**Lemma 11.** *For $x_0 \geq 0$, $0 \leq i \leq \lfloor \frac{N}{2B} \rfloor$, $\eta \leq \frac{B}{513LN}$, and $\frac{L}{\mu} \geq 7695$,*

$$\mathbb{E}[(L1_{x_i \leq 0} + \mu 1_{x_i > 0})x_i] \leq \frac{6}{7}Lx_0 - \frac{\eta L \nu}{1536}\sqrt{\frac{i}{MB}}.$$

**Lemma 12.** *For $x_0 \geq 0$, $0 \leq i \leq \frac{N}{B} - 1$, and $\eta \leq \frac{B}{513LN}$,*

$$\mathbb{E}[(L1_{x_i \leq 0} + \mu 1_{x_i > 0})x_i] \leq \mu\left(1 + \frac{513i\eta L}{512}\right)x_0 + \frac{513\eta\mu\nu}{512}\sqrt{\frac{i}{MB}}.$$

The key intuition is that, for $\frac{L}{\mu}$ big enough, we can use the lemmas above in (127) to get

$$\mathbb{E}[x_{\frac{N}{B}} - x_0] = -\eta \sum_{i=0}^{\frac{N}{B}-1} (L1_{x_i \leq 0} + \mu 1_{x_i > 0})x_i$$
$$\approx \Omega\left(\eta\frac{N}{B}\eta L\nu\sqrt{\frac{N/B}{MB}}\right),$$

whenever $|x_0|$ is small. Multiplying the above by $K$ (for $K$ epochs) will give us the required lower bound (up to factors of $\frac{L}{\mu}$). We will make this approximate calculation precise in the rest of the proof.

Using the two lemmas above in (127), we get that

$$\mathbb{E}[x_{\frac{N}{B}} - x_0]$$
$$= -\eta \sum_{i=0}^{\frac{N}{B}-1} \mathbb{E}[(L1_{x_i \leq 0} + \mu 1_{x_i > 0})x_i]$$
$$= -\eta \sum_{i=0}^{\lfloor \frac{N}{2B} \rfloor} \mathbb{E}[(L1_{x_i \leq 0} + \mu 1_{x_i > 0})x_i] - \eta \sum_{i=\lfloor \frac{N}{2B} \rfloor+1}^{\frac{N}{B}-1} \mathbb{E}[(L1_{x_i \leq 0} + \mu 1_{x_i > 0})x_i]$$
$$\geq -\eta \sum_{i=0}^{\lfloor \frac{N}{2B} \rfloor} \left(\frac{6}{7}Lx_0 - \frac{\eta L\nu}{1536}\sqrt{\frac{i}{MB}}\right)$$
$$- \eta \sum_{i=\lfloor \frac{N}{2B} \rfloor+1}^{\frac{N}{B}-1} \left(\mu\left(1 + \frac{513i\eta L}{512}\right)x_0 + \frac{513\eta\mu\nu}{512}\sqrt{\frac{i}{MB}}\right). \tag{128}$$

Since $i\eta L \leq \frac{\eta LN}{B} \leq \frac{1}{513}$, $\mu \leq \frac{L}{7695}$, and $N/B \geq 2$, the following bound holds:

$$\sum_{i=0}^{\lfloor \frac{N}{2B} \rfloor} \frac{6}{7}L + \sum_{i=\lfloor \frac{N}{2B} \rfloor+1}^{\frac{N}{B}-1} \mu\left(1 + \frac{513i\eta L}{512}\right)$$
$$\leq \left(\left\lfloor\frac{N}{2B}\right\rfloor + 1\right)\frac{6L}{7} + \left(\frac{N}{B} - \left\lfloor\frac{N}{2B}\right\rfloor - 1\right)\frac{L}{7695}\left(1 + \frac{1}{512}\right)$$
$$\leq \frac{6LN}{7B} + \frac{LN}{7680B} \leq \frac{7LN}{8B}. \tag{129}$$

Also, note that $\lfloor \frac{N}{2B} \rfloor \geq \frac{N}{3B}$ whenever $N/B \geq 2$. We have

$$\sum_{i=0}^{\lfloor \frac{N}{2B} \rfloor} \sqrt{i} \geq \int_0^{\lfloor \frac{N}{2B} \rfloor} \sqrt{t}dt = \frac{2}{3}\left(\left\lfloor \frac{N}{2B} \right\rfloor\right)^{3/2} \geq \frac{2}{3}\left(\frac{N}{3B}\right)^{3/2} = \frac{2N^{3/2}}{9\sqrt{3}B^{3/2}}, \tag{130}$$

$$\sum_{i=\lfloor \frac{N}{2B} \rfloor+1}^{\frac{N}{B}-1} \sqrt{i} \leq \int_{\lfloor \frac{N}{2B} \rfloor+1}^{\frac{N}{B}} \sqrt{t}dt \leq \frac{2}{3}\left[\left(\frac{N}{B}\right)^{3/2} - \left(\left\lfloor \frac{N}{2B} \right\rfloor+1\right)^{3/2}\right]$$

$$\leq \frac{2}{3}\left[\left(\frac{N}{B}\right)^{3/2} - \left(\frac{N}{2B}\right)^{3/2}\right] = \frac{(2\sqrt{2}-1)N^{3/2}}{3\sqrt{2}B^{3/2}} \leq \frac{N^{3/2}}{2B^{3/2}}. \tag{131}$$

Substituting the bounds (129), (130), and (131) into (128), and using $\mu \leq \frac{L}{7695}$,

$$\mathbb{E}[x_{\frac{N}{B}} - x_0] \geq -\frac{7\eta LN}{8B}x_0 + \frac{\eta^2 L\nu N^{3/2}}{6912\sqrt{3}M^{1/2}B^2} - \frac{513\eta^2\mu\nu N^{3/2}}{1024M^{1/2}B^2}$$

$$\geq -\frac{7\eta LN}{8B}x_0 + \frac{\eta^2 L\nu N^{3/2}}{56000M^{1/2}B^2}. \tag{132}$$

For the other case $x_0 < 0$, we have the following Lemma, which we prove in Appendix F.3:

**Lemma 13.** *If $\eta \leq \frac{B}{513NL}$ and an epoch starts at $x_0 < 0$, then*

$$\mathbb{E}[x_{\frac{N}{B}} \mid x_0 < 0] \geq \left(1 - \frac{7\eta LN}{8B}\right)x_0.$$

*Further, if the first epoch of the algorithm is initialized at $0$, then for any starting iterate $x_0$ of any following epoch, we have $\mathbb{P}(x_0 \geq 0) \geq 1/2$.*

Using (132) and Lemma 13 we get

$$\mathbb{E}[x_{\frac{N}{B}}] = \mathbb{P}(x_0 \geq 0)\mathbb{E}[x_{\frac{N}{B}} \mid x_0 \geq 0] + \mathbb{P}(x_0 < 0)\mathbb{E}[x_{\frac{N}{B}} \mid x_0 < 0]$$

$$\geq \mathbb{P}(x_0 \geq 0)\left(\left(1 - \frac{7\eta LN}{8B}\right)x_0 + \frac{\eta^2 L\nu N^{3/2}}{56000M^{1/2}B^2}\right) + \mathbb{P}(x_0 < 0)\left(1 - \frac{7\eta LN}{8B}\right)x_0$$

$$\geq \left(1 - \frac{7\eta LN}{8B}\right)x_0 + \frac{\eta^2 L\nu N^{3/2}}{112000M^{1/2}B^2}.$$

Thus far, we have characterized the expected per-epoch update, starting from the initial iterate $x_0$ and iterating until the last iterate $x_{\frac{N}{B}}$ of the epoch. Now recall that we run the algorithm for $K$ epochs. Using $x_{k,i}$ to denote the $i$-th iterate of the $k$-th epoch, we get a lower bound on the expectation of the last iterate $x_{k,\frac{N}{B}}$ if we initialize at $x_{1,0} = 0$:

$$\mathbb{E}[x_{K,\frac{N}{B}}] \geq \left(1 - \frac{7\eta LN}{8B}\right)^K x_{1,0} + \frac{\eta^2 L\nu N^{3/2}}{112000M^{1/2}B^2}\sum_{k=0}^{K-1}\left(1 - \frac{7\eta LN}{8B}\right)^k$$

$$= \frac{\eta^2 L\nu N^{3/2}}{112000M^{1/2}B^2}\frac{1 - \left(1 - \frac{7\eta LN}{8B}\right)^K}{\frac{7\eta LN}{8B}}$$

$$= \frac{\eta\nu N^{1/2}}{98000M^{1/2}B}\left(1 - \left(1 - \frac{7\eta LN}{8B}\right)^K\right)$$

$$\geq \frac{\eta\nu N^{1/2}}{98000M^{1/2}B}\left(1 - \left(1 - \frac{7L}{8\mu K}\right)^K\right). \qquad \text{(Since } \eta \geq \frac{B}{\mu NK}\text{)}$$

Note that since $\frac{L}{\mu} \geq 7695$ and $K \geq \frac{513L}{\mu}$ (which is implied by $\frac{B}{\mu NK} \leq \eta \leq \frac{B}{513LN}$),

$$1 - \left(1 - \frac{7L}{8\mu K}\right)^K \geq 1 - e^{-\frac{7L}{8\mu}} \geq 1 - e^{-6733} \approx 1.$$

Hence, we get from $\eta \geq \frac{B}{\mu N K}$ that

$$\mathbb{E}[x_{K, \frac{N}{B}}] = \Omega\left(\frac{\eta \nu N^{1/2}}{M^{1/2} B}\right) = \Omega\left(\frac{\nu}{\mu M^{1/2} N^{1/2} K}\right),$$

and by Jensen's inequality, we finally have

$$\mathbb{E}[F(x_{K, \frac{N}{B}})] \geq \frac{1}{2}\mathbb{E}[\mu x_{K, \frac{N}{B}}^2] = \Omega(\mu \mathbb{E}[x_{K, \frac{N}{B}}]^2) = \Omega\left(\frac{\nu^2}{\mu M N K^2}\right).$$

## E.3   LOWER BOUND FOR $\eta \geq \frac{B}{\mu N K}$ AND $\eta \geq \frac{B}{513 L N}$

Similar to earlier parts of the proof, here as well, each machine will have the same component functions, that is, there will be no inter-machine deviation. The proof uses a similar construction as Safran & Shamir (2020; 2021):

$$F_3(x) := \frac{1}{N}\left(\sum_{i=1}^{\frac{N}{2}} f_{+1}(x) + \sum_{i=\frac{N}{2}+1}^{N} f_{-1}(x)\right), \text{ where}$$

$$f_{+1}(x) := \frac{Lx^2}{2} + \nu x, \text{ and } f_{-1}(x) := \frac{Lx^2}{2} - \nu x.$$

Hence, $F_3(x) = \frac{Lx^2}{2}$, and has its minimizer at 0.

We first compute the expected "progress" over a given epoch. For simplicity, let us omit the subscript for epochs for now. Let $x_0$ denote the iterate at the beginning of the epoch and $x_i$ denote the iterate after the $i$-th communication round in that epoch. For a given epoch, let $\sigma^m$ be the permutation of $\frac{N}{2}+1$'s and $\frac{N}{2}-1$'s, sampled by machine $m$. Then,

$$x_{\frac{N}{B}} = x_{\frac{N}{B}-1} - \frac{\eta}{MB}\sum_{m=1}^{M}\sum_{j=(\frac{N}{B}-1)B+1}^{N}(Lx_{\frac{N}{B}-1} + \nu\sigma_j^m)$$

$$= (1-\eta L)x_{\frac{N}{B}-1} - \frac{\eta\nu}{MB}\sum_{m=1}^{M}\sum_{j=(\frac{N}{B}-1)B+1}^{N}\sigma_j^m$$

$$= \cdots$$

$$= (1-\eta L)^{\frac{N}{B}}x_0 - \frac{\eta\nu}{MB}\sum_{i=1}^{\frac{N}{B}}(1-\eta L)^{\frac{N}{B}-i}\sum_{m=1}^{M}\sum_{j=(i-1)B+1}^{iB}\sigma_j^m.$$

For the rest of this proof, $x_i^2$ refers to the square of the $i$-th iterate. Then,

$$\mathbb{E}[x_{\frac{N}{B}}^2] = (1-\eta L)^{\frac{2N}{B}}x_0^2 - \frac{2\eta\nu(1-\eta L)^{\frac{N}{B}}x_0}{MB}\mathbb{E}\left[\left(\sum_{i=1}^{\frac{N}{B}}(1-\eta L)^{\frac{N}{B}-i}\sum_{m=1}^{M}\sum_{j=(i-1)B+1}^{iB}\sigma_j^m\right)\right]$$

$$+ \frac{\eta^2\nu^2}{M^2 B^2}\mathbb{E}\left[\left(\sum_{i=1}^{\frac{N}{B}}(1-\eta L)^{\frac{N}{B}-i}\sum_{m=1}^{M}\sum_{j=(i-1)B+1}^{iB}\sigma_j^m\right)^2\right]$$

$$= (1-\eta L)^{\frac{2N}{B}}x_0^2 + \frac{\eta^2\nu^2}{M^2 B^2}\mathbb{E}\left[\left(\sum_{i=1}^{\frac{N}{B}}(1-\eta L)^{\frac{N}{B}-i}\sum_{m=1}^{M}\sum_{j=(i-1)B+1}^{iB}\sigma_j^m\right)^2\right], \quad (133)$$

where we used the fact that $\mathbb{E}[\sigma_j^m] = 0$. Further, because $\sigma^m$ and $\sigma^{m'}$ are independent and identically distributed for different $m$ and $m'$, we get that

$$\mathbb{E}\left[\left(\sum_{i=1}^{\frac{N}{B}}(1-\eta L)^{\frac{N}{B}-i}\sum_{m=1}^{M}\sum_{j=(i-1)B+1}^{iB}\sigma_j^m\right)^2\right]$$

$$= \mathbb{E}\left[\left(\sum_{m=1}^{M}\sum_{i=1}^{\frac{N}{B}}(1-\eta L)^{\frac{N}{B}-i}\sum_{j=(i-1)B+1}^{iB}\sigma_j^m\right)^2\right]$$

$$= \sum_{m=1}^{M}\mathbb{E}\left[\left(\sum_{i=1}^{\frac{N}{B}}(1-\eta L)^{\frac{N}{B}-i}\sum_{j=(i-1)B+1}^{iB}\sigma_j^m\right)^2\right]$$

$$+ \sum_{m\neq m'}\mathbb{E}\left[\sum_{i=1}^{\frac{N}{B}}(1-\eta L)^{\frac{N}{B}-i}\sum_{j=(i-1)B+1}^{iB}\sigma_j^m\right]\mathbb{E}\left[\sum_{i=1}^{\frac{N}{B}}(1-\eta L)^{\frac{N}{B}-i}\sum_{j=(i-1)B+1}^{iB}\sigma_j^{m'}\right]$$

$$= M\mathbb{E}\left[\left(\sum_{i=1}^{\frac{N}{B}}(1-\eta L)^{\frac{N}{B}-i}\sum_{j=(i-1)B+1}^{iB}\sigma_j^1\right)^2\right],$$

where the last equality used the fact that $\mathbb{E}[\sigma_j^m] = 0$ for all $m \in [M]$ and $i \in [N]$, and that $\sigma^m$ are identically distributed. Since we only consider the permutation $\sigma^1$ (i.e., the one for machine 1) from now on, we henceforth omit the superscript. Substituting this to (133) gives

$$\mathbb{E}[x_{\frac{N}{B}}^2] = (1-\eta L)^{\frac{2N}{B}}x_0^2 + \frac{\eta^2\nu^2}{MB^2}\underbrace{\mathbb{E}\left[\left(\sum_{i=1}^{\frac{N}{B}}(1-\eta L)^{\frac{N}{B}-i}\sum_{j=(i-1)B+1}^{iB}\sigma_j\right)^2\right]}_{=:\Phi}. \tag{134}$$

From (134), we have calculated the per-epoch expected update. Recall that we run the algorithm for $K$ epochs. Using $x_{k,i}$ to denote the $i$-th iterate of the $k$-th epoch, we get a lower bound on the expectation of the last iterate $x_{k,\frac{N}{B}}$ squared:

$$\mathbb{E}[x_{k,\frac{N}{B}}^2] = (1-\eta L)^{\frac{2NK}{B}}x_{1,0}^2 + \frac{\eta^2\nu^2}{MB^2}\Phi\sum_{k=0}^{K-1}(1-\eta L)^{\frac{2Nk}{B}} \geq \frac{\eta^2\nu^2}{MB^2}\Phi, \tag{135}$$

where the inequality used $x_{1,0} = 0$ and $\sum_{k=0}^{K-1}(1-\eta L)^{\frac{2Nk}{B}} \geq (1-\eta L)^0 = 1$. Next, we analyze the expectation term, i.e., $\Phi$, defined in (134).

$$\Phi := \mathbb{E}\left[\left(\sum_{i=1}^{\frac{N}{B}}(1-\eta L)^{\frac{N}{B}-i}\sum_{j=(i-1)B+1}^{iB}\sigma_j\right)^2\right]$$

$$= \sum_{j=1}^{N}(1-\eta L)^{2(\frac{N}{B}-\lfloor\frac{j-1}{B}\rfloor-1)}\sigma_j^2 + \sum_{j\neq j'}(1-\eta L)^{\frac{N}{B}-\lfloor\frac{j-1}{B}\rfloor-1}(1-\eta L)^{\frac{N}{B}-\lfloor\frac{j'-1}{B}\rfloor-1}\mathbb{E}[\sigma_j\sigma_{j'}]$$

Noting that $\sigma_j^2 = 1$ and $\mathbb{E}[\sigma_j\sigma_{j'}] = -\frac{1}{N-1}$, we get

$$\Phi = B\sum_{j=0}^{\frac{N}{B}-1}(1-\eta L)^{2j} - \frac{1}{N-1}\sum_{j\neq j'}(1-\eta L)^{\frac{N}{B}-\lfloor\frac{j-1}{B}\rfloor-1}(1-\eta L)^{\frac{N}{B}-\lfloor\frac{j'-1}{B}\rfloor-1}$$

$$= B\sum_{j=0}^{\frac{N}{B}-1}(1-\eta L)^{2j} - \frac{1}{N-1}\left(\left(\sum_{j=1}^{N}(1-\eta L)^{\frac{N}{B}-\lfloor\frac{j-1}{B}\rfloor-1}\right)^2 - \sum_{j=1}^{N}(1-\eta L)^{2(\frac{N}{B}-\lfloor\frac{j-1}{B}\rfloor-1)}\right)$$

$$= B\sum_{j=0}^{\frac{N}{B}-1}(1-\eta L)^{2j} - \frac{1}{N-1}\left(B^2\left(\sum_{j=0}^{\frac{N}{B}-1}(1-\eta L)^j\right)^2 - B\sum_{j=0}^{\frac{N}{B}-1}(1-\eta L)^{2j}\right)$$

$$= \frac{B^2}{N-1}\left(\frac{N}{B}\sum_{j=0}^{\frac{N}{B}-1}(1-\eta L)^{2j} - \left(\sum_{j=0}^{\frac{N}{B}-1}(1-\eta L)^j\right)^2\right)$$

$$= \frac{B^2(\frac{N}{B} - 1)}{N - 1} \left( \left(1 + \frac{1}{\frac{N}{B} - 1}\right) \sum_{j=0}^{\frac{N}{B}-1} (1 - \eta L)^{2j} - \frac{1}{\frac{N}{B} - 1} \left( \sum_{j=0}^{\frac{N}{B}-1} (1 - \eta L)^j \right)^2 \right). \tag{136}$$

Note that the term in the parenthesis is exactly the right hand side of Equation (23) in Safran & Shamir (2020) modulo $n$ and $\alpha$ replaced with $\frac{N}{B}$ and $\eta L$, respectively. Hence, by Lemma 1 of Safran & Shamir (2020), we have

$$\left(1 + \frac{1}{\frac{N}{B} - 1}\right) \sum_{j=0}^{\frac{N}{B}-1} (1 - \eta L)^{2j} - \frac{1}{\frac{N}{B} - 1} \left( \sum_{j=0}^{\frac{N}{B}-1} (1 - \eta L)^j \right)^2 \geq c \cdot \min \left\{ \frac{1}{\eta L}, \frac{\eta^2 L^2 N^3}{B^3} \right\},$$
$$\tag{137}$$

for some universal constant $c > 0$. Using the fact that $\eta \geq \frac{B}{513LN}$, it is easy to check that the RHS of (137) is lower-bounded by $\frac{c'}{\eta L}$, where $c' > 0$ is a universal constant. Combining (135), (136), and (137) gives

$$\mathbb{E}[x_{k, \frac{N}{B}}^2] \geq \frac{\eta^2 \nu^2}{MB^2} \cdot \frac{B^2(\frac{N}{B} - 1)}{N - 1} \cdot \frac{c'}{\eta L} = \frac{c' \eta \nu^2}{LMB} \frac{N - B}{N - 1}.$$

Since $2B$ divides $N$, we have $B \leq N/2$. Since $N \geq 2$, we have $\frac{N-B}{N-1} \geq \frac{N}{2(N-1)} \geq \frac{1}{2}$. Using this and the fact that $\eta \geq \frac{B}{\mu NK}$, we get

$$\mathbb{E}[F_3(x_{k, \frac{N}{B}})] = \frac{L}{2} \mathbb{E}[x_{k, \frac{N}{B}}^2] \geq \frac{c' \nu^2}{4\mu MNK}.$$

## F  PROOFS OF HELPER LEMMAS FOR APPENDIX E

### F.1  PROOF OF LEMMA 11

First, if $i = 0$ then the lemma trivially holds, because $x_0 \geq 0$ gives

$$\mathbb{E}[(L1_{x_0 \leq 0} + \mu 1_{x_0 > 0})x_0] = \mu x_0 \leq \frac{6}{7} L x_0.$$

The inequality holds because $\frac{L}{\mu} \geq 7695$.

For the rest of the proof, we consider the case $1 \leq i \leq \frac{N}{2B}$. By the law of total expectation we have

$$\mathbb{E}[(L1_{x_i \leq 0} + \mu 1_{x_i > 0})x_i] = \mathbb{P}\left(\sum_{m=1}^{M} \sum_{j=1}^{iB} \sigma_j^m > 0\right) \mathbb{E}\left[(L1_{x_i \leq 0} + \mu 1_{x_i > 0})x_i \,\middle|\, \sum_{m=1}^{M} \sum_{j=1}^{iB} \sigma_j^m > 0\right]$$

$$+ \mathbb{P}\left(\sum_{m=1}^{M} \sum_{j=1}^{iB} \sigma_j^m \leq 0\right) \mathbb{E}\left[(L1_{x_i \leq 0} + \mu 1_{x_i > 0})x_i \,\middle|\, \sum_{m=1}^{M} \sum_{j=1}^{iB} \sigma_j^m \leq 0\right]$$

$$\leq \mathbb{P}\left(\sum_{m=1}^{M} \sum_{j=1}^{iB} \sigma_j^m > 0\right) L\mathbb{E}\left[x_i \,\middle|\, \sum_{m=1}^{M} \sum_{j=1}^{iB} \sigma_j^m > 0\right]$$

$$+ \mathbb{P}\left(\sum_{m=1}^{M} \sum_{j=1}^{iB} \sigma_j^m \leq 0\right) \mu\mathbb{E}\left[x_i \,\middle|\, \sum_{m=1}^{M} \sum_{j=1}^{iB} \sigma_j^m \leq 0\right], \tag{138}$$

where the last inequality used the fact that $(L1_{t \leq 0} + \mu 1_{t > 0})t \leq Lt$ and $(L1_{t \leq 0} + \mu 1_{t > 0})t \leq \mu t$ for any $t \in \mathbb{R}$.

Define $\mathcal{E} := \sum_{m=1}^{M} \sum_{j=1}^{iB} \sigma_j^m$. We handle each of the two expectations in (138) separately. We first bound $\mathbb{E}[x_i | \mathcal{E} > 0]$.

$$\mathbb{E}[x_i | \mathcal{E} > 0] = \mathbb{E}\left[x_0 - \sum_{j=0}^{i-1} \frac{\eta}{MB} \sum_{m=1}^{M} \sum_{k=jB+1}^{(j+1)B} (\nu \sigma_k^m + (L1_{x_j \leq 0} + \mu 1_{x_j > 0})x_j) \,\middle|\, \mathcal{E} > 0\right]$$




$$= \mathbb{E}\left[x_0 - \sum_{j=0}^{i-1} \frac{\eta}{MB} \sum_{m=1}^{M} \sum_{k=jB+1}^{(j+1)B} (\nu\sigma_k^m + (L1_{x_j \leq 0} + \mu 1_{x_j > 0})(x_j - x_0)) \middle| \mathcal{E} > 0\right]$$

$$+ \mathbb{E}\left[-\sum_{j=0}^{i-1} \frac{\eta}{MB} \sum_{m=1}^{M} \sum_{k=jB+1}^{(j+1)B} (L1_{x_j \leq 0} + \mu 1_{x_j > 0})x_0 \middle| \mathcal{E} > 0\right]$$

$$= x_0 \mathbb{E}\left[1 - \eta \sum_{j=0}^{i-1}(L1_{x_j \leq 0} + \mu 1_{x_j > 0}) \middle| \mathcal{E} > 0\right] - \frac{\eta\nu}{MB}\mathbb{E}\left[\mathcal{E} | \mathcal{E} > 0\right]$$

$$- \eta \sum_{j=0}^{i-1} \mathbb{E}\left[(L1_{x_j \leq 0} + \mu 1_{x_j > 0})(x_j - x_0) \middle| \mathcal{E} > 0\right]$$

$$\leq x_0 \mathbb{E}\left[1 - \eta \sum_{j=0}^{i-1}(L1_{x_j \leq 0} + \mu 1_{x_j > 0}) \middle| \mathcal{E} > 0\right] - \frac{\eta\nu}{MB}\mathbb{E}\left[\mathcal{E} | \mathcal{E} > 0\right]$$

$$+ \eta L \sum_{j=0}^{i-1} \mathbb{E}[|x_j - x_0| \,|\, \mathcal{E} > 0]. \tag{139}$$

Next, we use the following lemma to bound the conditional expectations that arise in (139). This lemma is proven in Appendix F.4 and it may be of independent interest to readers.

**Lemma 14.** *For $m \in [M]$, let $\sigma^m$ be a random permutation of $\frac{N}{2}$ +1's and $\frac{N}{2}$ −1's. Then, for any $i \leq \frac{N}{2}$ and $k \leq \frac{B}{2}$, we have*

$$\frac{1}{64}\left(\sqrt{\frac{i}{M}} + \sqrt{k}\right) \leq \mathbb{E}\left[\left|\left(\frac{1}{M}\sum_{m=1}^{M}\sum_{j=1}^{i}\sigma_j^m\right) + \sum_{j=i+1}^{i+k}\sigma_j^M\right|\right].$$

*Furthermore, for any $0 \leq i \leq N$ and $0 \leq k \leq N$ satisfying $i + k \leq N$, we have*

$$\mathbb{E}\left[\left|\left(\frac{1}{M}\sum_{m=1}^{M}\sum_{j=1}^{i}\sigma_j^m\right) + \sum_{j=i+1}^{i+k}\sigma_j^M\right|\right] \leq \sqrt{\frac{i}{M}} + \sqrt{k}.$$

*Lastly, for any $0 \leq i \leq \frac{N}{2}$ and $0 \leq k \leq \frac{B}{2}$ satisfying $i + k \geq 1$, we have*

$$\mathbb{P}\left(\sum_{m=1}^{M}\sum_{j=1}^{i}\sigma_j^m + M\sum_{j=i+1}^{i+k}\sigma_j^M > 0\right) = \mathbb{P}\left(\sum_{m=1}^{M}\sum_{j=1}^{i}\sigma_j^m + M\sum_{j=i+1}^{i+k}\sigma_j^M < 0\right) \geq \frac{1}{6}.$$

Lemma 14 implies that $\frac{1}{M}\mathbb{E}\left[\mathcal{E}|\mathcal{E} > 0\right] \in \left[\frac{1}{64}\sqrt{\frac{iB}{M}}, \sqrt{\frac{iB}{M}}\right]$ and $\mathbb{P}(\mathcal{E} > 0) = \mathbb{P}(\mathcal{E} < 0) \geq 1/6$. From this, we get

$$\frac{\eta\nu}{MB}\mathbb{E}\left[\mathcal{E}|\mathcal{E} > 0\right] \geq \frac{\eta\nu}{64}\sqrt{\frac{i}{MB}}, \tag{140}$$

$$\mathbb{E}[|x_j - x_0| \,|\, \mathcal{E} > 0] \leq \frac{\mathbb{E}[|x_j - x_0|]}{\mathbb{P}(\mathcal{E} > 0)} \leq 6\mathbb{E}[|x_j - x_0|]. \tag{141}$$

Also, since $\eta \leq \frac{B}{LN}$ we have

$$1 - i\eta\mu \geq 1 - \eta\sum_{j=0}^{i-1}(L1_{x_j \leq 0} + \mu 1_{x_j > 0}) \geq 1 - \frac{\eta LN}{B} \geq 0,$$

which implies that

$$x_0\mathbb{E}\left[1 - \eta\sum_{j=0}^{i-1}(L1_{x_j \leq 0} + \mu 1_{x_j > 0}) \middle| \mathcal{E} > 0\right] \leq (1 - i\eta\mu)x_0. \tag{142}$$

Substituting (140), (141), and (142) to (139), we obtain

$$\mathbb{E}\left[x_i | \mathcal{E} > 0\right] \leq (1 - i\eta\mu)x_0 - \frac{\eta\nu}{64}\sqrt{\frac{i}{MB}} + 6\eta L \sum_{j=0}^{i-1} \mathbb{E}[|x_j - x_0|]. \tag{143}$$

Next, we have the following lemma that we can apply to $\mathbb{E}[|x_j - x_0|]$. Proof of Lemma 15 can be found in Appendix F.5.

**Lemma 15.** *For $x_0 \geq 0$, $0 \leq i \leq \frac{N}{B} - 1$ and $\eta \leq \frac{B}{513LN}$,*

$$\mathbb{E}[|x_i - x_0|] \leq \frac{513}{512}\eta\nu\sqrt{\frac{i}{MB}} + \frac{513}{512}i\eta Lx_0.$$

Applying this lemma to (143), we get

$$\mathbb{E}\left[x_i | \mathcal{E} > 0\right] \leq (1 - i\eta\mu)x_0 - \frac{\eta\nu}{64}\sqrt{\frac{i}{MB}} + \frac{1539\eta^2 L\nu}{256\sqrt{MB}}\sum_{j=0}^{i-1}\sqrt{j} + \frac{1539\eta^2 L^2 x_0}{256}\sum_{j=0}^{i-1}j$$

$$\leq (1 - i\eta\mu)x_0 - \frac{\eta\nu}{64}\sqrt{\frac{i}{MB}} + \frac{513i^{3/2}\eta^2 L\nu}{128\sqrt{MB}} + \frac{1539i^2\eta^2 L^2}{512}x_0$$

$$= \left(1 - i\eta\mu + \frac{1539i^2\eta^2 L^2}{512}\right)x_0 - \left(\frac{1}{64} - \frac{513i\eta L}{128}\right)\eta\nu\sqrt{\frac{i}{MB}}$$

$$\leq \left(1 - i\eta\mu + \frac{3i\eta L}{512}\right)x_0 - \frac{\eta\nu}{128}\sqrt{\frac{i}{MB}}. \tag{144}$$

where we got the last inequality by using the fact that $i\eta L \leq \frac{\eta LN}{B} \leq \frac{1}{513}$, which follows from $\eta \leq \frac{B}{513LN}$. So far, we have obtained an upper bound for $\mathbb{E}\left[x_i | \mathcal{E} > 0\right]$.

Recall that there is another conditional expectation in (138) that we want to bound, namely $\mathbb{E}\left[x_i | \mathcal{E} \leq 0\right]$. We bound it below, using the tools developed so far. For $i \leq \frac{N}{2B}$,

$$\mathbb{E}\left[x_i | \mathcal{E} \leq 0\right] = x_0 + \mathbb{E}\left[x_i - x_0 \mid \mathcal{E} \leq 0\right]$$

$$\leq x_0 + \mathbb{E}\left[|x_i - x_0| \mid \mathcal{E} \leq 0\right]$$

$$\leq x_0 + \frac{\mathbb{E}\left[|x_i - x_0|\right]}{\mathbb{P}(\mathcal{E} \leq 0)}$$

$$\leq x_0 + 6\mathbb{E}\left[|x_i - x_0|\right] \qquad \text{(Using Lemma 14)}$$

$$\leq x_0 + \frac{1539\eta\nu}{256}\sqrt{\frac{i}{MB}} + \frac{1539i\eta Lx_0}{256} \qquad \text{(Using Lemma 15)}$$

$$\leq \left(1 + \frac{1539i\eta L}{256}\right)x_0 + \frac{1539\eta\nu}{256}\sqrt{\frac{i}{MB}} \tag{145}$$

Using (144) and (145) in (138), we get that for $i \leq \frac{N}{2B}$:

$$\mathbb{E}[(L1_{x_i \leq 0} + \mu 1_{x_i > 0})x_i]$$

$$\leq \mathbb{P}\left(\mathcal{E} > 0\right)L\mathbb{E}\left[x_i | \mathcal{E} > 0\right] + \mathbb{P}\left(\mathcal{E} \leq 0\right)\mu\mathbb{E}\left[x_i | \mathcal{E} \leq 0\right]$$

$$\leq \mathbb{P}\left(\mathcal{E} > 0\right)L\left(\left(1 - i\eta\mu + \frac{3i\eta L}{512}\right)x_0 - \frac{\eta\nu}{128}\sqrt{\frac{i}{MB}}\right)$$

$$+ \mathbb{P}\left(\mathcal{E} \leq 0\right)\mu\left(\left(1 + \frac{1539i\eta L}{256}\right)x_0 + \frac{1539\eta\nu}{256}\sqrt{\frac{i}{MB}}\right). \tag{146}$$

From Lemma 14, note that $\frac{1}{6} \leq \mathbb{P}\left(\mathcal{E} > 0\right) \leq \frac{5}{6}$ and $\frac{1}{6} \leq \mathbb{P}\left(\mathcal{E} \leq 0\right) \leq \frac{5}{6}$. We use these inequalities, along with $i\eta L \leq \frac{\eta LN}{B} \leq \frac{1}{513}$ and $\frac{L}{\mu} \geq 7695$, to bound the terms appearing in (146).

$$\mathbb{P}\left(\mathcal{E} > 0\right)L\left(1 - i\eta\mu + \frac{3i\eta L}{512}\right)x_0 + \mathbb{P}\left(\mathcal{E} \leq 0\right)\mu\left(1 + \frac{1539i\eta L}{256}\right)x_0$$

$$\leq \frac{5}{6}L\left(1 + \frac{1}{87552}\right)x_0 + \frac{5}{6} \cdot \frac{L}{7695}\left(1 + \frac{3}{256}\right)x_0 \leq \frac{6}{7}Lx_0. \tag{147}$$

We also have

$$-\mathbb{P}\left(\mathcal{E} > 0\right)\frac{\eta L\nu}{128}\sqrt{\frac{i}{MB}} + \mathbb{P}\left(\mathcal{E} \leq 0\right)\frac{1539\eta\mu\nu}{256}\sqrt{\frac{i}{MB}}$$
$$\leq -\frac{\eta L\nu}{768}\sqrt{\frac{i}{MB}} + \frac{2565\eta\mu\nu}{512}\sqrt{\frac{i}{MB}}$$
$$\leq -\frac{\eta L\nu}{1536}\sqrt{\frac{i}{MB}}, \tag{148}$$

where we used the assumption $\frac{L}{\mu} \geq 7695$. Substituting (147) and (148) to (146), we get

$$\mathbb{E}[(L1_{x_i \leq 0} + \mu 1_{x_i > 0})x_i] \leq \frac{6}{7}Lx_0 - \frac{\eta L\nu}{1536}\sqrt{\frac{i}{MB}},$$

as desired.

### F.2 PROOF OF LEMMA 12

$$\mathbb{E}[(L1_{x_i \leq 0} + \mu 1_{x_i > 0})x_i] \leq \mu\mathbb{E}[x_i]$$
$$= \mu x_0 + \mu\mathbb{E}[x_i - x_0]$$
$$\leq \mu x_0 + \mu\mathbb{E}[|x_i - x_0|]$$
$$\leq \mu x_0 + \frac{513}{512}i\eta L\mu x_0 + \frac{513}{512}\eta\mu\nu\sqrt{\frac{i}{MB}}. \qquad \text{(Using Lemma 15)}$$

### F.3 PROOF OF LEMMA 13

We consider iterates within a single epoch, and hence we will omit the subscripts denoting epochs. In our construction, each machine has the same set of component functions, that is, there will be no inter-machine deviation. We therefore omit the superscript $m$ from the local component functions. Consider the function

$$G_2(x) := \frac{1}{N}\left(\sum_{i=1}^{\frac{N}{2}} g_{+1}(x) + \sum_{i=\frac{N}{2}+1}^{N} g_{-1}(x)\right), \text{ where}$$

$$g_{+1}(x) := \frac{Lx^2}{2} + \nu x, \text{ and } g_{-1}(x) := \frac{Lx^2}{2} - \nu x.$$

Hence, $G_2(x) = \frac{Lx^2}{2}$. We will prove the lemma by coupling iterates corresponding to $F_2$ and $G_2$. In particular, we will perform minibatch RR on $F_2$ and $G_2$ such that both start the given epoch at $x_0$ and all the corresponding machines use the same random permutations. Let $x_{i,F}$ be the iterate after the $i$-th round of communication for $F_2$ and $x_{i,G}$ be the iterate after the $i$-th round of communication for $G_2$. We use mathematical induction to prove that $x_{i,F} \geq x_{i,G}$ for all $i = 0, \ldots, \frac{N}{B}$. After that, we will use this to prove our desired statement $\mathbb{E}[x_{\frac{N}{B},F} \mid x_0 < 0] \geq (1 - \frac{7\eta LN}{8B})x_0$. Let $\sigma^m$ be a random permutation of $\frac{N}{2}$ +1's and $\frac{N}{2}$ −1's.

**Base case.** $x_{0,F} \geq x_{0,G}$ since both start the epoch at the same point $x_0$.

**Inductive case.** There can be three cases:

- Case 1: $x_{i,F} \geq x_{i,G} \geq 0$. Then,

$$x_{i+1,F} - x_{i+1,G} = x_{i,F} - x_{i,G} - \frac{\eta}{MB}\sum_{m=1}^{M}\sum_{j=iB+1}^{(i+1)B}\left(\nabla f_{\sigma_j^m}(x_{i,F}) - \nabla g_{\sigma_j^m}(x_{i,G})\right)$$

$$= x_{i,F} - x_{i,G} - \frac{\eta}{MB} \sum_{m=1}^{M} \sum_{j=iB+1}^{(i+1)B} \left( \mu x_{i,F} + \nu \sigma_j^m - L x_{i,G} - \nu \sigma_j^m \right)$$

$$= x_{i,F} - x_{i,G} - \eta \left( \mu x_{i,F} - L x_{i,G} \right)$$

$$= x_{i,F}(1 - \eta\mu) - x_{i,G}(1 - \eta L)$$

$$\geq 0.$$

- Case 2: $0 \geq x_{i,F} \geq x_{i,G}$. Then,

$$x_{i+1,F} - x_{i+1,G} = x_{i,F} - x_{i,G} - \frac{\eta}{MB} \sum_{m=1}^{M} \sum_{j=iB+1}^{(i+1)B} \left( \nabla f_{\sigma_j^m}(x_{i,F}) - \nabla g_{\sigma_j^m}(x_{i,G}) \right)$$

$$= x_{i,F} - x_{i,G} - \frac{\eta}{MB} \sum_{m=1}^{M} \sum_{j=iB+1}^{(i+1)B} \left( L x_{i,F} + \nu \sigma_j^m - L x_{i,G} - \nu \sigma_j^m \right)$$

$$= x_{i,F} - x_{i,G} - \eta \left( L x_{i,F} - L x_{i,G} \right)$$

$$= x_{i,F}(1 - \eta L) - x_{i,G}(1 - \eta L)$$

$$\geq 0.$$

- Case 3: $x_{i,F} \geq 0 \geq x_{i,G}$. Then,

$$x_{i+1,F} - x_{i+1,G} = x_{i,F} - x_{i,G} - \frac{\eta}{MB} \sum_{m=1}^{M} \sum_{j=iB+1}^{(i+1)B} \left( \nabla f_{\sigma_j^m}(x_{i,F}) - \nabla g_{\sigma_j^m}(x_{i,G}) \right)$$

$$= x_{i,F} - x_{i,G} - \frac{\eta}{MB} \sum_{m=1}^{M} \sum_{j=iB+1}^{(i+1)B} \left( \mu x_{i,F} + \nu \sigma_j^m - L x_{i,G} - \nu \sigma_j^m \right)$$

$$= x_{i,F} - x_{i,G} - \eta \left( \mu x_{i,F} - L x_{i,G} \right)$$

$$= x_{i,F}(1 - \eta\mu) - x_{i,G}(1 - \eta L).$$

Note that since $\eta \leq \frac{B}{LN}$, $x_{i,F}(1 - \eta\mu) \geq 0$ and $x_{i,G}(1 - \eta L) \leq 0$, which proves that $x_{i+1,F} - x_{i+1,G} \geq 0$.

Thus, we see that $x_{i+1,F} \geq x_{i+1,G}$. Further, by linearity of expectation and gradient, it is easy to check that

$$\mathbb{E}[x_{\frac{N}{B},G}] = \mathbb{E}[x_{\frac{N}{B}-1,G} - \eta \nabla G_2(x_{\frac{N}{B}-1,G})]$$

$$= (1 - \eta L)\mathbb{E}[x_{\frac{N}{B}-1,G}]$$

$$= \cdots$$

$$= (1 - \eta L)^{\frac{N}{B}} x_0.$$

Using the result that $x_{\frac{N}{B},F} \geq x_{\frac{N}{B},G}$ which we proved above, we get $\mathbb{E}[x_{\frac{N}{B},F}] \geq (1 - \eta L)^{\frac{N}{B}} x_0$ for any initial iterate $x_0$. Specifically for $x_0 < 0$, this implies $\mathbb{E}[x_{\frac{N}{B},F} \mid x_0 < 0] \geq (1 - \eta L)^{\frac{N}{B}} x_0$.

Further, since $\eta L \leq \frac{B}{513N}$, we have $(1 - \eta L)^{\frac{N}{B}} \leq 1 - \frac{7\eta LN}{8B}$. This is because $1 - \frac{7zN}{8B} - (1-z)^{\frac{N}{B}}$ is nonnegative on the interval $\left[ 0, 1 - (7/8)^{\frac{1}{N/B-1}} \right]$, and $1 - (7/8)^{\frac{1}{N/B-1}} \geq \frac{B}{513N}$ for all $\frac{N}{B} \geq 2$. To see why, note that $(1 - \frac{1}{513(n-1)})^{n-1} \geq \frac{7}{8}$ for all $n \geq 2$, and this gives $1 - (7/8)^{\frac{1}{n-1}} \geq \frac{1}{513(n-1)}$, which then implies $1 - (7/8)^{\frac{1}{n-1}} \geq \frac{1}{513n}$ for all $n \geq 2$. Therefore, for $x_0 < 0$, we have $\mathbb{E}[x_{\frac{N}{B},F} \mid x_0 < 0] \geq (1 - \frac{7\eta LN}{8B})x_0$.

For the last statement of the lemma, note that by symmetry of the function $G_2$, if we initialize the Algorithm 2 at 0, then for any starting iterate of an epoch we have $\mathbb{P}(x_{0,G} \geq 0) \geq 1/2$. This combined with the fact that $x_{i,F} \geq x_{i,G}$ gives us that $\mathbb{P}(x_{0,F} \geq 0) \geq 1/2$.

### F.4 PROOF OF LEMMA 14

For $m = 1, \ldots, M$, let $\sigma^m$ be a random permutation of $\frac{N}{2}$ +1's and $\frac{N}{2}$ −1's. Then, we first show that for any $i \leq N/2$ and $k \leq B/2$,

$$\frac{1}{64}\left(\sqrt{\frac{i}{M}} + \sqrt{k}\right) \leq \mathbb{E}\left[\left|\left(\frac{1}{M}\sum_{m=1}^{M}\sum_{j=1}^{i}\sigma_j^m\right) + \sum_{j=i+1}^{i+k}\sigma_j^M\right|\right].$$

To prove the lower bound, we will use Khintchine's inequality along with Lemma 12 from Rajput et al. (2020). Let us define random variables $\mathrm{a}_m := |\frac{1}{M}\sum_{j=1}^{i}\sigma_j^m|$, $\mathrm{x}_m := \text{sign}(\sum_{j=1}^{i}\sigma_j^m)$, $\mathrm{b}_M := |\sum_{j=i+1}^{i+k}\sigma_j^M|$, and $\mathrm{y}_M := \text{sign}(\sum_{j=i+1}^{i+k}\sigma_j^M)$. For $\mathrm{x}_m$, if the sum $\sum_{j=1}^{i}\sigma_j^m = 0$ then $\mathrm{x}_m$ is +1 with probability 0.5 and −1 with probability 0.5. Ties occurring in $\boldsymbol{y}_M$ are also broken similarly. We can note that $\mathrm{x}_m$'s and $\boldsymbol{y}_M$ are i.i.d. Rademacher random variables, which allows us to apply Khintchine's inequality, Then, by Khintchine's inequality,

$$\mathbb{E}\left[\left|\left(\frac{1}{M}\sum_{m=1}^{M}\sum_{j=1}^{i}\sigma_j^m\right) + \sum_{j=i+1}^{i+k}\sigma_j^M\right|\right] = \mathbb{E}\left[\left|\sum_{m=1}^{M}\mathrm{a}_m\mathrm{x}_m + \mathrm{b}_M\mathrm{y}_M\right|\right]$$

$$\geq \frac{1}{\sqrt{2}}\mathbb{E}\left[\left(\sum_{m=1}^{M}\mathrm{a}_m^2 + \mathrm{b}_M^2\right)^{1/2}\right].$$

By applying $\|\boldsymbol{z}\|_2 \geq \frac{1}{\sqrt{d}}\|\boldsymbol{z}\|_1$ for $\boldsymbol{z} \in \mathbb{R}^d$ twice, we get

$$\frac{1}{\sqrt{2}}\mathbb{E}\left[\left(\sum_{m=1}^{M}\mathrm{a}_m^2 + \mathrm{b}_M^2\right)^{1/2}\right] \geq \frac{1}{2}\mathbb{E}\left[\left(\sum_{m=1}^{M}\mathrm{a}_m^2\right)^{1/2} + \mathrm{b}_M\right] \geq \frac{1}{2}\mathbb{E}\left[\frac{1}{\sqrt{M}}\sum_{m=1}^{M}\mathrm{a}_m + \mathrm{b}_M\right].$$

Next, noticing that $\mathrm{a}_m$'s are i.i.d.,

$$\frac{1}{2}\mathbb{E}\left[\frac{1}{\sqrt{M}}\sum_{m=1}^{M}\mathrm{a}_m + \mathrm{b}_M\right] = \frac{1}{2}\mathbb{E}\left[\sqrt{M}\mathrm{a}_M + \mathrm{b}_M\right] = \frac{1}{2}\mathbb{E}\left[\frac{1}{\sqrt{M}}\left|\sum_{j=1}^{i}\sigma_j^M\right| + \left|\sum_{j=i+1}^{i+k}\sigma_j^M\right|\right]$$

$$\geq \frac{1}{64}\left(\sqrt{\frac{i}{M}} + \sqrt{k}\right). \qquad \text{(Lemma 12 from Rajput et al. (2020))}$$

Note that Lemma 12 from Rajput et al. (2020) has the requirement that $N \geq 256$. However, that requirement is for the entire lemma to hold, whereas we need only the first inequality in the lemma. For that, the requirement is simply $N \geq 8$. Further, note that for $N = 2, 4$, and 6 it can be manually verified that the required inequalities in Lemma 12 of Rajput et al. (2020) hold. Hence, this lemma holds for all even $N$.

The upper bound comes from Jensen's inequality:

$$\mathbb{E}\left[\left|\left(\frac{1}{M}\sum_{m=1}^{M}\sum_{j=1}^{i}\sigma_j^m\right) + \sum_{j=i+1}^{i+k}\sigma_j^M\right|\right] \leq \frac{1}{M}\mathbb{E}\left[\left|\sum_{m=1}^{M}\sum_{j=1}^{i}\sigma_j^m\right|\right] + \mathbb{E}\left[\left|\sum_{j=i+1}^{i+k}\sigma_j^M\right|\right]$$

$$\leq \frac{1}{M}\sqrt{\mathbb{E}\left[\left(\sum_{m=1}^{M}\sum_{j=1}^{i}\sigma_j^m\right)^2\right]} + \sqrt{\mathbb{E}\left[\left(\sum_{j=i+1}^{i+k}\sigma_j^M\right)^2\right]}$$

$$= \frac{1}{M}\sqrt{\sum_{m=1}^{M}\mathbb{E}\left[\left(\sum_{j=1}^{i}\sigma_j^m\right)^2\right]} + \sqrt{\mathbb{E}\left[\left(\sum_{j=i+1}^{i+k}\sigma_j^M\right)^2\right]}$$

(Since $\sum_{j=1}^{i}\sigma_j^m$ for $m = 1, 2\ldots$ are mean 0 and independent.)

$$= \sqrt{\frac{1}{M} \mathbb{E}\left[\left(\sum_{j=1}^{i} \sigma_j^M\right)^2\right]} + \sqrt{\mathbb{E}\left[\left(\sum_{j=i+1}^{i+k} \sigma_j^M\right)^2\right]}$$

$$= \sqrt{\frac{1}{M}\left(i + \sum_{j \neq l} \mathbb{E}[\sigma_j^M \sigma_l^M]\right)} + \sqrt{k + \sum_{j \neq l} \mathbb{E}[\sigma_j^M \sigma_l^M]}.$$

$\mathbb{E}[\sigma_j^M \sigma_l^M] \leq 0$ because $\sigma_j^M$ and $\sigma_l^M$ are negatively correlated. Hence, we get

$$\mathbb{E}\left[\left|\left(\frac{1}{M}\sum_{m=1}^{M}\sum_{j=1}^{i}\sigma_j^m\right) + \sum_{j=i+1}^{i+k}\sigma_j^M\right|\right] \leq \sqrt{\frac{i}{M}} + \sqrt{k},$$

as desired.

Next, it is left to show that for $0 \leq i \leq \frac{N}{2}$ and $0 \leq k \leq \frac{B}{2}$ satisfying $i + k \geq 1$, we have

$$\mathbb{P}\left(\sum_{m=1}^{M}\sum_{j=1}^{i}\sigma_j^m + M\sum_{j=i+1}^{i+k}\sigma_j^M > 0\right) = \mathbb{P}\left(\sum_{m=1}^{M}\sum_{j=1}^{i}\sigma_j^m + M\sum_{j=i+1}^{i+k}\sigma_j^M < 0\right) \geq \frac{1}{6}.$$

By symmetry, proving the equality is straightforward, and hence it is sufficient prove that

$$\mathbb{P}\left(\sum_{m=1}^{M}\sum_{j=1}^{i}\sigma_j^m + M\sum_{j=i+1}^{i+k}\sigma_j^M = 0\right) \leq \frac{2}{3}. \tag{149}$$

For this, it in fact suffices to show that

$$\mathbb{P}\left(\sum_{j=1}^{l}\sigma_j^1 = 0\right) \leq \frac{2}{3} \text{ for all } 1 \leq l \leq N - 1, \tag{150}$$

because (149) can be derived from (150). We first explain why (150) implies (149), and then show (150).

Suppose (150) is true. Then,

- Case 1: If $i = 0$, then (149) becomes $\mathbb{P}(\sum_{j=1}^{k}\sigma_j^M = 0) \leq \frac{2}{3}$, which is true due to (150).

- Case 2: If $M = 1$, then (149) becomes $\mathbb{P}(\sum_{j=1}^{i+k}\sigma_j^1 = 0) \leq \frac{2}{3}$, which is true due to (150).

- Case 3: If $i \geq 1$ and $M \geq 2$, then we can consider two events that partition the probability space:

  1. $E_1 := \{\sum_{m=2}^{M}\sum_{j=1}^{i}\sigma_j^m + M\sum_{j=i+1}^{i+k}\sigma_j^M = 0\}$. Conditioned on this event $E_1$,

  $$\mathbb{P}\left(\sum_{m=1}^{M}\sum_{j=1}^{i}\sigma_j^m + M\sum_{j=i+1}^{i+k}\sigma_j^M = 0 \mid E_1\right) = \mathbb{P}\left(\sum_{j=1}^{i}\sigma_j^1 = 0\right) \leq \frac{2}{3}$$

  due to independence of machines and (150).

  2. $E_2 := \{\sum_{m=2}^{M}\sum_{j=1}^{i}\sigma_j^m + M\sum_{j=i+1}^{i+k}\sigma_j^M \neq 0\}$. Conditioned on this event $E_2$, let $c := \sum_{m=2}^{M}\sum_{j=1}^{i}\sigma_j^m + M\sum_{j=i+1}^{i+k}\sigma_j^M$. Then,

  $$\mathbb{P}\left(\sum_{m=1}^{M}\sum_{j=1}^{i}\sigma_j^m + M\sum_{j=i+1}^{i+1}\sigma_j^M = 0 \mid E_2\right) = \mathbb{P}\left(\sum_{j=1}^{i}\sigma_j^1 = -c\right).$$

  However, by symmetry, $\mathbb{P}\left(\sum_{j=1}^{i}\sigma_j^1 = -c\right) = \mathbb{P}\left(\sum_{j=1}^{i}\sigma_j^1 = c\right) \leq \frac{1}{2}$.

From these two events, we conclude that (149) must hold.

It is now left to prove (150). It is clear that $\mathbb{P}(\sum_{j=1}^{i} \sigma_j^1 = 0) = 0$ for all odd $i$, so we assume that $i$ is even. Also note that $\mathbb{P}(\sum_{j=1}^{i} \sigma_j^1 = 0) = \mathbb{P}(\sum_{j=i+1}^{N} \sigma_j^1 = 0) = \mathbb{P}(\sum_{j=1}^{N-i} \sigma_j^1 = 0)$, since $\sigma_j^1$'s sum to zero. Therefore, for the rest of the proof, we can focus on even $i$'s in the range $2 \leq i \leq \frac{N}{2}$. Note that $\mathbb{P}(\sum_{j=1}^{i} \sigma_j^m = 0)$ is just the probability of having $\frac{i}{2}$ $+1$'s and $\frac{i}{2}$ $-1$'s in the first $i$ spots in a random shuffling of $\frac{N}{2}$ $+1$'s and $\frac{N}{2}$ $-1$'s. This is equivalent to choosing $\frac{i}{2}$ indices (for $+1$) out of the first $i$, and then choosing $\frac{N-i}{2}$ indices out of the remaining $N - i$. Thus,

$$\mathbb{P}\left(\sum_{j=1}^{i} \sigma_j^m = 0\right) = \frac{\binom{i}{i/2} \cdot \binom{N-i}{\frac{N-i}{2}}}{\binom{N}{N/2}}.$$

Note that the term above is a decreasing function of $i$ for $i \leq N/2$. Hence, putting $i = 2$ to the RHS we get

$$\mathbb{P}\left(\sum_{j=1}^{i} \sigma_j^m = 0\right) \leq \frac{\binom{2}{1} \cdot \binom{N-2}{\frac{N-2}{2}}}{\binom{N}{N/2}} = \frac{N}{2(N-1)} \leq \frac{2}{3},$$

where the inequality holds for $N \geq 4$.

### F.5 Proof of Lemma 15

$$\mathbb{E}[\|x_i - x_0\|] = \mathbb{E}\left[\left\|\frac{\eta}{MB} \sum_{j=0}^{i-1} \sum_{m=1}^{M} \sum_{k=jB+1}^{(j+1)B} \nu \sigma_k^m + (L1_{x_j<0} + \mu 1_{x_j \geq 0})x_j\right\|\right]$$

$$\leq \frac{\eta \nu}{B} \sqrt{\frac{iB}{M}} + \eta \mathbb{E}\left[\left\|\sum_{j=0}^{i-1}(L1_{x_j<0} + \mu 1_{x_j \geq 0})x_j\right\|\right] \qquad \text{(By Lemma 14)}$$

$$\leq \eta \nu \sqrt{\frac{i}{MB}} + \eta L \sum_{j=0}^{i-1} \mathbb{E}[\|x_j\|]$$

$$\leq \eta \nu \sqrt{\frac{i}{MB}} + i\eta L x_0 + \eta L \sum_{j=0}^{i-1} \mathbb{E}[\|x_j - x_0\|].$$

Let $h(i) := \eta \nu \sqrt{\frac{i}{MB}} + i\eta L x_0 + \eta L \sum_{j=0}^{i-1} h(j)$, starting with $h(0) = 0$. Then using induction, it can be seen that $\mathbb{E}[\|x_i - x_0\|] \leq h(i)$. Further, since $h(i)$ is an increasing function of $i$, we get

$$h(i) \leq \frac{\eta \nu \sqrt{\frac{i}{MB}} + i\eta L x_0}{1 - i\eta L}.$$

Since $i \leq \frac{N}{B}$ and $\eta \leq \frac{B}{513LN}$, we get that $\frac{1}{1-i\eta L} \leq \frac{513}{512}$, so $\mathbb{E}[\|x_i - x_0\|] \leq \frac{513}{512}\eta \nu \sqrt{\frac{i}{MB}} + \frac{513}{512} i\eta L x_0$.

## G  Proof of lower bound for local RR: homogeneous case (Theorem 4)

Recall that Theorem 4 gives the bound for local RR in the homogeneous setting, where all machines have the same local objectives. Similar to Theorem 3, we consider three step-size ranges and do case analysis for each of them. We construct functions for each corresponding step-size regime such that the convergence of local RR is "slow" for the functions on their corresponding step-size regime. The final lower bound is the minimum among the lower bounds obtained for the three regimes. More concretely, we will construct three one-dimensional functions $F_1(x)$, $F_2(x)$, and $F_3(x)$ satisfying $L$-smoothness (1), $\mu$-PŁ condition (2), and Assumption 2 such that[15]

---

[15]Again, the functions constructed in this theorem are $\mu$-strongly convex, which is stronger than $\mu$-PŁ required in Definition 1. Also, our functions satisfy Assumption 3 with $\tau = 0$, $\rho = 1$.

- Local RR on $F_1(x)$ with $\eta \leq \frac{1}{\mu N K}$ and initialization $y_0 = \frac{\nu}{\mu}$ results in

$$\mathbb{E}[F_1(y_{K, \frac{N}{B}})] = \Omega\left(\frac{\nu^2}{\mu}\right).$$

- Local RR on $F_2(x)$ with $\eta \geq \frac{1}{\mu N K}$ and $\eta \leq \frac{1}{1025 L N}$ and initialization $y_0 = 0$ results in

$$\mathbb{E}[F_2(y_{K, \frac{N}{B}})] = \Omega\left(\frac{\nu^2}{\mu M N K^2} + \frac{\nu^2 B}{\mu N^2 K^2}\right).$$

Note that the step-size range requires $K \geq 1025\kappa$, hence this lower bound occurs only in the "large-epoch" regime, i.e., $K \gtrsim \kappa$.

- Local RR on $F_3(x)$ with $\eta \geq \frac{1}{\mu N K}$ and $\eta \geq \frac{1}{1025 L N}$ and initialization $y_0 = 0$ results in

$$\mathbb{E}[F_3(y_{K, \frac{N}{B}})] = \Omega\left(\frac{\nu^2}{\mu M N K}\right).$$

Then, the three dimensional function $F([x, y, z]^\top) = F_1(x) + F_2(y) + F_3(z)$ will show bad convergence in any step-size regime. Furthermore,

$$\mu \boldsymbol{I} \preceq \min(\nabla^2 F_1, \nabla^2 F_2, \nabla^2 F_3)\boldsymbol{I} \preceq \nabla^2 F \preceq \max(\nabla^2 F_1, \nabla^2 F_2, \nabla^2 F_3)\boldsymbol{I} \preceq L\boldsymbol{I},$$

that is, if $F_1$, $F_2$ and $F_3$ are $\mu$-strongly convex and $L$-smooth, then so is $F$. Moreover, since the component functions in each coordinate are designed to satisfy Assumption 2 with $\nu$, the resulting three dimensional function $F$ also satisfies Assumption 2 with $\sqrt{3}\nu$.

Since the final lower bound is the minimum among the lower bounds obtained in the step-size ranges, the lower bound becomes $\Omega\left(\frac{\nu^2}{\mu M N K^2} + \frac{\nu^2 B}{\mu N^2 K^2}\right)$ if $K \geq 1025\kappa$ and $K \geq \frac{MB}{N}$ (this inequality is required to make sure $\frac{\nu^2 B}{\mu N^2 K^2} \leq \frac{\nu^2}{\mu M N K}$) and $\Omega\left(\frac{\nu^2}{\mu M N K}\right)$ otherwise.

In the subsequent subsections, we prove the lower bounds for $F_1$, $F_2$, and $F_3$ separately.

## G.1 LOWER BOUND FOR $\eta \leq \frac{1}{\mu N K}$

Consider the case where every function at every machine is the same: for all $i \in [N]$ and $m \in [M]$, $f_i^m(x) := \frac{\mu x^2}{2}$. Hence, $F_1(x) = \frac{\mu x^2}{2}$.

Since all $f_i^m$'s are the same, the local updates in all the machines are identical. Hence, for this subsection we omit the superscript for local machines. Let $x_{k,0}$ and $x_{k,N}$ denote the local iterates at the beginning and end of the $k$-th epoch. Then,

$$x_{k,N} = (1 - \eta\mu)^N x_{k,0}.$$

Initializing at $x_{1,0} = \frac{\nu}{\mu}$ and repeating this for $K$ epochs, we get that after $K$ epochs, the last iterate $y_{K, \frac{N}{B}} = x_{K,N}$ satisfies

$$y_{K, \frac{N}{B}} = x_{K,N} = (1 - \eta\mu)^{NK} \cdot \frac{\nu}{\mu} \geq \left(1 - \frac{1}{NK}\right)^{NK} \cdot \frac{\nu}{\mu} \geq \frac{\nu}{4\mu},$$

since $N \geq 2$, and $K \geq 1$. Hence, $F_1(y_{K, \frac{N}{B}}) = \Omega(\frac{\nu^2}{\mu})$.

## G.2 LOWER BOUND FOR $\eta \geq \frac{1}{\mu N K}$ AND $\eta \leq \frac{1}{1025 L N}$

For most part of this subsection, we consider iterates within a single epoch, and hence we will omit the subscripts denoting epochs. Let $x_0$ denote the iterate at the beginning of the epoch (which is the same across all the machines), and $x_i^m$ denote the $i$-th local iterate for machine $m$. After every $B$ local iterates, the server aggregates the local iterates $x_{iB}^m$, computes their average $y_i := \frac{1}{M} \sum_{m=1}^M x_{iB}^m$, and synchronizes all the machines $x_{iB}^m := y_i$.

Let $y_0$ denote the iterate at the beginning of the epoch (which is the same across all the machines $x_0^m = y_0$), and $x_i^m$ denote the $i$-th local iterate at machine $m$. In our construction, each machine will have the same set of component functions, that is, there will be no inter-machine deviation. We therefore omit the superscript $m$ from the local component functions $f_i^m$. The function we construct for the lower bound and its component functions are as follows:

$$F_2(x) := \frac{1}{N} \left( \sum_{i=1}^{\frac{N}{2}} f_{+1}(x) + \sum_{i=\frac{N}{2}+1}^{N} f_{-1}(x) \right), \text{ where}$$

$$f_{+1}(x) := (L1_{x \le 0} + \mu 1_{x>0}) \frac{x^2}{2} + \nu x, \text{ and}$$

$$f_{-1}(x) := (L1_{x \le 0} + \mu 1_{x>0}) \frac{x^2}{2} - \nu x$$

Note that the function $F_2(x) = (L1_{x \le 0} + \mu 1_{x>0}) \frac{x^2}{2}$ is $\mu$-strongly convex and $L$-smooth with minimizer at 0, and also satisfies Assumption 2.

Let $\sigma^m$ be a random permutation of $\frac{N}{2}$ +1's and $\frac{N}{2}$ −1's. Then, machine $m$ computes gradients on $f_{-1}$ and $f_{+1}$ in the order given by $\sigma^m$. Let $\sigma_j^m$ denote the $j$-th ordered element of $\sigma^m$. Then,

$$\nabla f_{\sigma_j^m}(x) = (L1_{x \le 0} + \mu 1_{x>0})x + \nu \sigma_j^m.$$

Hence, the last iterate of an epoch, $y_{\frac{N}{B}}$, is given by

$$y_{\frac{N}{B}} - y_0 = \sum_{i=0}^{\frac{N}{B}-1} \left( -\frac{\eta}{M} \sum_{m=1}^{M} \sum_{j=0}^{B-1} \nabla f_{\sigma_{iB+j+1}^m}(x_{iB+j}^m) \right)$$

$$= -\frac{\eta}{M} \sum_{i=0}^{\frac{N}{B}-1} \sum_{m=1}^{M} \sum_{j=0}^{B-1} ((L1_{x_{iB+j}^m \le 0} + \mu 1_{x_{iB+j}^m > 0})x_{iB+j}^m + \nu \sigma_{iB+j+1}^m)$$

$$= -\frac{\eta}{M} \sum_{i=0}^{\frac{N}{B}-1} \sum_{m=1}^{M} \sum_{j=0}^{B-1} (L1_{x_{iB+j}^m \le 0} + \mu 1_{x_{iB+j}^m > 0})x_{iB+j}^m,$$

where, in the last line, we used the fact that $\sum_{j=1}^{N} \sigma_j^m = 0$.

Recall that in the construction, each machine has the same component functions. Hence,

$$\mathbb{E}[y_{\frac{N}{B}} - y_0] = -\frac{\eta}{M} \mathbb{E} \left[ \sum_{i=0}^{\frac{N}{B}-1} \sum_{m=1}^{M} \sum_{j=0}^{B-1} (L1_{x_{iB+j}^m \le 0} + \mu 1_{x_{iB+j}^m > 0})x_{iB+j}^m \right]$$

$$= -\eta \sum_{i=0}^{\frac{N}{B}-1} \sum_{j=0}^{B-1} \mathbb{E} \left[ (L1_{x_{iB+j}^1 \le 0} + \mu 1_{x_{iB+j}^1 > 0})x_{iB+j}^1 \right], \tag{151}$$

where the last equality holds because the iterates $x_{iB+j}^m$ are identically distributed across different $m \in [M]$. Hence, we need to bound $\mathbb{E}[(L1_{x_{iB+j}^1 \le 0} + \mu 1_{x_{iB+j}^1 > 0})x_{iB+j}^1]$. As we did for Theorem 3, we want to prove that $\mathbb{E}[y_{\frac{N}{B}}]$ keeps increasing over an epoch, that is $\mathbb{E}[y_{\frac{N}{B}} - y_0] > 0$ when $y_0$ is close enough to the minimizer 0.

For this, we first consider the case where the first iterate $y_0$ of the epoch satisfies $y_0 \ge 0$. The $y_0 < 0$ case will be considered later. For the case $y_0 \ge 0$, we will show that whenever $y_0$ is small, the expected amount of update made in the $(iB+j+1)$-th iteration, $\mathbb{E}[(L1_{x_{iB+j}^1 \le 0} + \mu 1_{x_{iB+j}^1 > 0})x_{iB+j}^1]$, is negative if $i \le \lfloor \frac{N}{2B} \rfloor$ and $\frac{B}{4} \le j \le \frac{B}{2}$, and not too big otherwise.

We use the following lemmas, proven in Appendices H.1 and H.2, respectively.

**Lemma 16.** *For $y_0 \ge 0$, $0 \le i \le \lfloor \frac{N}{2B} \rfloor$, $\frac{B}{4} \le j \le \frac{B}{2}$, $\eta \le \frac{1}{1025LN}$, and $\frac{L}{\mu} \ge \frac{15375}{2}$,*

$$\mathbb{E}[(L1_{x_{iB+j}^1 \le 0} + \mu 1_{x_{iB+j}^1 > 0})x_{iB+j}^1] \le \frac{6}{7}Ly_0 - \frac{\eta L \nu}{1536} \left( \sqrt{\frac{iB}{M}} + \sqrt{j} \right).$$

**Lemma 17.** *For $y_0 \geq 0$, $0 \leq i \leq \frac{N}{B} - 1$, $0 \leq j \leq B - 1$, and $\eta \leq \frac{1}{1025LN}$,*

$$\mathbb{E}[(L1_{x^1_{iB+j} \leq 0} + \mu 1_{x^1_{iB+j} > 0}) x^1_{iB+j}] \leq \mu \left(1 + \frac{1025(iB+j)\eta L}{1024}\right) y_0 + \frac{1025\eta\mu\nu}{1024} \left(\sqrt{\frac{iB}{M}} + \sqrt{j}\right).$$

Next, we apply the two lemmas above in (151). For now, we consider the case $N/B \geq 2$. The case $B = N$ will be handled separately at the end of this subsection. For simplicity of notation, define $E_{iB+j} := \mathbb{E}[(L1_{x^1_{iB+j} \leq 0} + \mu 1_{x^1_{iB+j} > 0}) x^1_{iB+j}]$. We will divide the summation in (151) into four groups; for one of them we can apply Lemma 16, and for the other three we apply Lemma 17.

$$\mathbb{E}[y_{\frac{N}{B}} - y_0] = -\eta \sum_{i=0}^{\frac{N}{B}-1} \sum_{j=0}^{B-1} E_{iB+j}$$

$$= -\eta \left( \sum_{i=0}^{\lfloor \frac{N}{2B} \rfloor} \sum_{j=0}^{\frac{B}{4}-1} E_{iB+j} + \sum_{i=0}^{\lfloor \frac{N}{2B} \rfloor} \sum_{j=\frac{B}{4}}^{\frac{B}{2}} E_{iB+j} + \sum_{i=0}^{\lfloor \frac{N}{2B} \rfloor} \sum_{j=\frac{B}{2}+1}^{B-1} E_{iB+j} + \sum_{i=\lfloor \frac{N}{2B} \rfloor+1}^{\frac{N}{B}-1} \sum_{j=0}^{B-1} E_{iB+j} \right)$$

$$\geq -\eta \sum_{i=0}^{\lfloor \frac{N}{2B} \rfloor} \sum_{j=0}^{\frac{B}{4}-1} \left( \mu \left(1 + \frac{1025(iB+j)\eta L}{1024}\right) y_0 + \frac{1025\eta\mu\nu}{1024} \left(\sqrt{\frac{iB}{M}} + \sqrt{j}\right) \right)$$

$$- \eta \sum_{i=0}^{\lfloor \frac{N}{2B} \rfloor} \sum_{j=\frac{B}{4}}^{\frac{B}{2}} \left( \frac{6}{7} L y_0 - \frac{\eta L \nu}{1536} \left(\sqrt{\frac{iB}{M}} + \sqrt{j}\right) \right)$$

$$- \eta \sum_{i=0}^{\lfloor \frac{N}{2B} \rfloor} \sum_{j=\frac{B}{2}+1}^{B-1} \left( \mu \left(1 + \frac{1025(iB+j)\eta L}{1024}\right) y_0 + \frac{1025\eta\mu\nu}{1024} \left(\sqrt{\frac{iB}{M}} + \sqrt{j}\right) \right)$$

$$- \eta \sum_{i=\lfloor \frac{N}{2B} \rfloor+1}^{\frac{N}{B}-1} \sum_{j=0}^{B-1} \left( \mu \left(1 + \frac{1025(iB+j)\eta L}{1024}\right) y_0 + \frac{1025\eta\mu\nu}{1024} \left(\sqrt{\frac{iB}{M}} + \sqrt{j}\right) \right)$$

$$\geq -\eta \sum_{i=0}^{\lfloor \frac{N}{2B} \rfloor} \sum_{j=\frac{B}{4}}^{\frac{B}{2}} \left( \frac{6}{7} L y_0 - \frac{\eta L \nu}{1536} \left(\sqrt{\frac{iB}{M}} + \sqrt{j}\right) \right)$$

$$- \eta \sum_{i=0}^{\frac{N}{B}-1} \sum_{j=0}^{B-1} \left( \mu \left(1 + \frac{1025(iB+j)\eta L}{1024}\right) y_0 + \frac{1025\eta\mu\nu}{1024} \left(\sqrt{\frac{iB}{M}} + \sqrt{j}\right) \right), \tag{152}$$

where the last inequality is true because the RHS of the inequality in Lemma 17 is nonnegative. First consider the terms in (152) that involve $y_0$. Since $(iB+j)\eta L \leq \eta LN \leq \frac{1}{1025}$, $\mu \leq \frac{2L}{15375}$, $N/B \geq 2$, and $B \geq 4$, we have the following loose bound:

$$\sum_{i=0}^{\lfloor \frac{N}{2B} \rfloor} \sum_{j=\frac{B}{4}}^{\frac{B}{2}} \frac{6}{7} L + \sum_{i=0}^{\frac{N}{B}-1} \sum_{j=0}^{B-1} \mu \left(1 + \frac{1025(iB+j)\eta L}{1024}\right)$$

$$\leq \left(\left\lfloor \frac{N}{2B} \right\rfloor + 1\right) \left(\frac{B}{4} + 1\right) \frac{6}{7} L + \frac{N}{B} \cdot B \cdot \frac{2L}{15375} \left(1 + \frac{1}{1024}\right)$$

$$\leq \frac{N}{B} \cdot \frac{B}{2} \cdot \frac{6}{7} L + \frac{LN}{7680} \leq \frac{7LN}{8}. \tag{153}$$

We next bound the terms in (152) that involve summation of square roots. From $N/B \geq 2$, we have $\lfloor \frac{N}{2B} \rfloor \geq \frac{N}{3B}$ and $\lfloor \frac{N}{2B} \rfloor + 1 \geq \frac{N}{2B}$, so

$$\sum_{i=0}^{\lfloor \frac{N}{2B} \rfloor} \sum_{j=\frac{B}{4}}^{\frac{B}{2}} \left(\sqrt{\frac{iB}{M}} + \sqrt{j}\right) = \left(\frac{B}{4} + 1\right) \sqrt{\frac{B}{M}} \sum_{i=0}^{\lfloor \frac{N}{2B} \rfloor} \sqrt{i} + \left(\left\lfloor \frac{N}{2B} \right\rfloor + 1\right) \sum_{j=\frac{B}{4}}^{\frac{B}{2}} \sqrt{j}$$

$$\geq \frac{B^{3/2}}{4M^{1/2}} \int_0^{\lfloor \frac{N}{2B} \rfloor} \sqrt{t} dt + \frac{N}{2B} \int_{\frac{B}{4}-1}^{\frac{B}{2}} \sqrt{t} dt$$

$$\geq \frac{B^{3/2}}{6M^{1/2}} \left( \frac{N}{3B} \right)^{3/2} + \frac{N}{3B} \left[ \left( \frac{B}{2} \right)^{3/2} - \left( \frac{B}{4} \right)^{3/2} \right]$$

$$= \frac{N^{3/2}}{18\sqrt{3}M^{1/2}} + \frac{(2\sqrt{2}-1)NB^{1/2}}{24}. \tag{154}$$

For the other sum, we have

$$\sum_{i=0}^{\frac{N}{B}-1} \sum_{j=0}^{B-1} \left( \sqrt{\frac{iB}{M}} + \sqrt{j} \right) = \frac{B^{3/2}}{M^{1/2}} \sum_{i=0}^{\frac{N}{B}-1} \sqrt{i} + \frac{N}{B} \sum_{j=0}^{B-1} \sqrt{j}$$

$$\leq \frac{B^{3/2}}{M^{1/2}} \int_0^{\frac{N}{B}} \sqrt{t} dt + \frac{N}{B} \int_0^{B} \sqrt{t} dt$$

$$= \frac{2B^{3/2}}{3M^{1/2}} \left( \frac{N}{B} \right)^{3/2} + \frac{2N}{3B} B^{3/2}$$

$$= \frac{2N^{3/2}}{3M^{1/2}} + \frac{2NB^{1/2}}{3}. \tag{155}$$

Substituting the bounds (153), (154), and (155) into (152), and using $\mu \leq \frac{L}{40000}$,

$$\mathbb{E}[y_{\frac{N}{B}} - y_0] \geq -\frac{7\eta LN}{8} y_0 + \frac{\eta^2 L\nu}{1536} \left( \frac{N^{3/2}}{18\sqrt{3}M^{1/2}} + \frac{(2\sqrt{2}-1)NB^{1/2}}{24} \right)$$

$$- \frac{1025\eta^2\mu\nu}{1024} \left( \frac{2N^{3/2}}{3M^{1/2}} + \frac{2NB^{1/2}}{3} \right)$$

$$\geq -\frac{7\eta LN}{8} y_0 + \frac{\eta^2 L\nu}{240000} \left( \frac{N^{3/2}}{M^{1/2}} + NB^{1/2} \right). \tag{156}$$

For the other case $y_0 < 0$, we have the following lemma, a local RR counterpart of Lemma 13. In Appendix H.3, we prove the following:

**Lemma 18.** *If $\eta \leq \frac{1}{1025LN}$ and an epoch starts at $y_0 < 0$, then*

$$\mathbb{E}\left[ y_{\frac{N}{B}} \mid y_0 < 0 \right] \geq \left( 1 - \frac{7\eta LN}{8} \right) y_0.$$

*Further, if the first epoch of the algorithm is initialized at $0$, then for any starting iterate $y_0$ of any following epoch, we have $\mathbb{P}(y_0 \geq 0) \geq 1/2$.*

Using (156) and Lemma 18, we get

$$\mathbb{E}[y_{\frac{N}{B}}]$$

$$= \mathbb{P}(y_0 \geq 0)\mathbb{E}[y_{\frac{N}{B}} \mid y_0 \geq 0] + \mathbb{P}(y_0 < 0)\mathbb{E}[y_{\frac{N}{B}} \mid y_0 < 0]$$

$$\geq \mathbb{P}(y_0 \geq 0) \left( \left( 1 - \frac{7\eta LN}{8} \right) y_0 + \frac{\eta^2 L\nu}{240000} \left( \frac{N^{3/2}}{M^{1/2}} + NB^{1/2} \right) \right) + \mathbb{P}(y_0 < 0) \left( 1 - \frac{7\eta LN}{8} \right) y_0$$

$$\geq \left( 1 - \frac{7\eta LN}{8} \right) y_0 + \frac{\eta^2 L\nu}{480000} \left( \frac{N^{3/2}}{M^{1/2}} + NB^{1/2} \right).$$

Thus far, we have characterized the expected per-epoch update, starting from the initial iterate $y_0$ and iterating until the last iterate $y_{\frac{N}{B}}$ of the epoch. Now recall that we run the algorithm for $K$ epochs. Using $y_{k,i}$ to denote the $i$-th aggregated iterate of the $k$-th epoch, we get a lower bound on the expectation of the last iterate $y_{k,\frac{N}{B}}$ if we initialize at $y_{1,0} = 0$:

$$\mathbb{E}[y_{K,\frac{N}{B}}] \geq \left( 1 - \frac{7\eta LN}{8} \right)^K y_{1,0} + \frac{\eta^2 L\nu}{480000} \left( \frac{N^{3/2}}{M^{1/2}} + NB^{1/2} \right) \sum_{k=0}^{K-1} \left( 1 - \frac{7\eta LN}{8} \right)^k$$

$$= \frac{\eta^2 L\nu}{480000} \left( \frac{N^{3/2}}{M^{1/2}} + NB^{1/2} \right) \frac{1 - \left(1 - \frac{7\eta LN}{8}\right)^K}{\frac{7\eta LN}{8}}$$

$$= \frac{\eta \nu}{420000} \left( \frac{N^{1/2}}{M^{1/2}} + B^{1/2} \right) \left( 1 - \left(1 - \frac{7\eta LN}{8}\right)^K \right)$$

$$\geq \frac{\eta \nu}{420000} \left( \frac{N^{1/2}}{M^{1/2}} + B^{1/2} \right) \left( 1 - \left(1 - \frac{7L}{8\mu K}\right)^K \right). \qquad \text{(Since } \eta \geq \frac{1}{\mu NK}\text{)}$$

Note that since $\frac{L}{\mu} \geq 40000$ and $K \geq \frac{1025L}{\mu}$ (which is implied by $\frac{1}{\mu NK} \leq \eta \leq \frac{1}{1025LN}$),

$$1 - \left(1 - \frac{7L}{8\mu K}\right)^K \geq 1 - e^{-\frac{7L}{8\mu}} \geq 1 - e^{-35000} \approx 1.$$

Hence, we get from $\eta \geq \frac{1}{\mu NK}$ that

$$\mathbb{E}[y_{K, \frac{N}{B}}] = \Omega \left( \frac{\eta \nu N^{1/2}}{M^{1/2}} + \eta \nu B^{1/2} \right) = \Omega \left( \frac{\nu}{\mu M^{1/2} N^{1/2} K} + \frac{\nu B^{1/2}}{\mu NK} \right),$$

and by Jensen's inequality, we finally have

$$\mathbb{E}[F(y_{K, \frac{N}{B}})] \geq \frac{1}{2} \mathbb{E}[\mu y_{K, \frac{N}{B}}^2] = \Omega(\mu \mathbb{E}[y_{K, \frac{N}{B}}]^2) = \Omega \left( \frac{\nu^2}{\mu MNK^2} + \frac{\nu^2 B}{\mu N^2 K^2} \right).$$

Recall that, from the paragraph below Lemmas 16 and 17 to this point, we have assumed $N/B \geq 2$. We handle the case $B = N$ now. In this case, notice that all the $\sqrt{\frac{iB}{M}}$ terms that appear in (152) disappear, because we always have $i = 0$. Therefore, the proof goes through in the same why, modulo the fact that we do not have the terms that originate from the $\sqrt{\frac{iB}{M}}$ terms in (152). Therefore, we can show

$$\mathbb{E}[F(y_{K, \frac{N}{B}})] \geq \frac{1}{2} \mathbb{E}[\mu y_{K, \frac{N}{B}}^2] = \Omega(\mu \mathbb{E}[y_{K, \frac{N}{B}}]^2) = \Omega \left( \frac{\nu^2}{\mu NK^2} \right).$$

## G.3 Lower bound for $\eta \geq \frac{1}{\mu NK}$ and $\eta \geq \frac{1}{1025LN}$

Similar to earlier parts of the proof, here as well, each machine will have the same component functions, that is, there will be no inter-machine deviation. The proof uses a similar construction as Safran & Shamir (2020; 2021):

$$F_3(x) := \frac{1}{N} \left( \sum_{i=1}^{\frac{N}{2}} f_{+1}(x) + \sum_{i=\frac{N}{2}+1}^{N} f_{-1}(x) \right), \quad \text{where}$$

$$f_{+1}(x) := \frac{Lx^2}{2} + \nu x, \text{ and } f_{-1}(x) := \frac{Lx^2}{2} - \nu x.$$

Hence, $F_3(x) = \frac{Lx^2}{2}$, and has its minimizer at 0.

We first compute the expected "progress" over a given epoch. For simplicity, let us omit the subscript for epochs for now. Let $y_0$ denote the iterate at the beginning of the epoch (which is the same across all the machines $x_0^m = y_0$) and $x_i^m$ denote the $i$-th local iterate for machine $m$. After every $B$ local iterates, the server aggregates the local iterates $x_{iB}^m$, computes their average $y_i := \frac{1}{M} \sum_{m=1}^{M} x_{iB}^m$, and synchronizes all the machines $x_{iB}^m := y_i$.

For the epoch, let $\sigma^m$ be the permutation of $\frac{N}{2}$ +1's and $\frac{N}{2}$ −1's sampled by machine $m$. Upon receiving the aggregated iterate $x_{iB}^m = y_i$, each machine $m$ performs $B$ local updates. Unrolling the local update rules, the iterate after the $B$ updates (and before synchronization) can be written as follows:

$$x_{(i+1)B}^m = x_{(i+1)B-1}^m - \eta \nabla f_{\sigma_{(i+1)B}^m}(x_{(i+1)B-1}^m)$$

$$
\begin{aligned}
&= x^m_{(i+1)B-1} - \eta(Lx^m_{(i+1)B-1} + \nu\sigma^m_{(i+1)B}) \\
&= (1-\eta L)x^m_{(i+1)B-1} - \eta\nu\sigma^m_{(i+1)B} \\
&= \cdots \\
&= (1-\eta L)^B x^m_{iB} - \eta\nu \sum_{j=iB+1}^{(i+1)B}(1-\eta L)^{(i+1)B-j}\sigma^m_j \\
&= (1-\eta L)^B y_i - \eta\nu \sum_{j=iB+1}^{(i+1)B}(1-\eta L)^{(i+1)B-j}\sigma^m_j,
\end{aligned}
$$

After synchronization, we get

$$
y_{i+1} := \frac{1}{M}\sum_{m=1}^M x^m_{(i+1)B} = (1-\eta L)^B y_i - \frac{\eta\nu}{M}\sum_{m=1}^M\sum_{j=iB+1}^{(i+1)B}(1-\eta L)^{(i+1)B-j}\sigma^m_j.
$$

Unrolling the equation above from $y_{\frac{N}{B}}$ (the final iterate of the epoch, after synchronization) to $y_0$ (the starting iterate), we get

$$
\begin{aligned}
y_{\frac{N}{B}} &= (1-\eta L)^B y_{\frac{N}{B}-1} - \frac{\eta\nu}{M}\sum_{m=1}^M\sum_{j=N-B+1}^N(1-\eta L)^{N-j}\sigma^m_j \\
&= \cdots \\
&= (1-\eta L)^N y_0 - \sum_{i=1}^{\frac{N}{B}}(1-\eta L)^{N-iB}\frac{\eta\nu}{M}\sum_{m=1}^M\sum_{j=(i-1)B+1}^{iB}(1-\eta L)^{iB-j}\sigma^m_j \\
&= (1-\eta L)^N y_0 - \frac{\eta\nu}{M}\sum_{m=1}^M\sum_{j=1}^N(1-\eta L)^{N-j}\sigma^m_j.
\end{aligned}
$$

Then, by squaring both sides and taking expectations,

$$
\begin{aligned}
\mathbb{E}[y^2_{\frac{N}{B}}] &= (1-\eta L)^{2N}y_0^2 - \frac{2\eta\nu(1-\eta L)^N x_0}{M}\mathbb{E}\left[\sum_{m=1}^M\sum_{i=1}^N(1-\eta L)^{N-i}\sigma^m_i\right] \\
&\quad + \frac{\eta^2\nu^2}{M^2}\mathbb{E}\left[\left(\sum_{m=1}^M\sum_{i=1}^N(1-\eta L)^{N-i}\sigma^m_i\right)^2\right] \\
&= (1-\eta L)^{2N}y_0^2 + \frac{\eta^2\nu^2}{M^2}\mathbb{E}\left[\left(\sum_{m=1}^M\sum_{i=1}^N(1-\eta L)^{N-i}\sigma^m_i\right)^2\right],
\end{aligned}
\tag{157}
$$

where we used the fact that $\mathbb{E}[\sigma^m_i] = 0$. Further, because $\sigma^m$ and $\sigma^{m'}$ are independent' and identically distributed for different $m$ and $m'$, we get that

$$
\begin{aligned}
&\mathbb{E}\left[\left(\frac{\eta\nu}{M}\sum_{m=1}^M\sum_{i=1}^N(1-\eta L)^{N-i}\sigma^m_i\right)^2\right] \\
&= \sum_{m=1}^M\mathbb{E}\left[\left(\sum_{i=1}^N(1-\eta L)^{N-i}\sigma^m_i\right)^2\right] + \sum_{m\neq m'}\mathbb{E}\left[\sum_{i=1}^N(1-\eta L)^{N-i}\sigma^m_i\right]\mathbb{E}\left[\sum_{i=1}^N(1-\eta L)^{N-i}\sigma^{m'}_i\right] \\
&= M\mathbb{E}\left[\left(\sum_{i=1}^N(1-\eta L)^{N-i}\sigma^1_i\right)^2\right],
\end{aligned}
$$

where the last equality used the fact that $\mathbb{E}[\sigma^m_j] = 0$ for all $m \in [M]$ and $i \in [N]$, and that $\sigma^m$ are identically distributed. Since we only consider the permutation $\sigma^1$ (i.e., the one for machine 1) from

now on, we henceforth omit the superscript. Substituting this to (157) gives

$$\mathbb{E}[y_{\frac{N}{B}}^2] = (1 - \eta L)^{2N} y_0^2 + \frac{\eta^2 \nu^2}{M} \underbrace{\mathbb{E}\left[\left(\sum_{i=1}^{N}(1-\eta L)^{N-i}\sigma_i\right)^2\right]}_{:=\Phi}. \tag{158}$$

From (158), we have calculated the per-epoch expected update, because the final iterate $y_{\frac{N}{B}}$ is also the initial iterate of the next epoch. Recall that we run the algorithm for $K$ epochs. We now use $y_{k,i}$ to denote the iterate after the $i$-th communication round in the $k$-th epoch. Using (158), we get a lower bound on the expectation of the last iterate $y_{K,\frac{N}{B}}$ squared:

$$\mathbb{E}[y_{K,\frac{N}{B}}^2] = (1-\eta L)^{2NK} y_{1,0}^2 + \frac{\eta^2\nu^2}{M}\Phi\sum_{k=0}^{K-1}(1-\eta L)^{2Nk} \geq \frac{\eta^2\nu^2}{M}\Phi. \tag{159}$$

where the inequality used that we initialize at $y_{1,0} = 0$ and $\sum_{k=0}^{K-1}(1-\eta L)^{2Nk} \geq (1-\eta L)^0 = 1$. Next, we bound the expectation term, i.e., $\Phi$, defined in (158). Using Lemma 1 from Safran & Shamir (2020) with $n$ and $\alpha$ replaced with $N$ and $\eta L$ respectively, we have

$$\Phi \geq c \cdot \min\left\{\frac{1}{\eta L}, \eta^2 L^2 N^3\right\}, \tag{160}$$

for some universal constant $c > 0$. Using the fact that $\eta \geq \frac{1}{1025LN}$, it is easy to check that the RHS of (160) is lower-bounded by $\frac{c'}{\eta L}$, where $c' > 0$ is a universal constant. Combining (159) and (160) gives

$$\mathbb{E}[y_{K,\frac{N}{B}}^2] \geq \frac{\eta^2\nu^2}{M}\cdot\frac{c'}{\eta L} = \frac{c'\eta\nu^2}{LM}.$$

Also using the fact that $\eta \geq \frac{1}{\mu NK}$, we get

$$\mathbb{E}[F_3(y_{K,\frac{N}{B}})] = \frac{L}{2}\mathbb{E}[y_{K,\frac{N}{B}}^2] \geq \frac{c'\nu^2}{2\mu MNK}.$$

# H   PROOFS OF HELPER LEMMAS FOR APPENDIX G

## H.1   PROOF OF LEMMA 16

The proof of Lemma 16 is similar to its minibatch RR counterpart, Lemma 11. From the given $0 \leq i \leq \frac{N}{B} - 1$, $0 \leq j \leq B - 1$, define $k := iB + j$, in order to simplify notation. We also define $\mathcal{E} := \frac{1}{M}\left(\sum_{m=1}^{M}\sum_{l=1}^{iB}\sigma_l^m\right) + \sum_{l=iB+1}^{k}\sigma_l^1$.

By the law of total expectation we have

$$\mathbb{E}[(L1_{x_k^1\leq 0} + \mu 1_{x_k^1>0})x_k^1] = \mathbb{P}(\mathcal{E} > 0)\,\mathbb{E}\left[(L1_{x_k^1\leq 0} + \mu 1_{x_k^1>0})x_k^1\Big|\mathcal{E} > 0\right]$$

$$+ \mathbb{P}(\mathcal{E} \leq 0)\,\mathbb{E}\left[(L1_{x_k^1\leq 0} + \mu 1_{x_k^1>0})x_k^1\Big|\mathcal{E} \leq 0\right]$$

$$\leq \mathbb{P}(\mathcal{E} > 0)\,L\mathbb{E}\left[x_k^1\big|\mathcal{E} > 0\right] + \mathbb{P}(\mathcal{E} \leq 0)\,\mu\mathbb{E}\left[x_k^1\big|\mathcal{E} \leq 0\right], \tag{161}$$

where the last inequality used the fact that $(L1_{t\leq 0} + \mu 1_{t>0})t \leq Lt$ and $(L1_{t\leq 0} + \mu 1_{t>0})t \leq \mu t$ for any $t \in \mathbb{R}$.

We handle each of the two expectations in (161) separately. We first bound $\mathbb{E}\left[x_k^1\big|\mathcal{E} > 0\right]$. Recall from the definition of algorithm iterates that

$$x_k^1 - y_0 = x_{iB+j}^1 - y_0 = -\frac{\eta}{M}\sum_{m=1}^{M}\sum_{l=0}^{iB-1}\nabla f_{\sigma_{l+1}^m}(x_l^m) - \eta\sum_{l=iB}^{k-1}f_{\sigma_{l+1}^1}(x_l^1). \tag{162}$$

Expanding (162) using the definition of $\nabla f_{\sigma_{l+1}^m}$'s, we obtain

$$\mathbb{E}\left[x_k^1\big|\mathcal{E} > 0\right]$$

$$
= \mathbb{E}\left[ y_0 - \frac{\eta}{M}\sum_{m=1}^{M}\sum_{l=0}^{iB-1}\left(\nu\sigma_{l+1}^m + (L1_{x_l^m\le 0} + \mu1_{x_l^m>0})x_l^m\right) \right.
$$
$$
\left. - \eta\sum_{l=iB}^{k-1}\left(\nu\sigma_{l+1}^1 + (L1_{x_l^1\le 0}+\mu1_{x_l^1>0})x_l^1\right)\Big|\mathcal{E}>0 \right]
$$

$$
= \mathbb{E}\left[ y_0 - \frac{\eta}{M}\sum_{m=1}^{M}\sum_{l=0}^{iB-1}(L1_{x_l^m\le 0}+\mu1_{x_l^m>0})(x_l^m - y_0) \right.
$$
$$
\left. -\eta\sum_{l=iB}^{k-1}(L1_{x_l^1\le 0}+\mu1_{x_l^1>0})(x_l^1 - y_0)\Big|\mathcal{E}>0 \right] - \eta\nu\mathbb{E}\left[\mathcal{E}|\mathcal{E}>0\right]
$$
$$
-\mathbb{E}\left[\frac{\eta}{M}\sum_{m=1}^{M}\sum_{l=0}^{iB-1}(L1_{x_l^m\le 0}+\mu1_{x_l^m>0})y_0 + \eta\sum_{l=iB}^{k-1}(L1_{x_l^1\le 0}+\mu1_{x_l^1>0})y_0\Big|\mathcal{E}>0\right]
$$

$$
= y_0\mathbb{E}\left[1 - \frac{\eta}{M}\sum_{m=1}^{M}\sum_{l=0}^{iB-1}(L1_{x_l^m\le 0}+\mu1_{x_l^m>0}) - \eta\sum_{l=iB}^{k-1}(L1_{x_l^1\le 0}+\mu1_{x_l^1>0})\Big|\mathcal{E}>0\right]
$$
$$
-\mathbb{E}\left[\frac{\eta}{M}\sum_{m=1}^{M}\sum_{l=0}^{iB-1}(L1_{x_l^m\le 0}+\mu1_{x_l^m>0})(x_l^m - y_0) \right.
$$
$$
\left. +\eta\sum_{l=iB}^{k-1}(L1_{x_l^1\le 0}+\mu1_{x_l^1>0})(x_l^1 - y_0)\Big|\mathcal{E}>0\right] - \eta\nu\mathbb{E}\left[\mathcal{E}|\mathcal{E}>0\right]
$$

$$
\le y_0\mathbb{E}\left[1 - \frac{\eta}{M}\sum_{m=1}^{M}\sum_{l=0}^{iB-1}(L1_{x_l^m\le 0}+\mu1_{x_l^m>0}) - \eta\sum_{l=iB}^{k-1}(L1_{x_l^1\le 0}+\mu1_{x_l^1>0})\Big|\mathcal{E}>0\right]
$$
$$
+ \frac{\eta L(M-1)}{M}\sum_{l=0}^{iB-1}\mathbb{E}\left[|x_l^2 - y_0|\mid\mathcal{E}>0\right] + \frac{\eta L}{M}\sum_{l=0}^{iB-1}\mathbb{E}\left[|x_l^1 - y_0|\mid\mathcal{E}>0\right]
$$
$$
+ \eta L\sum_{l=iB}^{k-1}\mathbb{E}\left[|x_l^1 - y_0|\mid\mathcal{E}>0\right] - \eta\nu\mathbb{E}\left[\mathcal{E}|\mathcal{E}>0\right],
$$

where the last inequality used the fact that $(L1_{t\le 0}+\mu1_{t>0})t \le Lt$ and $(L1_{t\le 0}+\mu1_{t>0})t \le \mu t$ for any $t\in\mathbb{R}$; and that for different $m=2,\ldots,\overline{M}$, the local iterates $x_l^m$ are identically distributed conditioned on $\mathcal{E}>0$. Next, we use the fact that for any nonnegative random variable $\upsilon$ and event $\Delta$, we have that $\mathbb{E}[\upsilon|\Delta]\le\mathbb{E}[\upsilon]/\mathbb{P}(\Delta)$. Hence,

$$
\mathbb{E}\left[x_k^1|\mathcal{E}>0\right]
$$
$$
\le y_0\mathbb{E}\left[1 - \frac{\eta}{M}\sum_{m=1}^{M}\sum_{l=0}^{iB-1}(L1_{x_l^m\le 0}+\mu1_{x_l^m>0}) - \eta\sum_{l=iB}^{k-1}(L1_{x_l^1\le 0}+\mu1_{x_l^1>0})\Big|\mathcal{E}>0\right]
$$
$$
+ \frac{\eta L(M-1)}{M}\sum_{l=0}^{iB-1}\frac{\mathbb{E}\left[|x_l^2 - y_0|\right]}{\mathbb{P}(\mathcal{E}>0)} + \frac{\eta L}{M}\sum_{l=0}^{iB-1}\frac{\mathbb{E}\left[|x_l^1 - y_0|\right]}{\mathbb{P}(\mathcal{E}>0)} + \eta L\sum_{l=iB}^{k-1}\frac{\mathbb{E}\left[|x_l^1 - y_0|\right]}{\mathbb{P}(\mathcal{E}>0)}
$$
$$
- \eta\nu\mathbb{E}\left[\mathcal{E}|\mathcal{E}>0\right]
$$
$$
= y_0\mathbb{E}\left[1 - \frac{\eta}{M}\sum_{m=1}^{M}\sum_{l=0}^{iB-1}(L1_{x_l^m\le 0}+\mu1_{x_l^m>0}) - \eta\sum_{l=iB}^{k-1}(L1_{x_l^1\le 0}+\mu1_{x_l^1>0})\Big|\mathcal{E}>0\right]
$$
$$
- \eta\nu\mathbb{E}\left[\mathcal{E}|\mathcal{E}>0\right] + \eta L\sum_{l=0}^{k-1}\frac{\mathbb{E}\left[|x_l^1 - y_0|\right]}{\mathbb{P}(\mathcal{E}>0)}\tag{163}
$$

where the last equality used the fact that for different $m\in[M]$, the local iterates $x_l^m$ are identically distributed when they are not conditioned.

Next, we use Lemma 14 again to bound the conditional expectations that arise in (163). We restate the lemma for the reader's convenience.

**Lemma 14.** *For $m \in [M]$, let $\sigma^m$ be a random permutation of $\frac{N}{2}$ +1's and $\frac{N}{2}$ −1's. Then, for any $i \leq \frac{N}{2}$ and $k \leq \frac{B}{2}$, we have*

$$\frac{1}{64} \left( \sqrt{\frac{i}{M}} + \sqrt{k} \right) \leq \mathbb{E} \left[ \left| \left( \frac{1}{M} \sum_{m=1}^{M} \sum_{j=1}^{i} \sigma_j^m \right) + \sum_{j=i+1}^{i+k} \sigma_j^M \right| \right].$$

*Furthermore, for any $0 \leq i \leq N$ and $0 \leq k \leq N$ satisfying $i + k \leq N$, we have*

$$\mathbb{E} \left[ \left| \left( \frac{1}{M} \sum_{m=1}^{M} \sum_{j=1}^{i} \sigma_j^m \right) + \sum_{j=i+1}^{i+k} \sigma_j^M \right| \right] \leq \sqrt{\frac{i}{M}} + \sqrt{k}.$$

*Lastly, for any $0 \leq i \leq \frac{N}{2}$ and $0 \leq k \leq \frac{B}{2}$ satisfying $i + k \geq 1$, we have*

$$\mathbb{P} \left( \sum_{m=1}^{M} \sum_{j=1}^{i} \sigma_j^m + M \sum_{j=i+1}^{i+k} \sigma_j^M > 0 \right) = \mathbb{P} \left( \sum_{m=1}^{M} \sum_{j=1}^{i} \sigma_j^m + M \sum_{j=i+1}^{i+k} \sigma_j^M < 0 \right) \geq \frac{1}{6}.$$

Lemma 14 implies that $\mathbb{E}\left[\mathcal{E} | \mathcal{E} > 0\right] \in \left[ \frac{1}{64} \left( \sqrt{\frac{iB}{M}} + \sqrt{j} \right), \sqrt{\frac{iB}{M}} + \sqrt{j} \right]$ and $\mathbb{P}(\mathcal{E} > 0) = \mathbb{P}(\mathcal{E} < 0) \geq 1/6$. From this, we get

$$\eta\nu\mathbb{E}\left[\mathcal{E} | \mathcal{E} > 0\right] \geq \frac{\eta\nu}{64} \left( \sqrt{\frac{iB}{M}} + \sqrt{j} \right), \tag{164}$$

$$\frac{\mathbb{E}\left[|x_l^1 - y_0|\right]}{\mathbb{P}(\mathcal{E} > 0)} \leq 6\mathbb{E}\left[|x_l^1 - y_0|\right]. \tag{165}$$

Also, since $\eta \leq \frac{1}{LN}$ we have

$$1 - k\eta\mu \geq 1 - \frac{\eta}{M} \sum_{m=1}^{M} \sum_{l=0}^{iB-1} (L1_{x_l^m \leq 0} + \mu 1_{x_l^m > 0}) - \eta \sum_{l=iB}^{k-1} (L1_{x_l^1 \leq 0} + \mu 1_{x_l^1 > 0}) \geq 1 - \eta LN \geq 0,$$

which implies that

$$y_0\mathbb{E}\left[ 1 - \frac{\eta}{M} \sum_{m=1}^{M} \sum_{l=0}^{iB-1} (L1_{x_l^m \leq 0} + \mu 1_{x_l^m > 0}) - \eta \sum_{l=iB}^{k-1} (L1_{x_l^1 \leq 0} + \mu 1_{x_l^1 > 0}) \middle| \mathcal{E} > 0 \right] \leq (1 - k\eta\mu)y_0. \tag{166}$$

Substituting (164), (165), and (166) to (163), we obatin

$$\mathbb{E}\left[x_k^1 | \mathcal{E} > 0\right] \leq (1 - k\eta\mu)y_0 - \frac{\eta\nu}{64} \left( \sqrt{\frac{iB}{M}} + \sqrt{j} \right) + 6\eta L \sum_{l=0}^{k-1} \mathbb{E}\left[|x_l^1 - y_0|\right]. \tag{167}$$

Next, we have the following lemma that we can apply to $\mathbb{E}[|x_l^1 - y_0|]$. Proof of Lemma 19 can be found in Appendix H.4.

**Lemma 19.** *For $y_0 \geq 0$, $0 \leq i \leq \frac{N}{B} - 1$, $0 \leq j \leq B - 1$, and $\eta \leq \frac{1}{1025LN}$,*

$$\mathbb{E}[|x_{iB+j}^1 - y_0|] \leq \frac{1025}{1024} \eta\nu \left( \sqrt{\frac{iB}{M}} + \sqrt{j} \right) + \frac{1025}{1024}(iB + j)\eta Ly_0.$$

Applying this lemma to (167) and arranging the bounds (recall that $k := iB + j$), we get

$$\frac{1024}{1025} \sum_{l=0}^{k-1} \mathbb{E}\left[|x_l^1 - y_0|\right]$$

$$\leq \eta\nu \sum_{l=0}^{k-1} \left( \sqrt{\frac{B\lfloor l/B \rfloor}{M}} + \sqrt{l - B\lfloor l/B \rfloor} \right) + \eta L y_0 \sum_{l=0}^{k-1} l$$

$$= \eta\nu \sum_{l=0}^{iB-1} \left( \sqrt{\frac{B\lfloor l/B \rfloor}{M}} + \sqrt{l - B\lfloor l/B \rfloor} \right)$$

$$+ \eta\nu \sum_{l=iB}^{iB+j-1} \left( \sqrt{\frac{B\lfloor l/B \rfloor}{M}} + \sqrt{l - B\lfloor l/B \rfloor} \right) + \eta L y_0 \sum_{l=0}^{k-1} l$$

$$= \eta\nu \left( B \sum_{l=0}^{i-1} \sqrt{\frac{lB}{M}} + j\sqrt{\frac{iB}{M}} + i \sum_{l=0}^{B-1} \sqrt{l} + \sum_{l=0}^{j-1} \sqrt{l} \right) + \eta L y_0 \sum_{l=0}^{k-1} l. \tag{168}$$

The terms in (168) can be bounded using $\sum_{l=0}^{c-1} \sqrt{l} \leq \int_0^c \sqrt{t}dt$:

$$\frac{1024}{1025} \sum_{l=0}^{k-1} \mathbb{E}\left[ |x_l^1 - y_0| \right] \leq \eta\nu \left( \frac{2i^{3/2}B^{3/2}}{3M^{1/2}} + \frac{i^{1/2}jB^{1/2}}{M^{1/2}} + \frac{2iB^{3/2}}{3} + \frac{2j^{3/2}}{3} \right) + \frac{k^2 \eta L y_0}{2}$$

$$\leq \eta\nu \left( \frac{(2iB + 3j)}{3} \sqrt{\frac{iB}{M}} + \frac{4iBj^{1/2}}{3} + \frac{2j^{3/2}}{3} \right) + \frac{k^2 \eta L y_0}{2}$$

$$= \eta\nu \left( \frac{(2iB + 3j)}{3} \sqrt{\frac{iB}{M}} + \frac{(4iB + 2j)}{3} \sqrt{j} \right) + \frac{k^2 \eta L y_0}{2}$$

$$\leq \frac{4k\eta\nu}{3} \left( \sqrt{\frac{iB}{M}} + \sqrt{j} \right) + \frac{k^2 \eta L y_0}{2}, \tag{169}$$

where the second last inequality used $\frac{B}{4} \leq j$ (and hence $B^{1/2} \leq 2j^{1/2}$), and the last inequality used $2iB + 3j \leq 4(iB + j) = 4k$ and $4iB + 2j \leq 4(iB + j) = 4k$. Substituting (169) to (167), we get

$$\mathbb{E}\left[ x_k^1 | \mathcal{E} > 0 \right] \tag{170}$$

$$\leq (1 - k\eta\mu)y_0 - \frac{\eta\nu}{64} \left( \sqrt{\frac{iB}{M}} + \sqrt{j} \right) + \frac{1025k\eta^2 L\nu}{128} \left( \sqrt{\frac{iB}{M}} + \sqrt{j} \right) + \frac{3075k^2\eta^2 L^2 y_0}{1024}$$

$$= \left( 1 - k\eta\mu + \frac{3075k^2\eta^2 L^2}{1024} \right) y_0 - \left( \frac{1}{64} - \frac{1025k\eta L}{128} \right) \eta\nu \left( \sqrt{\frac{iB}{M}} + \sqrt{j} \right)$$

$$\leq \left( 1 - k\eta\mu + \frac{3k\eta L}{1024} \right) y_0 - \frac{\eta\nu}{128} \left( \sqrt{\frac{iB}{M}} + \sqrt{j} \right). \tag{171}$$

The last inequality here used $k\eta L \leq \eta L N \leq \frac{1}{1025}$, which follows from $\eta \leq \frac{1}{1025LN}$. Thus far, we have obtained an upper bound for $\mathbb{E}\left[ x_k^1 | \mathcal{E} > 0 \right]$.

Recall that there is another conditional expectation in (161) that we want to bound, namely $\mathbb{E}\left[ x_k^1 | \mathcal{E} \leq 0 \right]$. We bound it below, using the tools developed so far. For $i \leq \frac{N}{2B}$ and $\frac{B}{4} \leq j \leq \frac{B}{2}$,

$$\mathbb{E}\left[ x_k^1 | \mathcal{E} \leq 0 \right] = y_0 + \mathbb{E}\left[ x_k^1 - y_0 \mid \mathcal{E} \leq 0 \right]$$

$$\leq y_0 + \mathbb{E}\left[ |x_k^1 - y_0| \mid \mathcal{E} \leq 0 \right]$$

$$\leq y_0 + \frac{\mathbb{E}\left[ |x_k^1 - x_0| \right]}{\mathbb{P}(\mathcal{E} \leq 0)}$$

$$\leq y_0 + 6\mathbb{E}\left[ |x_k^1 - x_0| \right] \qquad \text{(Using Lemma 14)}$$

$$\leq y_0 + \frac{3075\eta\nu}{512} \left( \sqrt{\frac{iB}{M}} + \sqrt{j} \right) + \frac{3075k\eta L y_0}{512} \qquad \text{(Using Lemma 19)}$$

$$\leq \left( 1 + \frac{3075k\eta L}{512} \right) y_0 + \frac{3075\eta\nu}{512} \left( \sqrt{\frac{iB}{M}} + \sqrt{j} \right). \tag{172}$$

Using (171) and (172) in (161), we get that for $i \leq \frac{N}{2B}$ and $\frac{B}{4} \leq j \leq \frac{B}{2}$:

$$\mathbb{E}[(L1_{x_k^1 \leq 0} + \mu 1_{x_k^1 > 0}) x_k^1]$$

$$\leq \mathbb{P}(\mathcal{E} > 0) L \mathbb{E}[x_k^1 | \mathcal{E} > 0] + \mathbb{P}(\mathcal{E} \leq 0) \mu \mathbb{E}[x_k^1 | \mathcal{E} \leq 0]$$

$$\leq \mathbb{P}(\mathcal{E} > 0) L \left( \left(1 - k\eta\mu + \frac{3k\eta L}{1024}\right) y_0 - \frac{\eta\nu}{128} \left(\sqrt{\frac{iB}{M}} + \sqrt{j}\right)\right)$$

$$+ \mathbb{P}(\mathcal{E} \leq 0) \mu \left( \left(1 + \frac{3075k\eta L}{512}\right) y_0 + \frac{3075\eta\nu}{512} \left(\sqrt{\frac{iB}{M}} + \sqrt{j}\right)\right). \tag{173}$$

From Lemma 14, note that $\frac{1}{6} \leq \mathbb{P}(\mathcal{E} > 0) \leq \frac{5}{6}$ and $\frac{1}{6} \leq \mathbb{P}(\mathcal{E} \leq 0) \leq \frac{5}{6}$. We use these inequalities, along with $k\eta L \leq \eta L N \leq \frac{1}{1025}$ and $\frac{L}{\mu} \geq \frac{15375}{2}$, to bound the terms appearing in (173).

$$\mathbb{P}(\mathcal{E} > 0) L \left(1 - k\eta\mu + \frac{3k\eta L}{1024}\right) y_0 + \mathbb{P}(\mathcal{E} \leq 0) \mu \left(1 + \frac{3075k\eta L}{512}\right) y_0$$

$$\leq \frac{5}{6} L \left(1 + \frac{3}{1049600}\right) y_0 + \frac{5}{6} \cdot \frac{2L}{15375} \left(1 + \frac{3}{512}\right) y_0 \leq \frac{6}{7} L y_0. \tag{174}$$

We also have

$$- \mathbb{P}(\mathcal{E} > 0) \frac{\eta L \nu}{128} \left(\sqrt{\frac{iB}{M}} + \sqrt{j}\right) + \mathbb{P}(\mathcal{E} \leq 0) \frac{3075 \eta\mu\nu}{512} \left(\sqrt{\frac{iB}{M}} + \sqrt{j}\right)$$

$$\leq -\frac{\eta L \nu}{768} \left(\sqrt{\frac{iB}{M}} + \sqrt{j}\right) + \frac{5125 \eta\mu\nu}{1024} \left(\sqrt{\frac{iB}{M}} + \sqrt{j}\right)$$

$$\leq -\frac{\eta L \nu}{1536} \left(\sqrt{\frac{iB}{M}} + \sqrt{j}\right), \tag{175}$$

where we used the assumption $\frac{L}{\mu} \geq \frac{15375}{2}$. Substituting (174) and (175) to (173), we get

$$\mathbb{E}[(L1_{x_k^1 \leq 0} + \mu 1_{x_k^1 > 0}) x_k^1] \leq \frac{6}{7} L y_0 - \frac{\eta L \nu}{1536} \left(\sqrt{\frac{iB}{M}} + \sqrt{j}\right),$$

which finishes the proof.

## H.2 PROOF OF LEMMA 17

$$\mathbb{E}[(L1_{x_{iB+j}^1 \leq 0} + \mu 1_{x_{iB+j}^1 > 0}) x_{iB+j}^1] \leq \mu \mathbb{E}[x_{iB+j}^1]$$

$$= \mu y_0 + \mu \mathbb{E}[x_{iB+j}^1 - y_0]$$

$$\leq \mu y_0 + \mu \mathbb{E}[|x_{iB+j}^1 - x_0|]$$

$$\leq \mu y_0 + \frac{1025(iB+j)\eta L \mu}{1024} y_0 + \frac{1025 \eta\mu\nu}{1024} \left(\sqrt{\frac{iB}{M}} + \sqrt{j}\right),$$

where the last inequality used Lemma 19.

## H.3 PROOF OF LEMMA 18

We consider iterates within a single epoch, and hence we omit the subscripts denoting epochs. In our construction, each machine has the same set of component functions, that is, there will be no inter-machine deviation. We therefore omit the superscript $m$ from the local component functions. Consider the function

$$G_2(x) := \frac{1}{N} \left(\sum_{i=1}^{\frac{N}{2}} g_{+1}(x) + \sum_{i=\frac{N}{2}+1}^{N} g_{-1}(x)\right), \text{ where}$$

$$g_{+1}(x) := \frac{Lx^2}{2} + \nu x, \text{ and } g_{-1}(x) := \frac{Lx^2}{2} - \nu x.$$

Hence, $G_2(x) = \frac{Lx^2}{2}$. We prove the lemma by coupling iterates corresponding to $F_2$ and $G_2$. In particular, we perform local RR on $F_2$ and $G_2$ such that both start the given epoch at $y_0$ and all the corresponding machines use the same random permutations. Let $x^m_{iB+j,F}$ and $x^m_{iB+j,G}$ denote the iterates (for $(iB + j)$-th iteration at machine $m$) for $F_2$ and $G_2$ respectively. We use mathematical induction to prove that $x^m_{iB+j,F} \geq x^m_{iB+j,G}$ for all $i = 0, \dots, \frac{N}{B} - 1$ and $j = 0, \dots, B$ and machines $m = 1, \dots, M$. After that, we will use this to prove our desired statement $\mathbb{E}[y_{\frac{N}{B},F} \mid y_0 < 0] = \mathbb{E}[\frac{1}{M} \sum_{m=1}^{M} x^m_{N,F} \mid y_0 < 0] \geq (1 - \frac{7\eta LN}{8B})y_0$.

Let $\sigma^m$ be a random permutation of $\frac{N}{2}$ +1's and $\frac{N}{2}$ −1's. First we consider $i = 0$ and $0 \leq j \leq B$.

**Base case.**  For the base case, we know that $x^m_{0,F} \geq x^m_{0,G}$, since $x^m_{0,F} = x^m_{0,G} = y_0$ for all $m$.

**Inductive case.**  There can be three cases:

- Case 1: $x^m_{iB+j,F} \geq x^m_{iB+j,G} \geq 0$. Then,

$$
\begin{aligned}
&x^m_{iB+j+1,F} - x^m_{iB+j+1,G} \\
&= x^m_{iB+j,F} - x^m_{iB+j,G} - \eta(\nabla f_{\sigma^m_{iB+j+1}}(x_{iB+j,F}) - \nabla g_{\sigma^m_{iB+j+1}}(x_{iB+j,G})) \\
&= x^m_{iB+j,F} - x^m_{iB+j,G} - \eta\left(\mu x^m_{iB+j,F} + \nu\sigma^m_{iB+j+1} - Lx^m_{iB+j,G} - \nu\sigma^m_{iB+j+1}\right) \\
&= x^m_{iB+j,F} - x^m_{iB+j,G} - \eta\left(\mu x^m_{iB+j,F} - Lx^m_{iB+j,G}\right) \\
&= x^m_{iB+j,F}(1 - \eta\mu) - x^m_{iB+j,G}(1 - \eta L) \geq 0.
\end{aligned}
$$

- Case 2: $0 \geq x^m_{iB+j,F} \geq x^m_{iB+j,G}$. Then,

$$
\begin{aligned}
&x^m_{iB+j+1,F} - x^m_{iB+j+1,G} \\
&= x^m_{iB+j,F} - x^m_{iB+j,G} - \eta(\nabla f_{\sigma^m_{iB+j+1}}(x_{iB+j,F}) - \nabla g_{\sigma^m_{iB+j+1}}(x_{iB+j,G})) \\
&= x^m_{iB+j,F} - x^m_{i,G} - \eta\left(Lx^m_{iB+j,F} + \nu\sigma^m_{iB+j+1} - Lx^m_{iB+j,G} - \nu\sigma^m_{iB+j+1}\right) \\
&= x^m_{iB+j,F} - x^m_{iB+j,G} - \eta\left(Lx^m_{iB+j,F} - Lx^m_{iB+j,G}\right) \\
&= x^m_{iB+j,F}(1 - \eta L) - x^m_{iB+j,G}(1 - \eta L) \geq 0.
\end{aligned}
$$

- Case 3: $x_{iB+j,F} \geq 0 \geq x_{iB+j,G}$. Then,

$$
\begin{aligned}
&x^m_{iB+j+1,F} - x^m_{iB+j+1,G} \\
&= x^m_{iB+j,F} - x^m_{iB+j,G} - \eta(\nabla f_{\sigma^m_{iB+j+1}}(x_{iB+j,F}) - \nabla g_{\sigma^m_{iB+j+1}}(x_{iB+j,G})) \\
&= x^m_{iB+j,F} - x^m_{iB+j,G} - \eta\left(\mu x^m_{iB+j,F} + \nu\sigma^m_{iB+j+1} - Lx^m_{iB+j,G} - \nu\sigma^m_{iB+j+1}\right) \\
&= x^m_{iB+j,F} - x^m_{iB+j,G} - \eta\left(\mu x^m_{iB+j,F} - Lx^m_{iB+j,G}\right) \\
&= x^m_{iB+j,F}(1 - \eta\mu) - x^m_{iB+j,G}(1 - \eta L).
\end{aligned}
$$

  Note that since $\eta \leq \frac{1}{LN}$, we get that $x^m_{iB+j,F}(1 - \eta\mu) \geq 0$ and $x^m_{iB+j,G}(1 - \eta L) \leq 0$, which proves that $x^m_{iB+j+1,F} - x^m_{iB+j+1,G} \geq 0$.

Thus, we see that $x^m_{iB+j+1,F} \geq x^m_{iB+j+1,G}$ for all the three cases, which proves by mathematical induction that $x^m_{iB+j,F} \geq x^m_{iB+j,G}$ for all $0 \leq j \leq B$ and $i = 0$. Note that this implies that, the aggregated averages $y_{1,F} := \frac{1}{M} \sum_{m=1}^{M} x^m_{B,F}$ and $y_{1,G} := \frac{1}{M} \sum_{m=1}^{M} x^m_{B,G}$ satisfy $y_{1,F} \geq y_{1,G}$. Hence, after synchronization is complete, we get that for $i = 1$ and $j = 0$, $x^m_{iB+j,F} \geq x^m_{iB+j,G}$ for all machines $m$. This proves the base case for $i = 1$. Now, we can repeat the Inductive cases for $1 \leq j \leq B$ and $i = 1$, and thereby prove that $y_{2,F} \geq y_{2,G}$. Continuing on this process, we get that $x^m_{iB+j,F} \geq x^m_{iB+j,G}$ for all $0 \leq j \leq B$ and $0 \leq i \leq \frac{N}{B} - 1$, and consequently, $y_{\frac{N}{B},F} \geq y_{\frac{N}{B},G}$. Further, by linearity of expectation and gradient, it is easy to check that for any machine $m$,

$$\mathbb{E}[y_{\frac{N}{B},G}] = (1 - \eta L)^N y_0.$$

Using the result that $y_{\frac{N}{B},F} \geq y_{\frac{N}{B},G}$ which we proved above, we get $\mathbb{E}[y_{\frac{N}{B},F}] \geq (1 - \eta L)^N y_0$ for any initial iterate $y_0$. Specifically for $y_0 < 0$, this implies $\mathbb{E}[y_{\frac{N}{B},F} \mid y_0 < 0] \geq (1 - \eta L)^N y_0$.

Further, since $\eta L \leq \frac{1}{1025N}$, we have $(1 - \eta L)^N \leq 1 - \frac{7\eta L N}{8}$. This is because $1 - \frac{7zN}{8} - (1 - z)^N$ is nonnegative on the interval $\left[0, 1 - (7/8)^{\frac{1}{N-1}}\right]$, and $1 - (7/8)^{\frac{1}{N-1}} \geq \frac{1}{1025N}$ for all $N \geq 2$. To see why, note that $(1 - \frac{1}{1025(n-1)})^{n-1} \geq \frac{7}{8}$ for all $n \geq 2$, and this gives $1 - (7/8)^{\frac{1}{n-1}} \geq \frac{1}{1025(n-1)}$, which then implies $1 - (7/8)^{\frac{1}{n-1}} \geq \frac{1}{1025n}$ for all $n \geq 2$. Therefore, for $y_0 < 0$, we have $\mathbb{E}[y_{\frac{N}{B},F} \mid y_0 < 0] \geq (1 - \frac{7\eta L N}{8})y_0$.

For the last statement of the lemma, note that by symmetry of the function $G_2$, if we initialize Algorithm 1 at 0, then for any starting iterate of an epoch we have $\mathbb{P}(y_{0,G} \geq 0) \geq 1/2$. This combined with the fact that $y_{i,F} \geq y_{i,G}$ gives us that $\mathbb{P}(y_{0,F} \geq 0) \geq 1/2$.

### H.4 Proof of Lemma 19

$$\mathbb{E}[|x_{iB+j}^1 - y_0|]$$

$$= \mathbb{E}\left[\left\|\frac{\eta}{M}\sum_{m=1}^{M}\sum_{l=0}^{iB-1}\nu\sigma_{l+1}^m + (L1_{x_l^m<0} + \mu1_{x_l^m\geq0})x_l^m + \eta\sum_{l=iB}^{iB+j-1}\nu\sigma_{l+1}^1 + (L1_{x_l^1<0} + \mu1_{x_l^1\geq0})x_l^1\right\|\right]$$

$$\leq \eta\nu\left(\sqrt{\frac{iB}{M}} + \sqrt{j}\right) + \eta\mathbb{E}\left[\left\|\sum_{l=0}^{iB+j-1}(L1_{x_l^1<0} + \mu1_{x_l^1\geq0})x_l^1\right\|\right] \qquad \text{(By Lemma 14)}$$

$$\leq \eta\nu\left(\sqrt{\frac{iB}{M}} + \sqrt{j}\right) + \eta L\sum_{l=0}^{iB+j-1}\mathbb{E}[|x_l^1|]$$

$$\leq \eta\nu\left(\sqrt{\frac{iB}{M}} + \sqrt{j}\right) + (iB+j)\eta Ly_0 + \eta L\sum_{l=0}^{iB+j-1}\mathbb{E}[|x_l^1 - y_0|].$$

Now define

$$h(k) := \eta\nu\left(\sqrt{\frac{B\lfloor k/B\rfloor}{M}} + \sqrt{k - B\lfloor k/B\rfloor}\right) + k\eta Ly_0 + \eta L\sum_{l=0}^{k-1}h(l).$$

In terms of $i$ and $j$, note that $k$ corresponds to $k = iB + j$. Then using induction, it can be seen that $\mathbb{E}[|x_k^1 - y_0|] \leq h(k)$. Further, since $h(k)$ is an increasing function of $k$, we get

$$h(k) = \eta\nu\left(\sqrt{\frac{B\lfloor k/B\rfloor}{M}} + \sqrt{k - B\lfloor k/B\rfloor}\right) + k\eta Ly_0 + \eta L\sum_{l=0}^{k-1}h(l)$$

$$\leq \eta\nu\left(\sqrt{\frac{B\lfloor k/B\rfloor}{M}} + \sqrt{k - B\lfloor k/B\rfloor}\right) + k\eta Ly_0 + k\eta Lh(k)$$

$$\implies h(k) \leq \frac{\eta\nu\left(\sqrt{\frac{B\lfloor k/B\rfloor}{M}} + \sqrt{k - B\lfloor k/B\rfloor}\right) + k\eta Ly_0}{1 - k\eta L}.$$

Since $k \leq N$ and $\eta \leq \frac{1}{1025LN}$, we get that

$$\mathbb{E}[|x_{iB+j}^1 - y_0|] \leq \frac{1025}{1024}\eta\nu\left(\sqrt{\frac{iB}{M}} + \sqrt{j}\right) + \frac{1025}{1024}(iB+j)\eta Ly_0,$$

as desired.

# I  PROOF OF LOWER BOUND FOR LOCAL RR: HETEROGENEOUS CASE (PROPOSITION 5)

Recall that Proposition 5 gives the bound for local RR in the heterogeneous setting, where different machines have different local objectives. In this section, we construct examples where there is no intra-machine variation (i.e., $f_1^m = f_2^m = \cdots = f_N^m$ for all $m \in [M]$), but there is certain level of heterogeneity among different machines.

Similar to the other two lower bounds, we consider four step-size ranges and do case analysis for each of them. This time, we construct a single function $F$ for these step-size regimes such that the convergence of local RR is "slow" for $F$. The final lower bound is the minimum among the lower bounds obtained for the four regimes. More concretely, we will construct a one-dimensional function $F(x)$ satisfying $L$-smoothness (1), $\mu$-PŁ condition (2), and Assumption 3 such that[16]

- Local RR on $F(x)$ with $\eta \leq \frac{1}{8\mu NK}$ and initialization $y_0 = \frac{\tau}{\mu}$ results in

$$\mathbb{E}[F(y_{K, \frac{N}{B}})] = \Omega\left(\frac{\tau^2}{\mu}\right).$$

- Local RR on $F(x)$ with $\frac{1}{8\mu NK} \leq \eta \leq \frac{1}{8\mu B}$ and initialization $y_0 = 0$ results in

$$\mathbb{E}[F(y_{K, \frac{N}{B}})] = \Omega\left(\frac{\tau^2 B^2}{\mu N^2 K^2}\right).$$

- Local RR on $F(x)$ with $\frac{1}{8\mu B} \leq \eta \leq \frac{1}{\mu}$ and initialization $y_0 = 0$ results in

$$\mathbb{E}[F(y_{K, \frac{N}{B}})] = \Omega\left(\frac{\tau^2}{\mu}\right).$$

- Local RR on $F(x)$ with $\eta \geq \frac{1}{\mu}$ and initialization $y_0 = \frac{\tau}{\mu}$ results in

$$\mathbb{E}[F(y_{K, \frac{N}{B}})] = \Omega\left(\frac{\tau^2}{\mu}\right).$$

In the subsequent subsections, we prove the lower bounds for $F$ for the four step-size intervals.

## I.1  LOWER BOUND FOR $\frac{1}{8\mu NK} \leq \eta \leq \frac{1}{8\mu B}$ AND $\frac{1}{8\mu B} \leq \eta \leq \frac{1}{\mu}$

We first consider the two intervals in the middle, because they are more interesting cases. The global objective function $F$ and its local objective functions are as follows.

$$F(x) := \frac{1}{M}\left(\sum_{i=1}^{\frac{M}{2}} f_1(x) + \sum_{i=\frac{M}{2}+1}^{M} f_2(x)\right), \text{ where}$$

$$f_1(x) := -\tau x, \text{ and } f_2(x) := \mu x^2 + \tau x$$

In this construction, $M/2$ machines will have the function $f_1$ as their $N$ local component functions (and hence their local objective functions) and the other $M/2$ machines will have the function $f_2$.

Then, $B$ local RR updates in each machine corresponds to $B$ updates using either $f_1$ or $f_2$. If we start from $x_{iB}^m = y_i$, the $B$ local updates on machine $m$ result in

$$x_{(i+1)B}^m = \begin{cases} x_{iB}^m + \eta\tau B & \text{if machine } m \text{ has } f_1, \\ (1-2\eta\mu)^B x_{iB}^m - \eta\tau \sum_{j=0}^{B-1}(1-2\eta\mu)^j & \text{if machine } m \text{ has } f_2. \end{cases}$$

Taking the average of the $M$ machines, we get that

$$y_{i+1} = \frac{1}{M}\left(\frac{M}{2}(y_i + \eta\tau B) + \frac{M}{2}\left((1-2\eta\mu)^B y_i - \eta\tau\sum_{j=0}^{B-1}(1-2\eta\mu)^j\right)\right)$$

---

[16]Again, the functions constructed in this theorem are $\mu$-strongly convex, which is stronger than $\mu$-PL required in Definition 1. Also, our functions satisfy Assumption 2 with $\nu = 0$.

$$= \frac{1}{2} \left(1 + (1 - 2\eta\mu)^B\right) y_i + \frac{\eta\tau}{2} \left(B - \sum_{j=0}^{B-1}(1 - 2\eta\mu)^j\right).$$

Since there are total $\frac{NK}{B}$ such communication rounds over $K$ epochs, at the end of the run we have

$$y_{K, \frac{N}{B}} = \left(\frac{1}{2}\left(1 + (1 - 2\eta\mu)^B\right)\right)^{\frac{NK}{B}} y_0$$

$$+ \frac{\eta\tau}{2} \left(B - \sum_{j=0}^{B-1}(1 - 2\eta\mu)^j\right) \sum_{l=0}^{\frac{NK}{B}-1} \left(\frac{1}{2}\left(1 + (1 - 2\eta\mu)^B\right)\right)^l$$

$$= \frac{\eta\tau}{2} \left(B - \frac{1 - (1 - 2\eta\mu)^B}{2\eta\mu}\right) \frac{1 - \left(\frac{1}{2}\left(1 + (1 - 2\eta\mu)^B\right)\right)^{\frac{NK}{B}}}{1 - \frac{1}{2}\left(1 + (1 - 2\eta\mu)^B\right)}, \tag{176}$$

where we used initialization $y_0 = 0$. Having defined the function and calculated its last iterate (176), let us now handle the two step-size regimes separately.

We first consider $\frac{1}{8\mu NK} \leq \eta \leq \frac{1}{8\mu B}$. In this case, we exploit the fact that

$$1 - 2\eta\mu B + \eta^2\mu^2 B^2 \leq (1 - 2\eta\mu)^B \leq 1 - 2\eta\mu B + 4\eta^2\mu^2 B^2, \tag{177}$$

when $0 \leq \eta \leq \frac{1}{8\mu B}$. To see why, consider substituting $z := 2\eta\mu$. Then $h_1(z) := 1 - Bz + B^2 z^2 - (1 - z)^B$ has $h_1''(z) \geq 0$ on $z \in [0, 1]$, $h_1'(0) = 0$, and $h_1(0) = 0$, implying that $h_1(z) \geq 0$ on $z \in [0, 1]$. On the other hand, let $h_2(z) := 1 - Bz + \frac{B^2 z^2}{4} - (1 - z)^B$. If $B = 2$, then $h_2 \equiv 0$. If $B > 2$, then it can be checked that $h_2(z) \leq 0$ for small enough interval $[0, \frac{1}{4B}]$.

Using (177) on (176),

$$y_{K, \frac{N}{B}} = \frac{\eta\tau}{2} \left(B - \frac{1 - (1 - 2\eta\mu)^B}{2\eta\mu}\right) \frac{1 - \left(\frac{1}{2}\left(1 + (1 - 2\eta\mu)^B\right)\right)^{\frac{NK}{B}}}{1 - \frac{1}{2}\left(1 + (1 - 2\eta\mu)^B\right)}$$

$$\geq \frac{\eta\tau}{2} \left(B - \frac{1 - (1 - 2\eta\mu B + \eta^2\mu^2 B^2)}{2\eta\mu}\right) \frac{1 - \left(\frac{1}{2}\left(1 + 1 - 2\eta\mu B + 4\eta^2\mu^2 B^2\right)\right)^{\frac{NK}{B}}}{1 - \frac{1}{2}\left(1 + 1 - 2\eta\mu B + \eta^2\mu^2 B^2\right)}$$

$$= \frac{\eta\tau}{2} \left(\frac{\eta\mu B^2}{2}\right) \frac{1 - \left(1 - \eta\mu B + 2\eta^2\mu^2 B^2\right)^{\frac{NK}{B}}}{\eta\mu B - \frac{1}{2}\eta^2\mu^2 B^2}$$

$$\geq \frac{\eta^2\mu\tau B^2}{4} \cdot \frac{1 - \left(1 - \eta\mu B + \frac{1}{4}\eta\mu B\right)^{\frac{NK}{B}}}{\eta\mu B}$$

$$\geq \frac{\eta\tau B}{4} \left(1 - \left(1 - \frac{3\eta\mu B}{4}\right)^{\frac{NK}{B}}\right),$$

where we used $\eta\mu B \leq \frac{1}{8}$. Now, substituting $\eta \geq \frac{1}{8\mu NK}$ to above, we obtain

$$y_{K, \frac{N}{B}} \geq \frac{\eta\tau B}{4} \left(1 - \left(1 - \frac{3\eta\mu B}{4}\right)^{\frac{NK}{B}}\right)$$

$$\geq \frac{\tau B}{32\mu NK} \left(1 - \left(1 - \frac{3B}{32NK}\right)^{\frac{NK}{B}}\right)$$

$$\geq \frac{(1 - e^{-3/32})\tau B}{32\mu NK}.$$

Therefore, $F(y_{K, \frac{N}{B}}) = \Omega\left(\frac{\tau^2 B^2}{\mu N^2 K^2}\right)$.

Next, consider $\frac{1}{8\mu B} \leq \eta \leq \frac{1}{\mu}$. We take a close look at the term that appears in (176):

$$B - \sum_{j=0}^{B-1}(1 - 2\eta\mu)^j = B - \frac{1 - (1 - 2\eta\mu)^B}{2\eta\mu}.$$

For this term, we would like to find a lower bound which holds for all $\eta \in [\frac{1}{8\mu B}, \frac{1}{\mu}]$. To this end, consider substituting $z := 2\eta\mu$. Then, the function

$$h_3(z) := B - \sum_{j=0}^{B-1}(1-z)^j \tag{178}$$

is increasing on $z \in [\frac{1}{4B}, 1]$, and we have

$$h_3\left(\tfrac{1}{4B}\right) = B - 4B\left(1 - \left(1 - \tfrac{1}{4B}\right)^B\right) \le h_3(z), \text{ for all } z \in \left[\tfrac{1}{4B}, 1\right].$$

Using $B \ge 2$, $h_3(\frac{1}{4B})$ can be lower-bounded as

$$h_3\left(\tfrac{1}{4B}\right) \ge B - 4B\left(1 - \left(1 - \tfrac{1}{8}\right)^2\right) = \tfrac{B}{16}.$$

Next, for $z \ge 1$, the derivative of $h_3(z) := B - \sum_{j=0}^{B-1}(1-z)^j = B - \frac{1-(1-z)^B}{z}$ is $h_3'(z) = \frac{1-(1-z)^{B-1}((B-1)z+1)}{z}$. Since $B \ge 2$ is assumed to be even, it is easy to check that $h_3'(z) \ge 0$ for $z \ge 1$, which means that $h_3$ keeps increasing on $[1, 2]$. Therefore, we conclude that

$$\tfrac{B}{16} \le h_3\left(\tfrac{1}{4B}\right) \le h_3(z), \text{ for all } z \in \left[\tfrac{1}{4B}, 2\right],$$

and hence

$$B - \sum_{j=0}^{B-1}(1-2\eta\mu)^j \ge \frac{B}{16},$$

for all $\frac{1}{8\mu B} \le \eta \le \frac{1}{\mu}$. Using $y_{K,\frac{N}{B}}$ from (176), we get

$$y_{K,\frac{N}{B}} = \frac{\eta\tau}{2}\left(B - \sum_{j=0}^{B-1}(1-2\eta\mu)^j\right)\sum_{l=0}^{\frac{NK}{B}-1}\left(\frac{1}{2}\left(1+(1-2\eta\mu)^B\right)\right)^l$$

$$\ge \frac{\eta\tau}{2}\left(B - \sum_{j=0}^{B-1}(1-2\eta\mu)^j\right) \ge \frac{\eta\tau B}{32} \ge \frac{\tau}{256\mu}.$$

Here, we used the fact that $\sum_{l=0}^{\frac{NK}{B}-1}\left(\frac{1}{2}\left(1+(1-2\eta\mu)^B\right)\right)^l \ge \left(\frac{1}{2}\left(1+(1-2\eta\mu)^B\right)\right)^0 = 1$. Hence, we obtain $F(y_{K,\frac{N}{B}}) = \Omega(\frac{\tau^2}{\mu})$, finishing the proof.

## I.2 Lower bound for $\eta \le \frac{1}{8\mu NK}$ and $\eta \ge \frac{1}{\mu}$

We now conclude with the "extreme" step-size regimes. We consider the same function $F$ as in the previous subsection, but with a different initialization $y_0 = \frac{\tau}{\mu}$.

For $F$, recall from (176) that

$$y_{K,\frac{N}{B}} = \left(\frac{1}{2}\left(1+(1-2\eta\mu)^B\right)\right)^{\frac{NK}{B}} y_0$$

$$+ \frac{\eta\tau}{2}\left(B - \sum_{j=0}^{B-1}(1-2\eta\mu)^j\right)\sum_{l=0}^{\frac{NK}{B}-1}\left(\frac{1}{2}\left(1+(1-2\eta\mu)^B\right)\right)^l. \tag{179}$$

This time, we want to lower-bound the second term on the RHS of (179) with zero and focus on the first term. To this end, we revisit our discussion on $h_3$ (178). It is easy to check that $h_3$ is in fact increasing on the entire $[0, \infty)$, and $h_3(0) = 0$. This shows $B - \sum_{j=0}^{B-1}(1-2\eta\mu)^j \ge 0$ for any $\eta \ge 0$. Next, since $B$ is even, $\frac{1}{2}\left(1+(1-2\eta\mu)^B\right) \ge 0$ for any $\eta \ge 0$. This gives

$$y_{K,\frac{N}{B}} \ge \left(\frac{1}{2}\left(1+(1-2\eta\mu)^B\right)\right)^{\frac{NK}{B}} y_0 = \left(\frac{1}{2}\left(1+(1-2\eta\mu)^B\right)\right)^{\frac{NK}{B}}\frac{\tau}{\mu}. \tag{180}$$

First, consider the interval $0 \leq \eta \leq \frac{1}{8\mu NK}$. Recall from (177) that for this $\eta$,

$$(1 - 2\eta\mu)^B \geq 1 - 2\eta\mu B + \eta^2\mu^2 B^2 \geq 1 - 2\eta\mu B, \tag{181}$$

so

$$y_{K,\frac{N}{B}} \geq \left(\frac{1}{2}\left(1 + (1 - 2\eta\mu)^B\right)\right)^{\frac{NK}{B}} \frac{\tau}{\mu} \geq (1 - \eta\mu B)^{\frac{NK}{B}} \frac{\tau}{\mu} \geq \left(1 - \frac{B}{8NK}\right)^{\frac{NK}{B}} \frac{\tau}{\mu} \geq \frac{7\tau}{8\mu},$$

since $\frac{NK}{B} \geq 1$. Hence, $F(y_{K,\frac{N}{B}}) = \Omega(\frac{\tau^2}{\mu})$.

Finally, if $\eta \geq \frac{1}{\mu}$, then we have $(1 - 2\eta\mu)^B \geq 1$, so

$$y_{K,\frac{N}{B}} \geq \left(\frac{1}{2}\left(1 + (1 - 2\eta\mu)^B\right)\right)^{\frac{NK}{B}} \frac{\tau}{\mu} \geq \frac{\tau}{\mu}.$$

As a result, $F(y_{K,\frac{N}{B}}) = \Omega(\frac{\tau^2}{\mu})$.

