# OpenReview forum: "Minibatch vs Local SGD with Shuffling: Tight Convergence Bounds and Beyond"
_ICLR.cc/2022/Conference — ICLR 2022 Oral_

### Official Review · Reviewer_6QN8 · 2021-11-02

**Correctness:** 4
**Technical Novelty And Significance:** 3
**Empirical Novelty And Significance:** Not applicable
**Recommendation:** 8
**Confidence:** 4

**Main Review:**

The paper presents a deep theoretical study of popular-in-practice methods that utilize shuffling of the data points. This paper has a good structure, all sections are well-written and fully described. All assumptions and statements of theorems are clear and understandable.

However, assumption 2 is questionable:

Assumption 2 (Intra-machine deviation). There exists $\nu \geq 0$ such that for all $m \in[M]$ and $i \in[N]$,
$$
\left\|\nabla f_{i}^{m}(\boldsymbol{x})-\nabla F^{m}(\boldsymbol{x})\right\| \leq \nu, \text { for all } \boldsymbol{x} \in \mathbb{R}^{d}
$$

Does this assumption hold in practice? Is it not too strict to have a uniform bound for this difference?

The theory is clear and it is easy to follow the proofs. The theoretical part is solid and the importance and novelty of obtained results are significant. Upper and lower bounds explain methods' behaviors in detail. These results should be interesting for the optimization community and the machine learning community in general. However, it might be useful to have a table with a comparison with rates of other methods for Federated Learning. Also, it is interesting to compare these results with results from https://arxiv.org/pdf/2102.06704.pdf. In this paper, local methods with shuffling are also considered.

This paper does not have any experimental results. I understand that this work is theoretical, but simple and small experiments on toy models are desirable. It can be useful for readers to have graphical illustrations of methods' behavior.

The synchronized shuffling method is not clearly described. The notation $\sigma_{k}^{m}(i):=\sigma\left(\left(i+\frac{N}{M} \pi(m)\right) \bmod N\right)$ is quite confusing. Can you explain why you shift the permutation that way?  Despite the main idea of this approach being understandable, some technical details are not easy to get. It might be useful to add some clarifications in this part.





**Summary Of The Paper:**

This paper studies two popular algorithms: local SGD and minibatch SGD using the data shuffling technique, which means sampling without replacement. Authors provide analysis under PL condition and show that in some cases these methods with shuffling outperform classical local SGD and minibatch SGD. Additionally, this paper provides lower bounds for minibatch RR and local RR in homogeneous and heterogeneous settings. At the end of the paper, a new technique of using synchronized permutations is proposed. In an almost homogeneous setting, this approach can improve rates.

**Summary Of The Review:**

This paper introduces a deep and wide theoretical study of permutation-based variants of local SGD and minibatch SGD. Authors provide upper bounds, as well as lower bounds, which makes the contribution solid. The novelty and tightness of results are significant, so the paper should be accepted. My grade is "accept, good paper".

---

> ### Author Response · Authors · 2021-11-19
> **Response to Reviewer 6QN8**
>
> We sincerely appreciate the reviewer for the thoughtful feedback. Below, we reply to the concerns and questions raised by the reviewer.
>
> 1. Assumption 2: Is it not too strict to have a uniform bound for this difference?
> - For Assumption 2 to hold with small $\nu$ in a real distributed learning scenario, each local machine should have local data points that are similar to each other; for example, a device having many dog images, another having many cat images, etc. To allow devices to have diverse images, we would have to make $\nu$ larger.
>
> - As we briefly discussed in Appendix A, we agree that the uniform bound required by Assumption 2 is stronger than the common "bounded variance" assumption, i.e., for all $x \in \mathbb R^d$ and $m \in [M]$, $\frac{1}{N}\sum_{i=1}^N || \nabla f^m_i(x) - \nabla F^m(x) ||^2 \leq \sigma^2$. In many existing results, this assumption is employed to prove in-expectation convergence bounds. In contrast, our assumption requires that the deviation for each $i \in [N]$, not the average, is bounded by a constant. However, we would like to emphasize that we use this stronger assumption to prove *with-high-probability* upper bounds, which is a departure from the in-expectation bounds in the literature. In a nutshell, we make a stronger assumption to prove stronger results.
>
> - In our revision, we also added Appendix D.7 to discuss how to circumvent the uniform bound over the whole $\mathbb R^d$. There, we show that by modifying the theorem statements a bit, we can show the same bounds without the "entire $\mathbb R^d$" requirement.
>
> 2. Useful to have a table with a comparison with rates of other methods.
> - In our initial version of Appendix A, we had already presented comparisons to existing assumptions and convergence results, mainly focusing on with-replacement local SGD. If the paper gets accepted, in the camera-ready version we will update Appendix A with up-to-date results and also a summary table. Thank you for the helpful suggestion.
>
> 3. Comparison with results from https://arxiv.org/pdf/2102.06704.pdf.
> - Thank you for kindly linking the paper (Mishchenko et al. (2021)). In our initial version, we had already discussed and compared our results against this paper. Please see the discussion after Proposition 5 in Section 4.2. Also, the last part of Appendix A provides further details.
>
> 4. Simple and small experiments on toy models.
> - As per the reviewers' suggestions, we performed numerical experiments to evaluate the performance of the algorithms and support our theoretical findings. Please see Appendix C of the updated manuscript. We also believe that the paper will benefit from additional real data experiments. Since these experiments are a bit more time-consuming to run, we plan to add them into the camera-ready version of the document if the paper gets accepted.
>
> 5. The synchronized shuffling method is not clearly described.
> - We agree that the notation is not easy to digest at first glance. In order to provide clearer motivation and illustration for the technique, we newly added Appendix B in the revised version. Please see if the description in Appendix B clarifies your concerns.
>
> We appreciate the reviewer again for the insightful comments and questions. Please let us know if you have any remaining questions/comments.
>
> Best,
>
> Authors

---

> > ### Comment · Reviewer_6QN8 · 2021-11-25
> > **Response to authors**
> >
> > I thank the authors for clarifying the questions. They have improved the paper according to the comments and suggestions provided by all reviewers. I believe that this paper has good quality and it deserves high evaluation. I would like to keep my evaluation unchanged, which is "8: accept, good paper".

---

### Official Review · Reviewer_9cRr · 2021-11-03

**Correctness:** 4
**Technical Novelty And Significance:** 3
**Empirical Novelty And Significance:** 3
**Recommendation:** 8
**Confidence:** 4

**Main Review:**

Strength. 1. The analyzed problems have significant importance in practice and lack a thorough theoretical understanding yet. 2. The technical contribution for analyzing gradient sampling with dependence is original, the technique for bounding gradient noise in local RR is involved and interesting. 3. The provided bounds are tight in some sense and variants with better performance (linear speedup) are analyzed. 4. The lower bound for mini batch RR in small-epoch regime is informative in that it concludes the weakness of mini batch RR for K \lesssim \kappa.

Some other points. 1. There exists a gap of \kappa^2 between upper bounds and lower bounds, which can be important given that \kappa itself is relevant in the considered regime. It may help to explain more on parameter dependence when claiming tightness. 2. The lower bounds and upper bounds are in different notions of convergence, it may help understanding if there are some explanations there. 3. In theorem 3, it might be useful to explain on the connections among c_1, c_2 and \kappa, does it affect the regime K \ge c_2 \kappa, if c_2 has dependence on \kappa given that c_1 has dependence on \kappa. 4. In theorem 4, it also assumes that K \ge MB/N, does it has influence on the followed discussions on the choice of B. It seems to me the cutoff between parameter regimes is not very clear, could you explain more on it, like, similarly, does c_4 has relationship with \kappa? 5. Although this is a theory paper, giving some experiments to validate the analysis especially for the discussions on different regimes would be appreciated.


**Summary Of The Paper:**

This work analyzes the convergence rate of local and mini-batch Random Reshuffling. For \mu-PL and smooth objectives, It provides high probability upper bounds, and matching expected lower bounds, and a special variant of the random reshuffling that outperforms the previous tight bounds with \sqrt{M} speed up in some regimes.


**Summary Of The Review:**

I think this is a good paper with solid theoretical analysis and contribution, the analyzed problem is of great importance and needs more thorough understanding like this work.

---

> ### Author Response · Authors · 2021-11-19
> **Response to Reviewer 9cRr**
>
> We thank the reviewer for the detailed review and insightful comments. Thank you also for pointing out the strengths of our paper. Below, we address the comments raised by the reviewer:
>
> 1. The gap of $\kappa^2$ between upper bounds and lower bounds.
> - We agree that the $\kappa^2$ gap is important, and we believe that closing this gap is an important future direction. We tried to be clear about the gap in the beginning of Section 4 and also in the discussion paragraphs after Theorems 3 and 4. In the revision, we have added further clarifications to Section 1.1 that our bounds are loose by $\kappa^2$.
>
> 2. The lower bounds and upper bounds are in different notions of convergence.
> - The high-probability upper bounds shown in the paper can be used to prove matching in-expectation bounds. The upper bounds we prove are hence stronger than in-expectation bounds common in the literature. In the revision, we added a proof of this extension at the end of Appendix D.2.
>
> 3. Dependence of $c_1$, $c_2$, and $\kappa$ in Theorem 3.
> - We believe there may be a slight confusion here. Allow us to first clarify on the "dependence": the constants $c_1$ and $c_2$ as well as $c_3$ and $c_4$ that appear in Theorem 4 are numerical constants that do *not* depend on any problem parameters. In Theorem 3, the condition number $\kappa$ and $c_1$ are indeed related by the requirement $\kappa \geq c_1$. This means that the theorem holds for the function class $\mathcal F_{\rm cmp} (L, \mu, \nu, 0)$ if the ratio between $L$ and $\mu$ is above a certain numerical threshold. In other words, for our theorem to hold, the function class must have large enough $\kappa$ to allow sufficiently ill-conditioned functions. Please let us know if this clarifies your question; we are happy to elaborate more.
>
> 4. The assumption $K \ge MB/N$ in Theorem 4 and its relation to parameter regimes.
> - The requirement $K \ge MB/N$ we put is to make sure that the second term in the large-epoch bound, $\frac{\nu^2B}{\mu N^2 K^2}$, is smaller than the small-epoch bound $\frac{\nu^2}{\mu MNK}$. Therefore, after making the assumption explicit, the two "if" cases in Eq (8) become $K < \max ( c_4 \kappa, \frac{MB}{N} )$ and $K \geq \max ( c_4 \kappa, \frac{MB}{N} )$.
> Therefore, if $\kappa \gtrsim \frac{MB}{N}$, then the "phase transition" threshold in Eq (8) becomes $\Theta(\kappa)$. In the revised version, we have updated the theorem statement as well as the discussion.
>
> 5. Some experiments to validate the analysis.
> - As per the reviewers' suggestions, we performed numerical experiments to evaluate the performance of the algorithms and support our theoretical findings. Please see Appendix C of the updated manuscript. We also believe that the paper will benefit from additional real data experiments. Since these experiments are a bit more time-consuming to run, we plan to add them into the camera-ready version of the document if the paper gets accepted.
>
> Thank you again for the valuable feedback. Please let us know if you have any other comments/questions.
>
> Best,
>
> Authors

---

### Official Review · Reviewer_dmha · 2021-11-04

**Correctness:** 4
**Technical Novelty And Significance:** 4
**Empirical Novelty And Significance:** 4
**Recommendation:** 8
**Confidence:** 4

**Main Review:**

The majority of the manuscript is well-written and easy to understand. This paper is solely theoretical with promising guarantees. The authors showed that if $K \geq c_2\kappa$ where $c_2>0$ and $\kappa = \dfrac{L}{\mu}$, then their convergence rates are faster that the with-replacement counterpart. However, for the ill-conditioned case, the number of epochs (theoretically) will be huge to satisfy the mentioned assumptions.

It would be ideal if the authors could also provide some numerical results to demonstrate how well their proposed methods performs in practice.



**Summary Of The Paper:**

This paper proposes two variants of stochastic gradient algorithms without replacement. For smooth functions satisfying the PŁ condition, the authors showed that the proposed shuffling-based variants converge faster than their with-replacement counterparts (for the case with large number of epochs). Moreover, the authors showed that their provided convergence analysis is tight for the case with not large number of epochs.

**Summary Of The Review:**

This paper provides some promising theoretical results for the variants of stochastic gradient algorithms without replacement.

---

> ### Author Response · Authors · 2021-11-19
> **Response to Reviewer dmha**
>
> We thank the reviewer for their time and efforts, as well as their valuable comments.
>
> As per the reviewers' suggestions, we performed numerical experiments to evaluate the performance of the algorithms and support our theoretical findings. Please see Appendix C of the updated manuscript. We also believe that the paper will benefit from additional real data experiments. Since these experiments are a bit more time-consuming to run, we plan to add them into the camera-ready version of the document if the paper gets accepted.
>
> Again, we appreciate the reviewer for the feedback and the positive evaluation. Please let us know if you have any other comments/questions.
>
> Best,
>
> Authors

---

### Author Response · Authors · 2021-11-19
**Revision of paper**

Dear all,

We would like to announce that we made a revision to our submission, according to the reviewers' valuable comments. We have marked newly added sentences and paragraphs in green.

Some noteworthy changes include:
- We added Appendix C with some simple experiments on the 7 algorithms (minibatch/local RR with and without synchronized shuffling, single-machine RR, with-replacement minibatch/local SGD) considered in this paper. We use the "hard instance" constructed in Theorems 3 and 4 to examine the convergence speed of these algorithms. We are excited to see that the experimental results align quite nicely with our theoretical findings.
- We newly added Appendix B which contains a more detailed description of synchronized shuffling proposed in Section 5. We discuss why it is needed and illustrate how it works.
- At the end of Appendix D.2 (previously B.2), we added a discussion on how we can use the high-probability upper bounds in our paper to prove matching in-expectation upper bounds.
- We newly added Appendix D.7 to discuss ways we can prove our upper bounds without having to assume that Assumptions 2, 3, and 4 hold for the *entire* $\mathbb R^d$ space. We hope that this can help alleviate concerns regarding our intra- and inter-machine deviation assumptions.

We also plan to add the following to the camera-ready version, if the paper gets accepted:
- Additional experiments on real data.
- Up-to-date coverage of existing results, including a summary table.

Thanks,

Authors

---

### Decision · Program_Chairs · 2022-01-20

**Decision:**

Accept (Oral)

**Comment:**

This paper analyzes local SGD under the random reshuffling data selection setting. As is the case for standard random reshuffling, better rates are shown for local SGD when random reshuffling is used. This would already be a nice contribution to a line of work on random shuffling methods—but the paper goes beyond that by showing a matching lower bound and designing a (theoretically) better variant algorithm. The reviewers were all in agreement that this paper should be accepted (as a result not much further discussion happened after the original reviews), and I agree with this consensus. The modification seems to improve the paper, although I did not look through it in detail.